# Oncolytic viruses expressing MATEs facilitate target-independent T-cell activation in tumors

Malin Peter[1], Bettina Mundt[1], Arne Menze[1], Norman Woller[1], Valery Volk[2], Amanda M Ernst[1], Leon A Öhler[1], Steven R Talbot[3], Heiner Wedemeyer[1], Christine Falk[4], Friedrich Feuerhake[2], Thomas C Wirth[1] & Florian Kühnel [ID][1✉]

## Abstract

**Oncolytic viruses (OV) expressing bispecific T-cell engagers (BiTEs) are promising tools for tumor immunotherapy but the range of target tumors is limited. To facilitate effective T-cell stimulation with broad-range applicability, we established membrane-associated T-cell engagers (MATEs) harboring the protein transduction domain of the HIV-Tat protein to achieve non-selective binding to target cells. In vitro, MATEs effectively activated murine T cells and improved killing of MC38 colon carcinoma cells. Similarly, humanized MATEs activated T cells in PBMCs from human donors. In MC38-tumors in mice, MATE-expression by the oncolytic adenovirus Ad5/11 facilitated intratumoral T-cell activation, reduced tumor growth and prolonged survival accompanied by infiltration of tumor-directed CD8+ T cells and improved CD8/CD4 T-cell ratio. Absence of early T-cell activation in tumor draining lymph nodes suggests the safe applicability of this strategy. Furthermore, MATE-expression by Ad5/11 was capable of breaking resistance to αPD-1 checkpoint therapy thereby promoting T-cell/tumor cell proximity and clustering of CD8+ and CD4+ T cells. In summary, we demonstrated that MATE expressing OVs are powerful T-cell activating tools suitable for local immunotherapy of a broad range of tumors.**

**Keywords** MATE; T-cell Engager; CD3; Oncolytic Adenovirus; Tat-PTD
**Subject Categories** Cancer; Immunology

## Introduction

Oncolytic viruses (OV) are promising means for immunotherapy of cancer. Virus-mediated tumor cell lysis and concomitant release of tumor antigen creates an inflammatory tumor microenvironment promoting the induction of antitumor T-cell responses (Nakamori et al, 2004; Benencia et al, 2008; Prestwich et al, 2009). Induction of these T-cell responses is an essential prerequisite for abscopal

antitumor activity and for long-lasting therapeutic effects that have been observed in clinical studies with the oncolytic herpesvirus T-Vec (Senzer et al, 2009). Consequently, it has been a reasonable strategy to amplify antitumoral immune activation by providing OVs with additional immunostimulatory payloads. The frequently used GM-CSF, also incorporated in T-Vec, is well-established for the maturation of dendritic cells, but may also promote MDSCs and immunosuppression in tumor tissue (Bayne et al, 2012). Therefore, other strategies for immunological arming of oncolytic viruses employed alternatives such as CCL3, CCL5, FLT3L, and XCL-1 (Bernt et al, 2005; Ramakrishna et al, 2009; Li et al, 2011; Sánchez-Paulete et al, 2018), focusing on engagement and activation of immune cells involved in the crosspresentation of tumor antigen. In order to directly stimulate T-cell immunity, oncolytic viruses have also been provided with cytokines supporting a Th1 response such as IL2, IL12, TNFα, and interferons (Vigil et al, 2007; Bortolanza et al, 2009; Santos et al, 2018). Furthermore, secreted forms of checkpoint inhibitors have been used to focus T-cell activation to tumor tissue thereby reducing unwanted side effects of these agents (Engeland et al, 2014).

An excellent molecular target to directly engage and trigger T cells is CD3, an integral part of the T-cell receptor complex. It has been shown that adapter proteins referred to as bispecific T-cell engagers (BiTEs) stimulate antitumoral T-cell activity independent of TCR-specificity, MHC restriction and the need for costimulation which is achieved via binding to CD3ε on T cells and to a defined tumor-specific antigen present on the surface of target tumors (Löffler et al, 2000; Baeuerle and Reinhardt, 2009). Systemic administration of the CD19-specific BiTE Blinatumomab has shown robust results in the treatment of B cell lymphoma but is associated with immune-related adverse events such as cytokine release syndrome and neurotoxicity illustrating the potential hazards of this potent technology (Bargou et al, 2008; Topp et al, 2011). In contrast, efficacy of BiTEs has been rather limited in solid tumors due to the presence of mechanical barriers, dysfunctional vascularization and immunosuppressive conditions. To overcome these restrictions, oncolytic viruses expressing BiTEs have been generated to deliver BiTEs directly to the tumor tissue (Scott et al, 2018; Heidbuechel and Engeland, 2021). However, arming OVs with an antigen-dependent immune stimulator severely restricts the

[1]Department of Gastroenterology, Hepatology, Infectious Diseases and Endocrinology, Hannover Medical School, Hannover, Germany. [2]Department of Pathology, Hannover Medical School, Hannover, Germany. [3]Institute for Laboratory Animal Science, Hannover Medical School, 30625 Hannover, Germany. [4]Institute of Transplantation Immunology, Hannover Medical School, Hannover, Germany. ✉E-mail: kuehnel.florian@mh-hannover.de

range of potential tumor targets to tumors with diagnostically verified antigen expression to warrant full activity and to avoid any off-target T-cell activation. To address this limitation, we generated in our present study soluble non-target specific adapter proteins, referred to as membrane-associated T-cell engagers (MATEs) consisting of an αCD3scFv linked to the HIV Tat protein transduction domain. We showed that MATEs strongly activate murine and human T cells in vitro and promote direct killing of murine MC38 colon carcinoma cells. In s.c. MC38 tumors in syngeneic mice, MATE expression by the oncolytic adenovirus Ad5/11 significantly activated T cells and improved the CD8/CD4 T-cell ratio. No signs of systemic toxicity including surrogate markers of a cytokine release syndrome were observed, even when using Ad5/11 variants with enhanced MATE expression suggesting safe applicability of this target-independent approach. Furthermore, application led to reduced tumor growth and prolonged survival. As illustrated by long-term survival of the majority of treated mice, MATE-expressing virotherapy effectively sensitized for αPD-1 checkpoint blockade. Spatial multiplex immunohistochemical analyses suggested that MATE expression by virotherapy promotes the formation of T-cell clusters and fosters T-cell/tumor cell proximity. In summary, non-target selective MATEs for T-cell activation are well suited for immunological arming of oncolytic viruses thereby allowing virus application in a broad range of target tumors.

# Results

## Generation and functional analysis of recombinant membrane-associated T-cell engager proteins (MATEs)

Expression of BiTEs by oncolytic viruses compromises the range of potential target tumors. In antigen-negative tumor tissue, BiTE molecules would be ineffective without the presence of their cognate target or may even cause off-target activity associated with the risk of adverse events. To develop an alternative strategy of T-cell engager proteins for intratumoral application in a broad range of tumors consistent with the potentially unlimited application potential of oncolytic viruses, we generated adapter proteins referred to as MATEs which are capable of binding to cells without requiring the presence of a particular cell surface antigen on potential target cells. In order to achieve a transient and effective but at the same time non-specific adhesion to target cells, we used the positively charged protein transduction domain of the HIV-Tat protein (Tat-PTD), which facilitates adherence to cellular membranes by electrostatic interactions (Schwarze et al, 1999). Interestingly, it has been reported that the Tat-PTD is capable of penetrating cellular membranes whereby the underlying mechanisms are not fully clear (Takeuchi and Futaki, 2016). Furthermore, it is well known that Tat-PTD conjugated proteins can be subject to rapid resorption which should help to prevent that MATEs, if secreted excessively may become systemic (Murriel and Dowdy, 2006). For investigation in mice, the Tat-PTD in MATEs was linked to a CD3ε-specific single chain antibody fragment (Gilliland et al, 1996). Both functional domains were separated by a non-functional IgG2b-derived domain to provide structural flexibility and distance to minimize interference between the two functional domains. To further enhance affinity of effector and target cells in order to

strengthen the immunologic synapse we also generated a trimerized version of MATEs by integrating a T4-fibritin trimerization motif as previously described (Kashentseva et al, 2002). The genetic setup and structure of MATEs is illustrated in Fig. 1A. For first functional investigations, MATEs were expressed by stably-transduced HEK293 cells and isolated from cell supernatants. The monomeric MATE (αCD3$_{TAT}$) and the trimer version (αCD3$_{TAT}$-Trimer) were detected by western blot analysis at the appropriate size of approx. 62 kD and 65 kD, respectively. Trimerization was confirmed by a Blue-Native PAGE and subsequent western blot (Fig. EV1A). First, we examined the binding of MATEs to CD3ε on T cells. We incubated freshly isolated murine splenocytes with purified MATE proteins and investigated T-cell binding (Fig. 1B). MATEs bound equally well to both CD4$^+$ and CD8$^+$ T cells. The results of flow cytometric analyses also show that MATEs bind equivalently well to CD8$^+$ T-cell subsets (Fig. EV1B). For functional assessment, we investigated T-cell activation in murine splenocytes after treatment with purified MATEs and control proteins. The central aim of T-cell engager proteins such as BiTEs is to mediate a close association of T cells with potential target tumor cells resulting in T-cell activation independent of specific TCR/MHC interactions. To confirm whether cell binding function of the Tat-PTD in MATEs is actually required for T cell activation, we also generated an αCD3-scFv in both monomeric and trimeric version lacking the Tat-PTD. Incubation with MATEs led to significantly elevated activation markers CD25 and CD69 in CD4$^+$ and CD8$^+$ T cells compared to the control proteins lacking the Tat-PTD (Fig. 1C). An increase of IFNγ production was exclusively observed in splenocytes coincubated with MATEs. It cannot be excluded that the Tat-PTD domain contributes to the binding of MATEs to T cells though we could not detect physical binding of Tat-PTD containing proteins without αCD3-scFv domain by FACS analysis. However, the membrane binding properties of Tat-PTD are well-established and we showed that the Tat-PTD was essential for the function of MATEs (Fig. 1C). Since the binding of MATEs to T cells could be easily visualized, this may reflect the rapid resorption of Tat-PTD-containing proteins such as MATEs. MATE-mediated activation of both CD4$^+$ and CD8$^+$ T cells was dose-dependent (Fig. EV1C). When comparing equivalent amounts of MATEs at higher concentrations, T-cell activation by the trimerized MATE (αCD3$_{TAT}$-Trimer) appeared to be slightly stronger compared to the monomer suggesting that the increased avidity of the trimer may support CD3 clustering more efficiently. Though stimulation by MATEs led to a significant activation-related increase of CD25 in the CD4$^+$ T-cell population, the number of FoxP3 positive cells in this subset was significantly lower compared to the untreated control suggesting that MATE-dependent T-cell activation did not induce Tregs (Fig. EV1D). Proliferation is a further important indicator of T-cell activation and function. We observed a strong increase of Ki67 in CD4$^+$ and CD8$^+$ T cells after incubation of splenocytes with MATEs (Fig. 1D) demonstrating that MATEs effectively promoted T-cell proliferation. To confirm whether the observed T-cell activation indeed reflects an increase in T-cell function, we investigated T-cell mediated cytotoxicity against tumor cells. For this purpose, we performed a T-cell killing assay by adding MATEs to a coculture of MC38 tumor cells and splenocytes. MATEs led to a significant increase in T-cell mediated cytotoxicity as demonstrated by increasing amounts of dead tumor cells (Fig. 1E). Concerning our

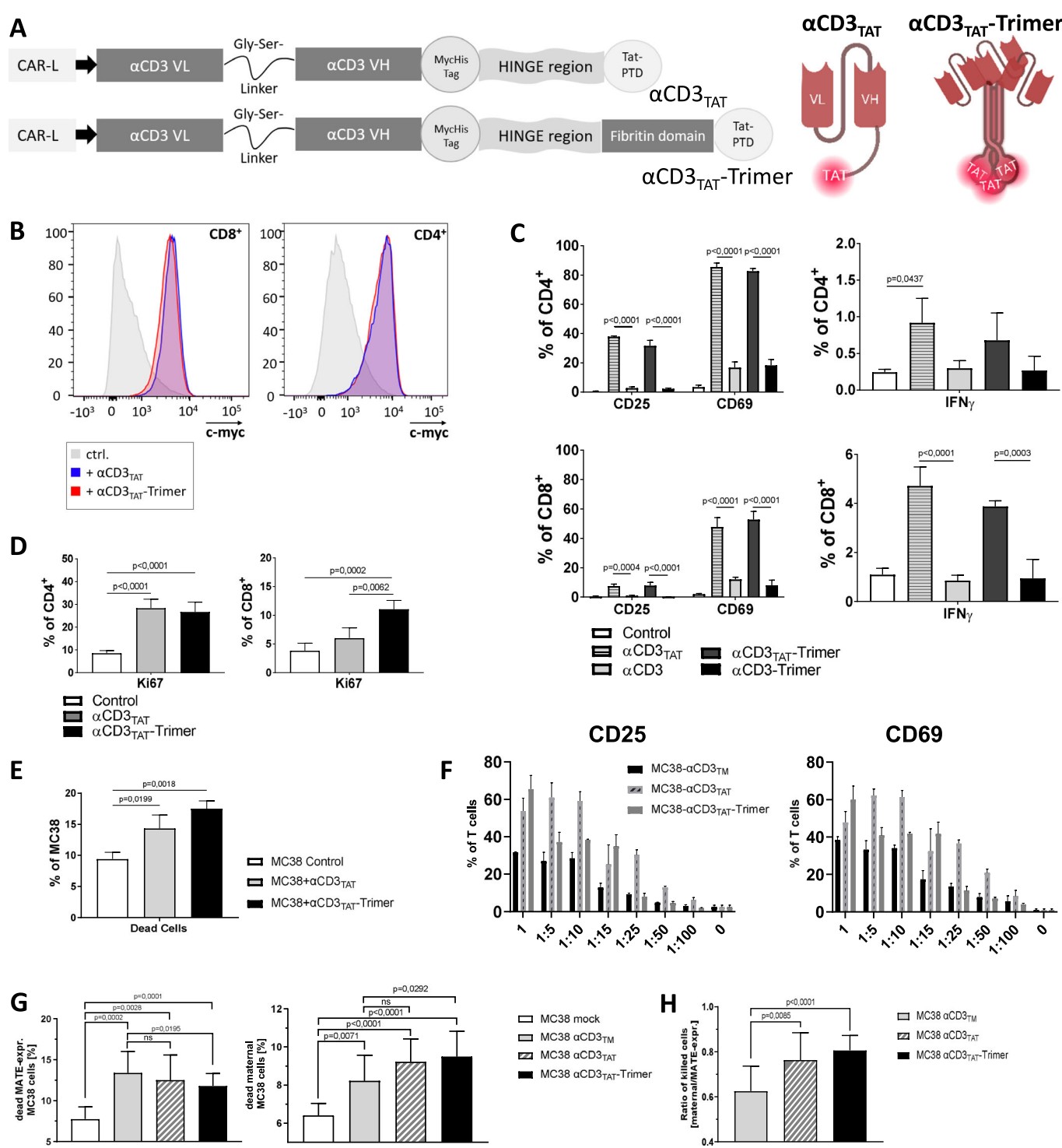

aim to arm oncolytic viruses with MATEs, we further hypothesized that the release of MATEs by infected cells may confer a bystander T-cell activation when distributed and bound to neighboring cells thus leading to an overall improved T-cell activation. To investigate this, we generated stable MATE-expressing MC38 murine colon carcinoma cells and MC38 control cells expressing an αCD3scFv linked to a transmembrane helix (αCD3$_{TM}$) to prevent release (Fig. EV1E,F). Coculture experiments with splenocytes showed that

both MATEs and transmembrane T-cell engager proteins, when expressed by MC38 cells activated murine T cells and induced T-cell proliferation (Fig. EV1G,H). Whereas CD8$^+$ T cells were comparably activated by the tested engager variants, MATEs more effectively activated CD4$^+$ T cells compared with the transmembrane version of αCD3. To investigate potential bystander activation, MATE-expressing MC38 cells or cells expressing αCD3$_{TM}$ were diluted with increasing amounts of parental MC38

◄ **Figure 1. Binding of MATEs to murine CD4⁺ and CD8⁺ T cells leads to activation and proliferation of T cells and mediates cytotoxicity against tumor cells in vitro.**

(A) Schematic illustration of MATE proteins consisting of a leader peptide (CAR-L), a single chain variable fragment specific for murine CD3, an IgG2b-derived hinge region and the Tat protein transduction domain (Tat-PTD). A trimeric version of MATEs was generated using a fibritin domain. (B) Murine splenocytes were incubated with 500 ng of purified MATEs. Binding of MATEs to CD4⁺ and CD8⁺ T cells was determined by flow cytometry. Representative plots are shown. (C) After incubation of splenocytes for 24 h with 1 μg purified MATEs, or control MATEs lacking the Tat-PTD, T cells were stained for activation markers and assessed by flow cytometry (left panels), IFNγ was assessed by ICS (right panels). (D) Murine splenocytes were incubated for 48 h with 1 μg of purified MATEs and T cell proliferation was determined by intranuclear staining of Ki-67. (E) Cytotoxicity was assessed 24 h after coculture of MC38 tumor cells and murine splenocytes (target:splenocyte ratio of 1:10) with 1 μg of purified MATEs or left without (Control). Dead MC38 cells were measured by Zombie Green™ Fixable Viability Kit and flow cytometry. (F) MC38 cells stably expressing MATEs or a transmembrane αCD3scFv (αCD3_{TM}) were mixed with increasing amounts of paternal cells according to the ratio provided on the x-axis, and with murine splenocytes at a target:splenocyte ratio of 1:10. After 24 h of coculture, T-cell activation was investigated. Each experiment (C–F) was measured in triplicates, with an exception for the control group in 1D with 6 measurements (mean ± SD; one-way ANOVA with Tukey's post hoc analysis). (G) Transgenic MATE-expressing or αCD3_{TM}-expressing MC38 cells were labeled with CFSE and seeded together with paternal MC38 cells at a ratio of 1:50. Target cells were co-cultivated with isolated splenocytes for 48 h. All cells were collected, dead cells were stained with Zombie NIR™ and analyzed via flow cytometry. The figure shows the killed transgenic MC38 cells (left panel) and the killed paternal non-transgenic MC38 cells (number of replicates was $n = 6$ for controls, all other groups $n = 12$; mean ± SD; two-tailed, unpaired t-test was used). The groups αCD3_{TAT}-Trimer vs. mock and αCD3_{TAT}-Trimer vs αCD3_{TM} were compared using Mann-Whitney test. The individual ratio of dead paternal MC38 cells to transgenic, MATE-expressing cells was determined and is shown in (H) (number of replicates was $n = 12$; mean ± SD; p values calculated with two-tailed unpaired t-test). Source data are available online for this figure.

and were then coincubated with splenocytes. Release of MATEs led to an increase in T-cell activation compared to the membrane-bound αCD3 as determined by activation markers CD69 and CD25 (Fig. 1F). By in-vitro cytotoxicity assays, we wanted to know whether this bystander activation also translates into improved target cell killing. For this purpose, CFSE-labeled MC38 cells either expressing MATEs or αCD3_{TM} were mixed with parental MC38 cells at a ratio of 1:50 as bystanders, and coincubated with splenocytes. We investigated target cell killing and discriminated MATE/αCD3_{TM}-expressing cells or bystander cells according to their CFSE-status. Our results show that killing of MATE/αCD3_{TM}-expressing cells had a higher trend in the group with the transmembrane T-cell engager compared with MATEs (Fig. 1G). More importantly, death of bystander cells was higher in the MATE-groups compared with the transmembrane variant. When calculating the ratio between the two fractions of killed cells (bystander vs. MATE/αCD3_{TM}-expressing cells, Fig. 1H), both MATEs showed significantly more bystander efficacy compared with the transmembrane-anchored T-cell engager confirming even in this non-dynamic system that MATEs can trigger T-cell mediated cytotoxicity against bystander cells. These findings also support our hypothesis that MATEs, once expressed by a tumor cell after infection with MATE-expressing oncolytic virus may effectively trigger T-cell activation in tumors.

## Humanized MATEs activate T cells in primary human PBMCs

In the previous experiments, we were able to show that murine MATEs effectively activate T cells and amplify antitumoral efficacy of oncolytic virotherapy in a murine tumor system. In order to realize an equivalent therapeutic option for application in humans, we generated humanized MATEs (αCD3h_{TAT}, αCD3h_{TAT}-Trimer, Fig. 2A) harboring the human CD3ε binding single chain variable fragment known from the αCD19 BiTE Blinatumomab (Löffler et al, 2000; Dreier et al, 2002). T-cell binding and activation properties of the humanized MATEs were evaluated using primary PBMCs from healthy human donors. The results show that humanized MATEs efficiently bind to different human CD8⁺ T-cell (Fig. 2B) and CD4⁺ T-cell subpopulations (Fig. 2C). For investigation of T-cell activation properties, PBMCs from 19

healthy donors were incubated with or without humanized MATEs for 24 h prior to flow cytometric assessment. Application of αCD3h_{TAT} as well as αCD3h_{TAT}-Trimer led to a significantly increased expression of CD69 on CD4⁺ and CD8⁺ T cells compared to untreated PBMCs (Fig. 2D). Furthermore, humanized MATEs increased expression of the activation marker CD137 that was most obvious in the CD8⁺ T-cell fraction (Fig. 2E). Furthermore, an increase in IFNγ production could be observed which was significant in CD8 T cells when treated with the monomer (Fig. 2F). Moreover, stimulation with MATEs led to production of TNFα in CD8⁺ and CD4⁺ T cells (Fig. 2G). Together, the generated humanized versions of MATEs showed in vitro similar binding properties compared to murine MATEs and were capable of activating primary human T cells demonstrating that humanized MATEs could be a promising tool for the generation of armed oncolytic viruses for future clinical development.

## MATE-expressing oncolytic adenoviruses activate T cells in vivo without risk of adverse events

Our results show that MATEs are functionally potent T-cell activators. However, as a non-specific targeting approach, MATEs are not suitable for systemic application but rely on a vector-based expression in the tumor, e.g. by intratumoral injection of a MATE-expressing oncolytic virus. We hypothesized a mutual benefit of oncolytic virotherapy and simultaneous MATE-expression because it is known that oncolytic virus infection promotes intense tumor infiltration with immune cells including T cells. Activation of these cells by MATEs could consequently confer cytotoxicity against infected and non-infected tumor cells as shown in the concept in Fig. 3A. As a carrier virus, we used a chimeric oncolytic adenovirus referred to as Ad5/11 to express MATEs during viral replication. The genetic setup of MATE-expressing Ad5/11 is illustrated in Fig. EV2A. In this virus, replication in normal cells is reduced by a p53-responsive expression of a microRNA-network against essential adenoviral genes as previously described (Gurlevik et al, 2009). Considering the high T-cell activating potency of MATEs and potential systemic hazards, a low expression level of MATEs was preferred and realized by using an IRES motif linked to the adenoviral E1B gene. Ad5/11 is predominantly a serotype 5 adenovirus that contains a small insertion of serotype 11 in the

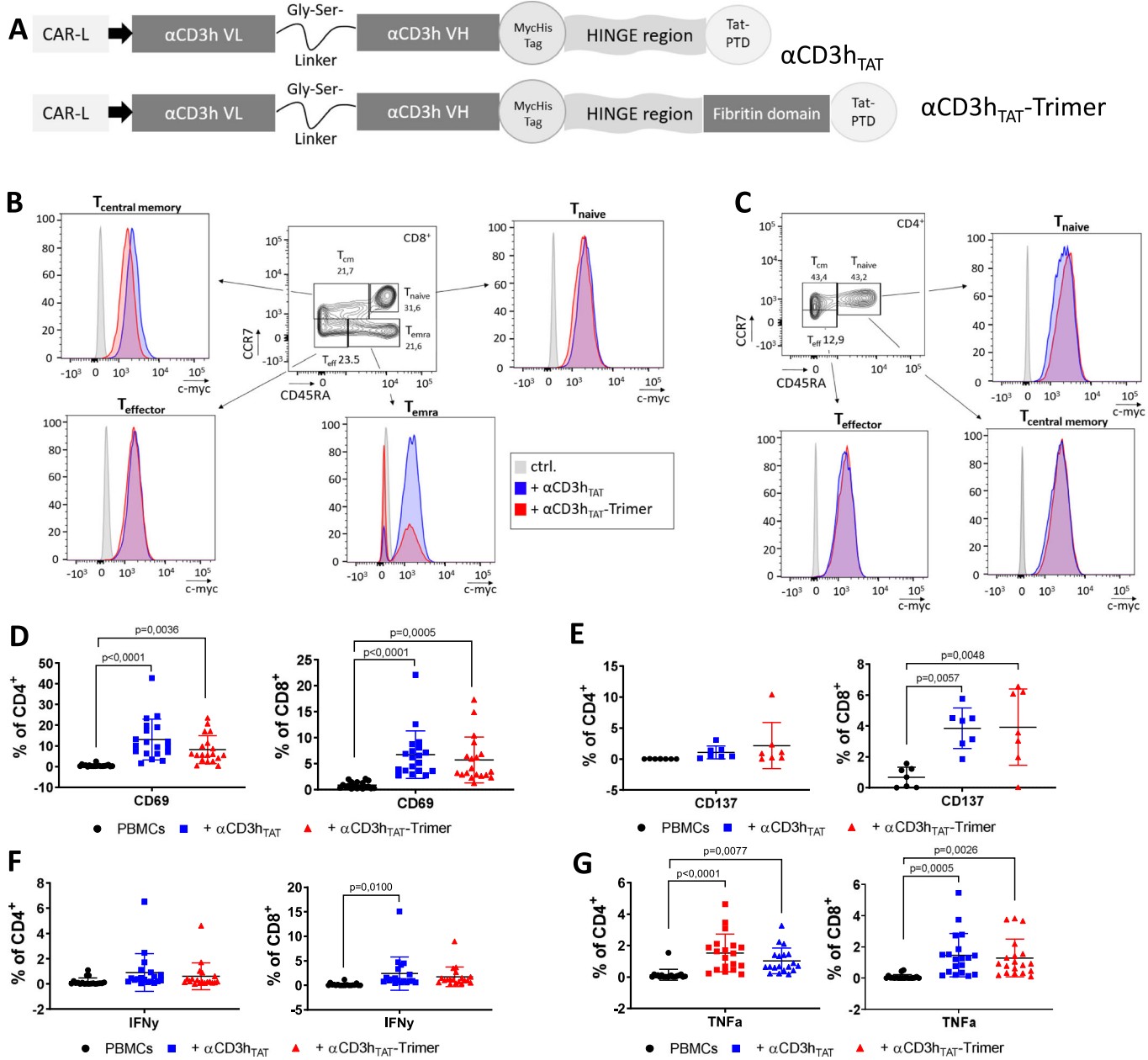

**Figure 2. Humanized versions of MATEs bind to and activate human CD4⁺ and CD8⁺ T cells.**

(A) Schematic illustration of humanized MATEs. (B, C) Human PBMCs from 19 healthy donors were isolated by pancoll gradient centrifugation. PBMCs were incubated with 1 µg of purified MATEs and binding to human CD8⁺ (B) and CD4⁺ T-cell subsets (C) was detected by flow cytometry. (D, E) After staining of CD69 ($n = 19$ human donors; mean ± SD) and CD137 ($n = 7$ human donors; mean ± SD), activation of CD4⁺ (left panels) and CD8⁺ (right panels) T cells was analyzed. (F, G) Functionality of MATE-activated human T cells was assessed by intracellular staining of IFNγ and TNFα ($n = 19$ human donors; mean ± SD). Statistical significance was assessed using a one-way ANOVA test with Tukey's post hoc analysis. Source data are available online for this figure.

E4orf3/orf4 region resulting in expression of a chimeric E4orf4 protein and a functional deletion of E4orf3. We considered these modifications to provide the virus with modestly, but not essentially reduced replication properties. Ad5/11 is able to replicate in and lyse different human tumor cells such as H1299 and Hep3B (Fig. EV2B,C) whereas no effective propagation in normal hepatocytes is detectable (Fig. EV2D). Replication in tumor cells is significantly lower compared with Ad5 wild type or the

already established serotype 5 based oncolytic adenovirus hTertAd (Wirth et al, 2003). Safe application of our strategy is therefore realized by a reduced replication potency in general, a diminished replication when infecting normal cells, and by a relatively low expression level of MATEs.

First, we investigated MATE-expression by Ad5/11 variants in the supernatants of infected HEK293 cells. The results show that the generated viruses actually secrete MATE proteins (Fig. 3B).

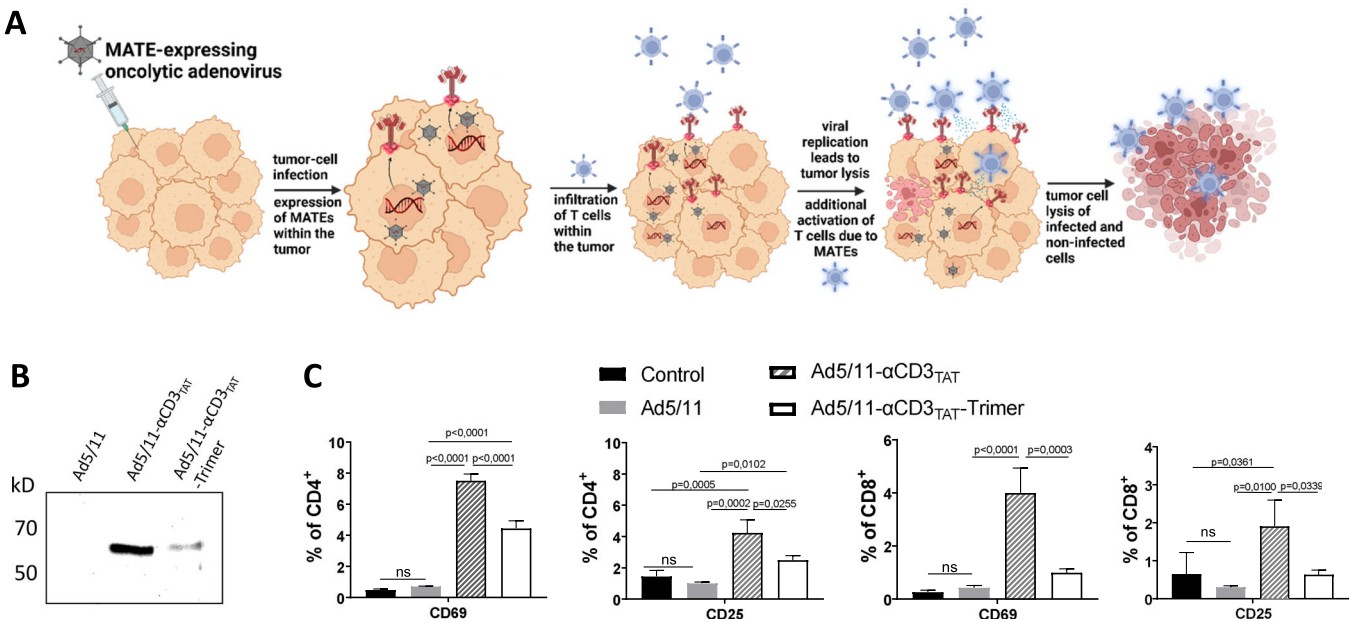

**Figure 3. MATE-expressing variants of the oncolytic adenovirus Ad5/11 activate CD4$^+$ and CD8$^+$ T cells in vitro and in vivo.**

(A) Schematic illustration showing the concept of intratumoral MATE-expression and secretion after infection of tumor cells with MATE-expressing oncolytic virus variants of Ad5/11. (B) MATE expression of infected HEK 293 cells 48 h after infection with control Ad5/11, or Ad5/11 variants expressing MATEs. Secreted MATEs were enriched by a Ni-NTA affinity resin and analyzed by western blot using an α-myc antibody. (C) HEK293 cells were infected with MATE-expressing Ad5/11 at an MOI of 25. After 48 h, supernatants of infected cells were subjected to murine splenocytes and activation of CD4$^+$ and CD8$^+$ T cells was investigated after 24 h of stimulation (number of replicates per group $n = 3$; mean, error bars represent SD; $p$ value by one-way ANOVA with Tukey's post hoc analysis). Source data are available online for this figure.

Detected levels of MATEs were indeed low, because detection required a Ni-NTA affinity purification from the supernatant prior to western blot analysis. At least in this infection assay, the detected levels of the monomeric MATE were higher compared with the trimeric MATE. This may reflect a higher resorption of the trimer due to its putatively higher affinity. Coincubation of murine splenocytes with the supernatants from infected HEK293 cells confirmed T-cell activation (Fig. 3C). In these assays, T-cell activation with the monomer was stronger, reflecting the higher levels and availability observed in the supernatants. Next, we wanted to examine the ability of MATE-expressing viruses to activate T cells in a syngeneic murine MC38 tumor model in vivo. It is known that human adenovirus propagation is species-specific and that viral particle production is low in murine cells due to a hampered translation of late structural proteins (Young et al, 2012). Though the model does not reflect a full-blown oncolysis in the fully permissive situation in humans, the model is well-established for immunotherapeutic analyses and was considered suitable to characterize MATE-dependent T-cell immune activation in vivo. In vitro, MC38 cells are fairly susceptible for human adenovirus entry (Fig. EV3A). Exposure of MC38 cells to a replication-incompetent Ad-EGFP at MOI of 50 led to moderate transduction whereas almost full transduction was achieved at MOI of 250. Replication of Ad5/11 in MC38 cells is detectable to some degree but on low levels when compared with the highly replication-permissive human H1299 cells (Fig. EV3B). We infected s.c. MC38 tumors in immunocompetent C57BL/6 mice with MATE-expressing Ad5/11 viruses or an Ad5/11 control virus and investigated activation

markers on T cells which had been prepared from tumor tissue 3 days following infection (Fig. 4A,B). In tumors treated with Ad5/11-αCD3$_{TAT}$, CD8$^+$ T cells showed a significantly enhanced expression of CD69 and CD25. Activation by the trimeric MATE (Ad5/11-αCD3$_{TAT}$-Trimer) was also elevated when compared with empty virus, but the difference was not significant due to high variability among the individuals. It has to be taken into account that differences in secretion of MATEs has already been observed in HEK293 cells (Fig. 3B,C) may affect these early measurements. Because of the target-independent cell surface association mediated by the Tat-PTD domain, a potential systemic release of MATEs from tumor tissue could potentially confer a significant risk of autoimmunity and adverse events. Therefore, we were interested if MATEs expressed during tumor virotherapy remain within the tumor tissue or might become a systemic threat by release into the lymphatic system. To this end, we analyzed T-cell activation in tumor-draining lymph nodes (TDLN). Expression of MATEs by these Ad5/11 variants during intratumoral virotherapy did not cause an increase of T-cell activation in TDLNs at this early time point (Fig. 4B). Also, we did not detect any external signs of adverse events in this cohort of mice. To further evaluate potential hazards by released MATEs we studied toxicity aspects more in detail. Hepatotoxicity may occur either as a consequence of the well-known hepatotropism of adenoviruses or to MATE-dependent T-cell autoimmunity. For detection of these early events, a new cohort of mice has been treated with MATE-expressing virotherapy and mice were analyzed at day two after treatment. Consistent with the healthy liver histology revealed by microscopic inspection of H/E-

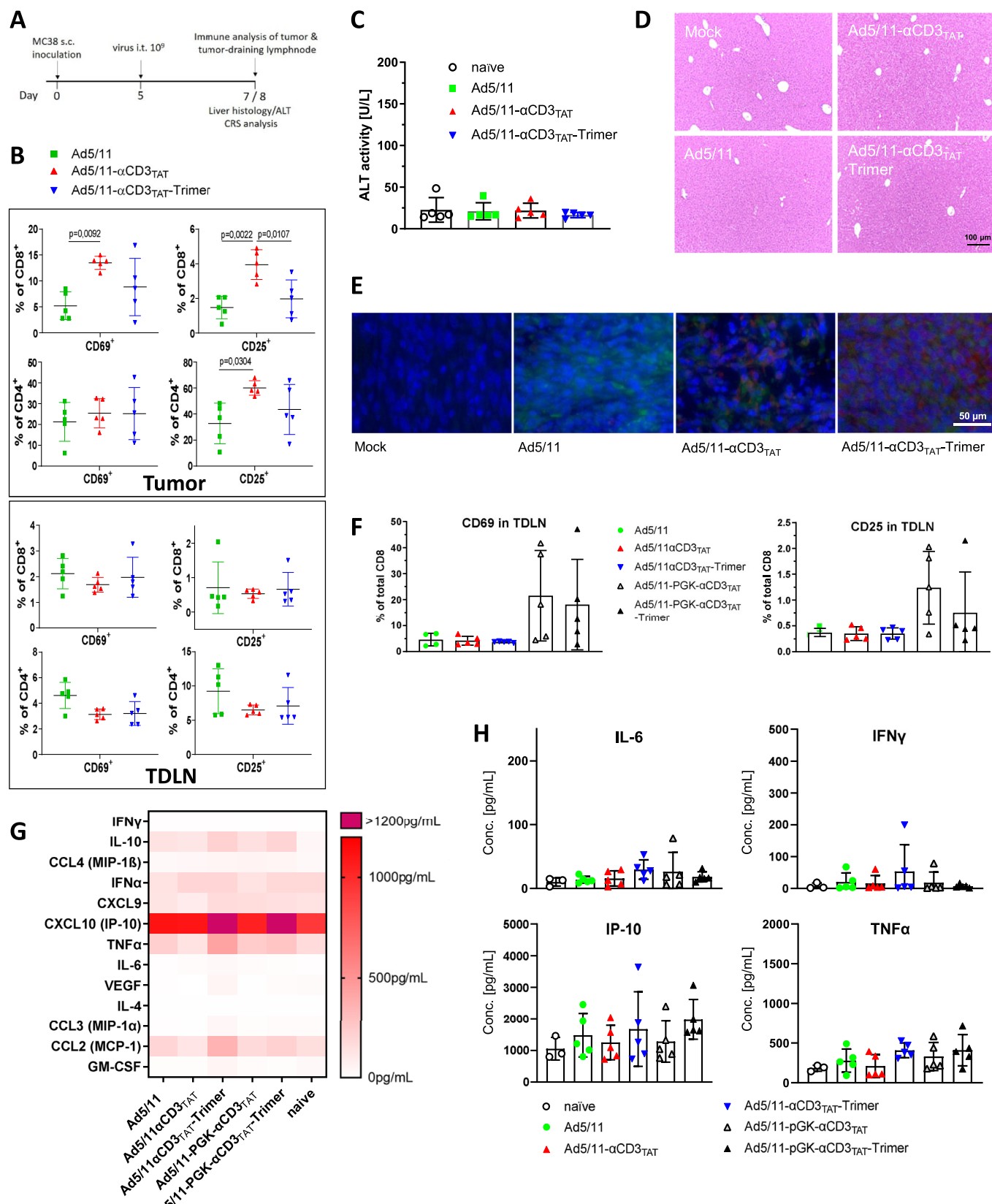

**Figure 4. Intratumoral virotherapy with MATE-expressing Ad5/11 activates intratumoral immune cells without eliciting systemic toxicity.**

(A) S.c. MC38 tumors in immunocompetent C57BL/6 Mice were treated according to the scheme. Five days after tumor inoculation, mice were treated with a single dose of $1 \times 10^9$ ifu of MATE-expressing variants of Ad5/11 or the control virus Ad5/11. (B) After 3 days, lymphocytes from tumor tissue and the tumor draining lymph nodes (TDLN) were prepared and T-cell activation was analyzed by flow cytometry ($n = 5$ mice per group; mean ± SD). In a further experimental cohort mice were sacrificed 48 h after infection with MATE-expressing Ad5/11 or the Ad5/11 control virus. To investigate systemic toxicity and adverse events, tumor-tissue, tumor-draining lymph nodes, contralateral lymph nodes, serum and liver tissue was obtained. Measured ALT-levels are shown in (C) ($n = 5$ mice per group; mean ± SD) and liver tissue integrity was investigated by H/E staining of tissue slices using FFPE material (D) (magnification 100×; scale bar indicates 100 μm; representative images from each group are shown). Obtained tumor tissue slices were investigated by immunohistochemistry using antibodies against E1A (green) and c-myc (red) to detect virus infection and MATEs, respectively, DAPI was used to stain nuclei (magnification 400×; scale bar indicates 50 μm; representative composite images from each group are shown) (E). (F) Including additional experimental groups with enhanced MATE-expression under transcriptional control of a PGK-promoter, T-cell-activation was analyzed in TDLNs using the markers CD69 (left panel) and CD25 (right panel) ($n = 4$ mice for Ad5/11 group, $n = 5$ for all other groups; mean ± SD). (G) Using a Legendplex mouse cytokine release syndrome panel (Biolegend), cytokine levels which are typically elevated during cytokine release syndrome were investigated in blood samples of mice 48 h after treatment. Experimental groups with MATE-expressing Ad5/11, control Ad5/11 and naïve mice were compared. All results are displayed as overview in (G) ($n = 3$ for naïve mice, $n = 5$ for all other groups; samples were measured in duplicates; results are depicted as median). Selected single cytokine measurements are depicted in (H) ($n = 3$ for naïve mice, $n = 5$ for all other groups; samples were measured as technical duplicates; mean ± SD). Source data are available online for this figure.

stained liver tissue slices, the liver enzyme ALT was not elevated in the serum of virotherapy-treated mice compared with naïve mice demonstrating that liver integrity was neither affected by systemic virus nor by MATE-dependent effects (Fig. 4C,D). In tumor tissue sections from these mice, adenoviral E1A and MATEs were detectable by immunohistochemistry (Fig. 4E) indicating successful infection of tumor cells in the treated tumors. In this experimental setting, our findings suggest that MATEs were effectively retained by the infected tumor tissue or had been sufficiently diluted to prevent aberrant T-cell activation outside the tumor tissue. However, due to the low permissivity of the model for adenovirus replication, levels of oncolysis and MATE release can presumably be higher in a fully permissive human tumor. To indirectly address this limitation and to investigate the implications of increased MATE levels on safe application of the approach, we have generated additional Ad5/11 variants with enhanced MATE expression. The generated variants of Ad5/11, referred to as Ad5/11-PGK-αCD3$_{TAT}$ and Ad5/11-PGK-αCD3$_{TAT}$-Trimer, respectively, were equipped with an expression cassette under transcriptional control of a strong phosphoglycerate kinase (PGK) promoter as illustrated in Fig. EV3C. By western blot analyses of MATEs in cell extracts and in Ni-affinity-enriched samples obtained from supernatants of infected MC38 cells (Fig. EV3D), we could observe that the novel virus variants secreted significantly higher levels of MATEs into the supernatant compared with the low-level expression variants which were hardly detectable in this setting. We also investigated T-cell activation in TDLNs as a surrogate for MATE release from infected tumors. Interestingly, infection with the novel Ad5/11-PGK-MATE variants led indeed to high T-cell activation in some of the treated animals (Fig. 4F). This observation suggests that excessively produced MATEs may indeed be released by the infected tumor into the lymphatic system. This effect was local because contralateral lymph nodes did not show any T-cell activation (Fig. EV3E). However, this putative release of MATEs beyond the tumor did not lead to liver toxicity as determined by ALT levels in serum and by liver histology (Fig. EV3F,G). To assess potential hazards caused by released MATEs with a more sensitive detection method, we evaluated a panel of cytokines in the serum of treated mice which are characteristically elevated during a cytokine release syndrome (CRS), a dreaded adverse event known from T-cell-based immunotherapies. IL-6 is considered a lead cytokine and is also the main therapeutic target to reduce CRS symptoms (Ferreri and Bhutani, 2024). The results of our cytokine array are

shown as overview in Fig. 4G, selected single cytokines including IL6, IFNγ, IP-10, and TNFα are displayed in Fig. 4H whereas all remaining measurements of single cytokines are shown in Fig. EV3H. In general, IL-6 as well as the other cytokines remained at low levels in all treatment groups even when compared with naïve animals. This is consistent with our observation that no animals showed any external signs of adverse events or liver toxicity. Only IL-10 was significantly and similarly elevated in all treatment groups which is probably tumor-mediated since a tumor-free mouse was used as naïve control and IL-10 plays a role in immunosuppression in the tumor microenvironment. Several individual data points suggested moderately elevated levels of certain cytokines, but those were attributable to different individuals or did not correlate with highest T cell activation in TDLN and therefore rather reflect differences in the injection procedure. Most importantly, there was no significant difference detectable when comparing the Ad5/11-PGK-MATE variants and their respective low-level MATE-expressing Ad5/11 variants. In summary, our results show that our investigated viruses are able to express functional MATEs, resulting in intratumoral T-cell activation. Even when using virus variants with enhanced MATE expression, adverse events could not be observed indicating that intratumoral application of MATE-expressing oncolytic adenoviruses is safe. Nevertheless, our data also suggest that high levels of secreted MATEs, which probably occur in a human virotherapeutic setting, can indeed reach the lymphatic system. It was therefore reasonable to prefer our initially generated Ad5/11 variants with low-level expression of MATEs for subsequent preclinical examinations.

## MATE-encoding oncolytic viruses inhibit tumor growth and lead to improved overall survival

We next assessed the therapeutic potential of MATE-expression during virotherapy after treatment of s.c. MC38 tumors with MATE-expressing oncolytic viruses according to the treatment scheme in Fig. 5A. Tumor growth and survival were monitored. The results show that treatment with MATE-expressing viruses delayed tumor growth more effectively compared with empty virus and significantly improved survival. In both groups treated with MATE-expressing oncolytic viruses even complete tumor remission could be observed (Fig. 5A). To better understand the underlying immune mechanisms we examined the T-cell composition in treated MC38 tumors at day

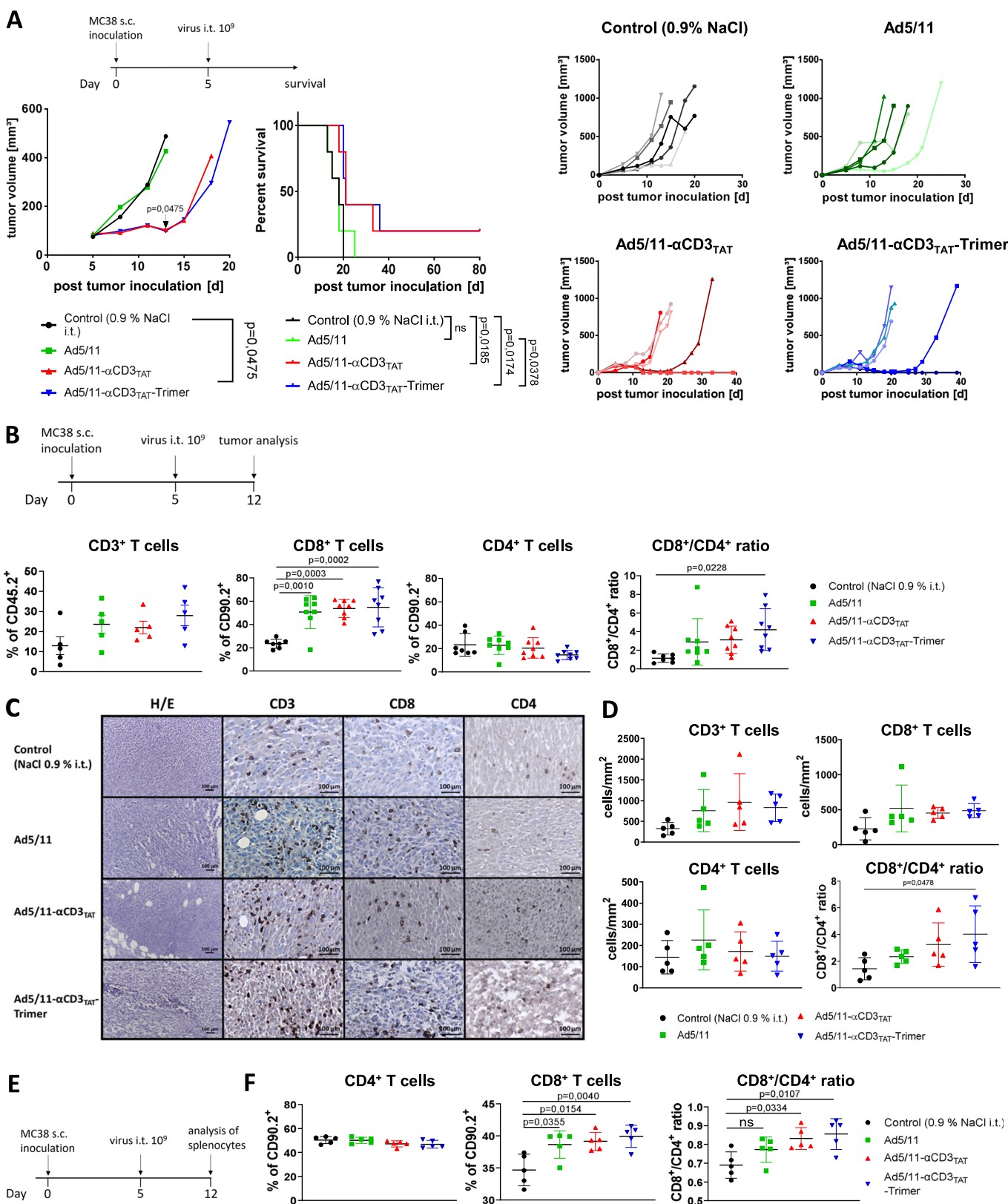

**Figure 5. Intratumoral virotherapy with MATE-expressing Ad5/11 improves overall survival of subcutaneous tumor-bearing mice and supports intratumoral T-cell infiltration.**

(A) S.c. MC38 tumors were established in C57BL/6 mice and were treated with a single intratumoral injection of MATE-expressing Ad5/11 or saline as outlined. Mean tumor volume was assessed (left) and survival was monitored (middle). The individual tumor volume is shown in the right hand panels. Median survival (ms): Saline: 18 d; Ad5/11 control: 18 d; Ad5/11-αCD3$_{TAT}$: 21 d; Ad5/11-αCD3$_{TAT}$-Trimer: 21 d. Log-rank (Mantel–Cox) test was used to calculate survival statistics. Significance in tumor-growth was assessed using a two-tailed unpaired $t$-test ($n = 5$ mice for all groups). (B) S.c. MC38 tumors were treated as illustrated. After 7 days, mice were sacrificed, intratumoral leukocytes were prepared and the T-cell subsets were analyzed via flow cytometry. CD3$^+$ T cells were calculated as a percentage of CD45.2$^+$ cells ($n = 5$ mice; mean ± SD; $p$ values by one-way ANOVA with Tukey's post hoc analysis). Amount of CD4$^+$ and CD8$^+$ T cells was calculated as percentage of CD90.2$^+$ lymphocytes ($n = 7$ mice for control group, $n = 8$ mice for all other groups; mean ± SD; $p$ values by one-way ANOVA with Tukey's post hoc analysis). (C) Distribution of T cells in treated tumors was examined by immunohistochemistry. Representative images from each group are depicted (left: magnification 200×; bars indicate 100 µm). (D) Cell numbers were determined by counting per area using ImageJ ($n = 5$ mice; mean ± SD; $p$ values by one-way ANOVA with Tukey's post hoc analysis). (E) Splenocytes of treated mice were isolated and analyzed via flow cytometry according to the scheme. (F) The amount of CD4$^+$ and CD8$^+$ T cells within the lymphocyte population as well as the CD8/CD4 T cell ratio was calculated ($n = 5$ mice; mean ± SD; $p$ values by one-way ANOVA with Tukey's post hoc analysis). Source data are available online for this figure.

seven after virotherapy according to the scheme in Fig. 5B. Virotherapy promoted total lymphocytic infiltration in the tumor, as determined by the frequency of CD3$^+$ cells within the CD45$^+$ leukocyte compartment and led to a significant increase of CD8$^+$ T cells (Fig. 5B, exemplary plots in Fig. EV4A). Within the CD8$^+$ T-cell compartment, the proportion of the effector CD8$^+$ T-cell subtype was significantly elevated at the cost of naïve and memory CD8$^+$ T-cell subpopulations which was further increased by expression of MATEs (Fig. EV4B,C). The amount of CD4$^+$ T cells appeared to decrease in response to MATE-expression (Fig. 5B). After calculating the individualized CD8$^+$ to CD4$^+$ T-cell ratio as a suitable surrogate of tumor immune activation, the results show that this ratio was significantly elevated in tumors treated with Ad5/11-αCD3$_{TAT}$-Trimer. To confirm these findings, we performed immunohistochemical stainings of s.c. MC38 tumors isolated 7 days after application of virotherapy (Fig. 5C). In line with the flow cytometric results, Qupath quantification showed elevated numbers of intratumoral CD3$^+$ T cells consistent with a virus-mediated influx of T lymphocytes in treated tumors and an improved CD8/CD4 T cell ratio that was most significantly altered in Ad5/11-αCD3$_{TAT}$-Trimer treated mice (Fig. 5D). Furthermore, we analyzed the amount of dendritic cells, macrophages, and NK cells (Fig. EV4D). After treatment with MATE-expressing viruses, we found a moderate increase in DCs, as well as a moderate decrease of macrophages. Though these observations did not reach statistical significance they are consistent with a higher immunoactivation illustrated by the CD8/CD4 T-cell ratio. As could be expected, NK cells were slightly elevated by virotherapy, but were not further affected by MATE expression. Next, we used splenocytes to investigate the impact of virotherapy +/− expression of MATEs on the systemic composition of T cells and different CD8$^+$ T-cell subsets (Figs. 5E,F and EV4E,F). Consistent with our observations described above, the results show that virotherapy led to a significant increase in the frequency of systemic CD8$^+$ T cells (Fig. 5F). Mirroring the shift in T-cell subsets that had been observed in tumor tissue, virotherapy and particularly MATE-virotherapy increased the amount of CD8$^+$ T cells resulting in a more favorable ratio of CD8$^+$ to CD4$^+$ T cells. The gain of effector CD8$^+$ T cells at the cost of naïve and central memory subtypes was most prominent in mice after treatment with the Ad5/11-αCD3$_{TAT}$-Trimer (Fig. EV4F) reflecting the improved immune activation.

Despite a potentially higher availability of the monomer, our data suggest a functional advantage of the trimer. Together, our findings described above demonstrate that MATEs are promising tools to achieve additional T-cell stimulation when expressed by oncolytic viruses.

## Therapeutic efficacy of MATEs depends on CD8$^+$ T cells and triggers PD-1 expression

The aim of MATEs when expressed by an oncolytic virus is a direct activation of tumor resident and infiltrating T cells, predominantly of CD8$^+$ T cells which are known as potent antitumoral effectors. Consistent with this aim, our data shown above suggest a role of CD8$^+$ T cells. To confirm this, we performed a CD8$^+$ T-cell depletion performed by application of a CD8$^+$ depleting antibody according to the treatment scheme in Fig. 6A. Histological staining of CD8$^+$ T cells within the tumor tissue after CD8$^+$ depletion confirmed the almost complete elimination of CD8$^+$ T cells (Fig. EV4G). Depletion of CD8$^+$ T cells abolished the additional therapeutic effect of MATEs compared to naked virus (Fig. 6A) demonstrating that the antitumor effect of MATEs depends on CD8$^+$ T cells. Tumor cell death by virotherapy and MATE activity should further trigger tumor-specific T-cell responses according to the mechanisms of the cancer immunity cycle (Chen and Mellman, 2017). The induction of tumor antigen-specific T cells is essential for a long-term antitumor immune response. To investigate triggering of antigen-specific CD8$^+$ T-cell dependent immunity we used the established neoepitope Adpgk-R304M of MC38 cells (Yadav et al, 2014). Immune cells isolated from tumors 7 days after virotherapy were stimulated with an Adpgk-R304M peptide to analyze tumor-specific CD8$^+$ T-cell responses by intracellular IFNγ staining. Though variance between single values was considerable, the levels of Adpgk-R304M-specific CD8$^+$ T cells were higher in MATE-expressing virus groups compared with virotherapy-treated and untreated tumors (Fig. 6B).

Apart from the impact of MATEs on different intratumoral immune cell subsets, we wanted to assess the intratumoral levels of chemokines and cytokines which provide further information on the intratumoral immune status. In a luminex multiplex analysis (Fig. 6C) performed at day seven following treatment we observed that the amount of the T-cell proliferation promoting interleukin-2 (IL-2) was generally increased in tumors treated with MATE-expressing viruses, which was statistically significant in tumors treated with Ad5/11-αCD3$_{TAT}$ compared with naked Ad5/11. Furthermore, this group showed an increase of intratumoral IFNγ

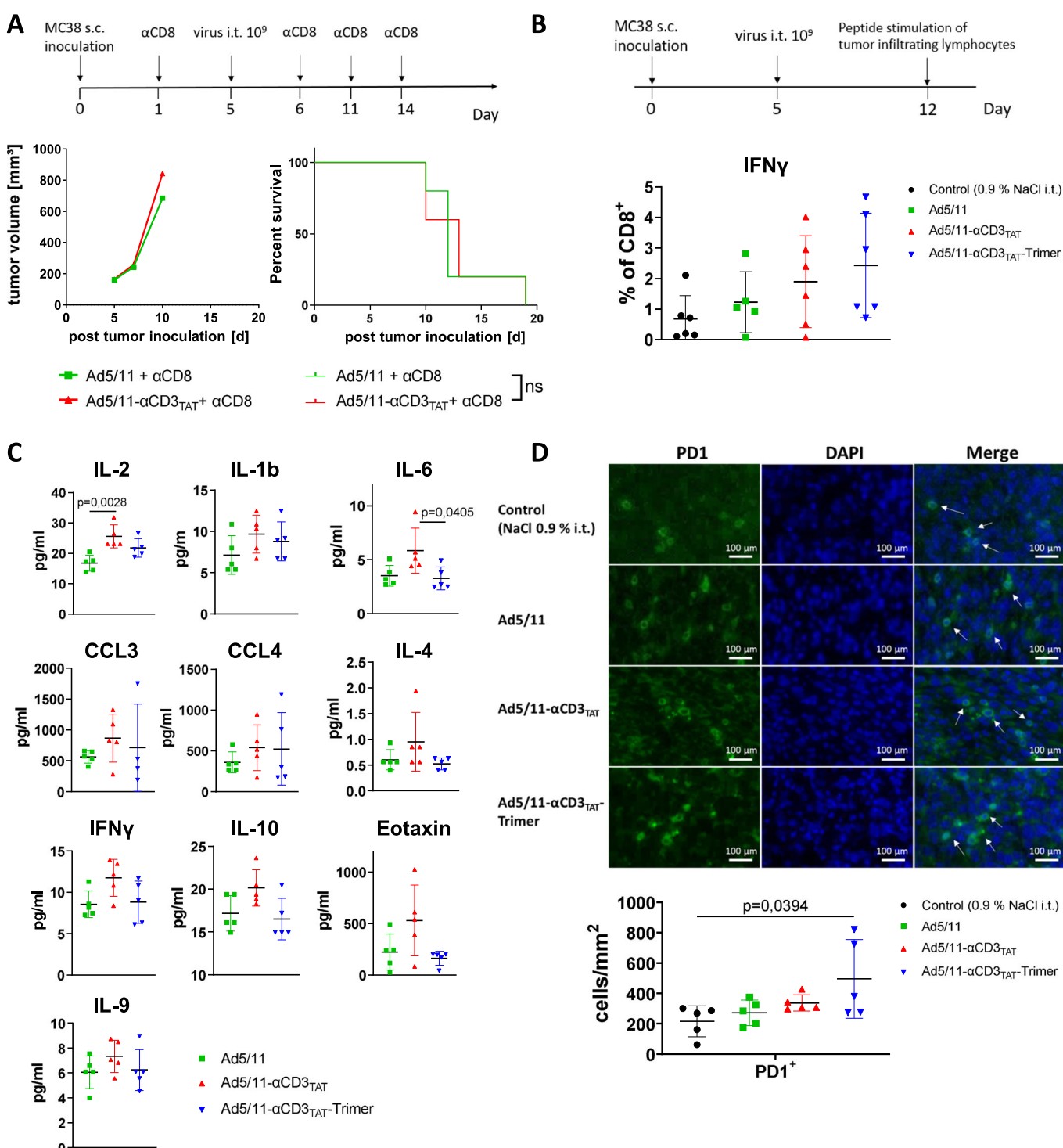

and significantly higher amounts of IL-6. Consistent with an inflammatory scenario, IL-9 and IL-1b were moderately increased in both MATE-expressing groups as well as the immune cell attracting chemokines CCL3 and CCL4, though with considerable variance among individuals. In line with a still ongoing inflammation induced by MATEs, eotaxin was elevated, an important attractor chemokine for granulocytes. Interestingly, treatment with Ad5/11-αCD3$_{TAT}$ therapy also led to an increase of the immunosuppressive cytokines IL-4 and IL-10 which was absent after treatment with the MATE-Trimer or was even lower compared with naked virus. In summary, the luminex analysis data suggest a still ongoing tumor inflammation in the Ad5/11-αCD3$_{TAT}$ group that was rather moderate in the trimer group. This observation might be a consequence of the lower levels of the trimer released by the corresponding virus. However, still ongoing inflammatory processes observed in the monomer group may also

Figure 6.   Therapeutic benefit of MATE-expressing Ad5/11 is mediated by CD8⁺ T cells.

(A) S.c. MC38 tumors were established in C57BL/6 mice and treated according to the illustration. In addition to virotherapy, a CD8⁺ T-cell depletion was performed starting the day after tumor inoculation. Mean tumor growth (left panel) and survival (right panel) were monitored. Log-rank (Mantel–Cox) test was used to calculate survival statistics ($n = 5$ mice; mean ± SD). (B) MC38 tumor-bearing mice were treated according to the scheme. Tumor infiltrating leukocytes were isolated 7 days following treatment and incubated with the peptide ASMTNMELM to investigate T-cell responses against the MC38-characteristic neoantigen Adpgk-R304M. Neoantigen-specific CD8⁺ T cells were identified by flow cytometry of cells subjected to peptide stimulation and ICS for IFNγ ($n = 5$ mice for Ad5/11 group, $n = 6$ mice for all other groups; mean ± SD). (C) Analysis of the cytokine and chemokine composition within the tumor tissue was performed using luminex multiplex assay ($n = 5$ mice; mean ± SD; $p$ values by one-way ANOVA with Tukey's post hoc analysis). (D) Seven days after virotherapy, intratumoral PD-1⁺ cells (green) were investigated by immunohistochemistry, and nuclei were stained with Hoechst 33258 (blue). Exemplary PD-1 positive cells are marked by white arrows. Representative composite images from each group are shown (magnification 200×; bars indicate 100 μm). Quantification as shown in the bottom panel was performed using ImageJ. ($n = 5$ mice; mean ± SD; $p$ values by one-way ANOVA with Tukey's post hoc analysis). Source data are available online for this figure.

promote the induction of anti-inflammatory, potentially immuno-suppressive mechanisms which could not be observed in the group treated with the MATE trimer.

PD-1 is frequently upregulated on T cells as a result of repeated antigen contact and regarded as an exhaustion marker. Consequent to their activity, intratumoral, antigen-specific T cells are frequently high in PD-1 and can be effectively silenced by PD-L1 expressed by tumor cells and immune cells in the tumor stroma. We stained PD1⁺ cells in tumor tissue 7 days after treatment (Fig. 6D). Immunohistochemical quantification of PD1⁺ cells in tumor tissue from mice treated with virotherapy or MATE-expressing viruses showed significantly elevated levels of PD1⁺ cells within tumors treated with Ad5/11-αCD3$_{TAT}$-Trimer compared to the saline control suggesting additional activity of tumor-specific T cells. Together, our results confirm the essential role of CD8⁺ T cells in MATE-virotherapy. The induction of PD-1 in response to T-cell activation further suggest a possible synergy of MATE virotherapy with PD1/PD-L1 directed checkpoint inhibition.

## MATE-expressing virotherapy sensitizes for αPD1 checkpoint blockade, enhances antitumor immunity and induces tumor-specific CD8⁺ T cells

Long-term survival after immunotherapy targeting the PD-1/PD-L1 checkpoint is limited to a small subgroup of cancer patients. T-cell infiltration within the tumor as well as PD1/PD-L1 expression are critical surrogates that patients may respond to therapy (Topalian et al, 2012). Our previous experiments indicate that MATE-expressing virotherapy led to an enhanced infiltration of CD8⁺ T cells within the tumor and promotes PD1 expression suggesting that MATE-expressing virotherapy could be a promising intervention prior to application of checkpoint inhibitors. In the majority of our previous experiments, the MATE trimer seemed to show a more favorable profile regarding the quality of T-cell activation such as the CD8/CD4 T cell ratio, triggering of tumor-specific T cells and a higher number of PD1-positive cells within the tumor. Therefore, we chose Ad5/11-αCD3$_{TAT}$-Trimer virotherapy to investigate the therapeutic efficacy of a combination of virotherapy and PD-1 checkpoint blockade in the MC38 tumor model. One day after intratumoral virotherapy we induced a PD-1 checkpoint blockade for more than two weeks and monitored tumor growth and survival (Fig. 7A). Whereas responsiveness of tumors to αPD-1 also if combined with naked Ad5/11 was rather limited, the combination of MATE-expressing virotherapy and checkpoint inhibition led to a significantly reduced tumor growth and significantly prolonged survival compared with checkpoint

inhibition alone. Moreover, three out of five mice showed a complete tumor remission clearly demonstrating the therapeutic benefit achieved by MATE expression. As expected, the combination of virotherapy and PD-1 blockade promoted tumor infiltration of CD3⁺ T lymphocytes including CD8⁺ T cells (Fig. 7B and exemplary plots shown in Fig. EV5B). Analysis of different CD8⁺ T-cell subsets showed a significant increase of effector T cells and reduced the frequency of central memory and naïve T cells when PD-1 therapy was combined with virotherapy (Fig. EV5C). However, tumors treated with Ad5/11-αCD3$_{TAT}$-Trimer showed a significantly reduced frequency of CD4⁺ T cells within the intratumoral lymphocyte population and a more favorable CD8⁺ to CD4⁺ ratio compared to PD1 monotherapy (Fig. 7B). In addition, we observed a decrease of macrophages within tumors treated with Ad5/11-αCD3$_{TAT}$-Trimer (Fig. 7C). However, frequencies of macrophages with M1 characteristics (MHCII$^{high}$) were stable suggesting that the decrease could primarily affect M2 macrophages (Fig. EV5D). Whereas the amount of DCs (CD11c⁺ MHC II$^{high}$ F4/80⁻) within the tumor decreased (Fig. 7C), we found that the frequency of the cross presenting XCR1⁺ subtype of DCs within the DC population was obviously higher in tumors treated with Ad5/11-αCD3$_{TAT}$-Trimer. Virotherapy and αPD-1 triggered NK cells whereby expression of MATEs did not make any significant difference (Fig EV5E). Taken together, these findings point to an effective immune activation in the tumor microenvironment by the MATE-expressing virus when compared to empty virus or PD1 monotherapy. Because the tumor burden in the trimer group was too small to obtain sufficient amounts of T cells for investigations on antitumoral epitope-specific CD8⁺ T cells, we analyzed tumor-directed Adpgk-R304M-specific CD8⁺ T cells in splenocytes isolated 7 days after virotherapy and two applications of αPD1 by intracellular IFNγ staining (Fig. 7D). A statistically significant number of systemic CD8⁺ T cells specific for the Adpgk neoepitope could only be identified in splenocytes from mice treated with Ad5/11-αCD3$_{TAT}$-Trimer but neither with PD-1 monotherapy nor when αPD1 was combined with the empty virus Ad5/11. Moreover, we analyzed in splenocytes systemic changes in the composition of T cells. The results show higher amounts of CD8⁺ T cells and a ratio of CD8⁺ to CD4⁺ in favor of CD8⁺ T cells (Figs. 7E and EV5F). Combined therapy of αPD-1 and Ad5/11-αCD3$_{TAT}$-Trimer showed the highest frequency of systemic effector CD8⁺ T cells (Fig. EV5F) consistent with the effective immunoactivation in tumor tissue. In summary, our results indicate that MATE-expressing oncolytic virotherapy effectively converts the tumor microenvironment into an immunoactivated state that sensitizes for PD1 immune checkpoint blockade. Thus, MATE-expressing

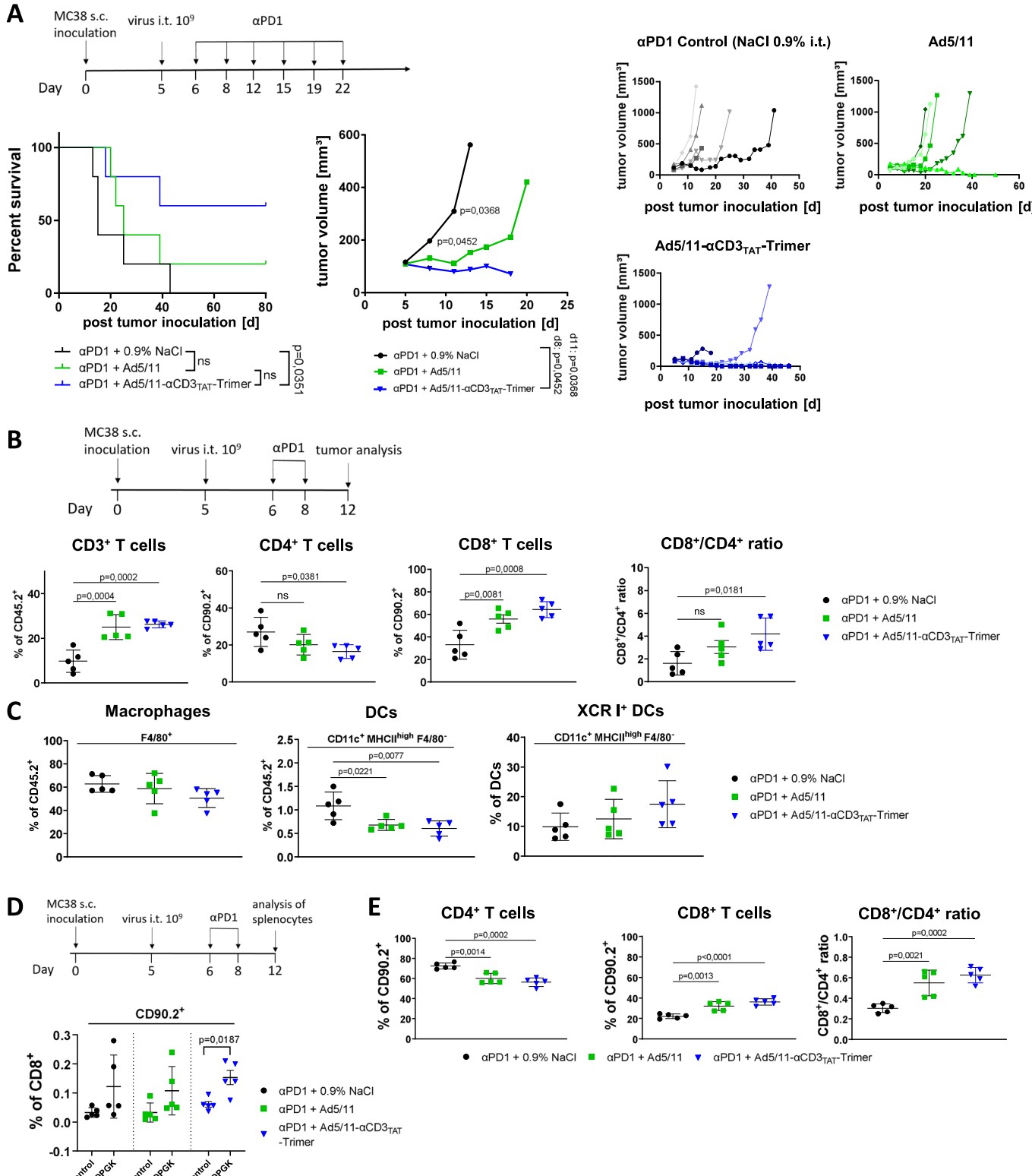

◀ **Figure 7. MATE-expressing virotherapy sensitizes tumors for αPD1 checkpoint blockade.**

(A) S.c. MC38 tumors were established and treated with MATE-expressing viruses as illustrated followed by an application of αPD1 antibodies twice a week over a total of 3 weeks. Survival and mean tumor growth were monitored (left and middle panels; $n = 5$ mice per group). Log-rank (Mantel–Cox) test was used for survival statistics. Significance between tumor volumes was assessed using two-tailed, unpaired $t$-test. Median survival (ms): Saline injection: 15 d; Ad5/11: 25 d; Ad5/11-αCD3$_{TAT}$-Trimer: undefined. Growth of individual tumors is shown in the panels on the right hand side. (B, C) S.c. MC38 tumors were established and treated as outlined by virotherapy followed by two doses of αPD1 antibodies per week. Seven days after virotherapy tumor tissue was examined for leukocyte infiltration via flow cytometry. The analysis of CD4$^+$ and CD8$^+$ T cells, and the CD8/CD4 T-cell ratio is shown in (B) ($n = 5$ mice; mean ± SD; $p$ values by one-way ANOVA with Tukey's post hoc analysis). The frequencies of macrophages (F4/80$^+$) and total DCs (CD11c$^+$, MHC II$^{high}$, F4/80$^-$) and the amount of crosspresenting XCR I$^+$ DC subtype of total DCs is shown in (C) ($n = 5$ mice; mean ± SD; $p$ values by one-way ANOVA with Tukey's post hoc analysis). (D) S.c. MC38 tumor-bearing mice were treated and analyzed as described in the scheme. Seven days after virotherapy, splenocytes were isolated. Splenocytes were examined for tumor-specific CD8$^+$ T cells by stimulation with the mutated MC38-specific peptide Adpgk-R304M (ASMTNMELM). Responding neoantigen-reactive CD8$^+$ T cells were identified by IFNγ-ICS and flow cytometry ($n = 5$ mice; mean ± SD; $p$ values by two-tailed unpaired $t$-test). (E) T-cell subpopulations were analyzed via flow cytometry and the CD8/CD4 ratio was calculated ($n = 5$ mice; mean ± SD; $p$ values by one-way ANOVA with Tukey's post hoc analysis). Source data are available online for this figure.

oncolytic viruses could be a promising tool to broaden the therapeutic response of PD1 immune checkpoint blockade to tumor species that are otherwise resistant to therapy.

## During αPD1 checkpoint blockade, MATE-expressing virotherapy leads to increased spatial proximity of T cells and tumor cells and promotes the formation of CD4/CD8 T-cell clusters

In the context of PD1 checkpoint inhibition, MATE expressing virotherapy leads to a more effective immunoactivation in tumors compared with the oncolytic carrier virus as suggested by various immune parameters including the altered T-cell composition, improved CD8/CD4 T-cell ratio as well as alterations in the macrophage and dendritic cell compartment. To investigate further mechanisms that might explain the significant therapeutic benefit obtained by MATEs we focused on spatial relations of immune cells in the tumor tissue, which cannot be addressed by conventional flow cytometric examinations. Therefore, we performed a multiplex immunohistochemical analysis on tumor samples of mice after treatment with αPD1 therapy + Ad5/11-αCD3$_{TAT}$-Trimer or the empty virus, respectively. Tumor tissue was immunohistochemically stained to identify CD8$^+$ T cells, CD4$^+$ T cells, CD161$^+$ NK cells, F4/80$^+$ macrophages and pan-Cadherin positive (PC$^+$) tumor cells. First, we determined total cell numbers per mm$^2$. Virotherapy and MATE-expressing virotherapy decreased tumor cell density suggesting a less compact structure of the tumor tissue consistent with immunoactivation (Fig. EV5G,H). In the investigated tumor areas, combination of PD-1 immune checkpoint blockade with Ad5/11-αCD3$_{TAT}$-Trimer showed a significantly higher amount of total T cells including both total CD8$^+$ and CD4$^+$ T cells (Fig. EV5H). According to our hypothesis that the redirection of cytotoxic T cells against tumor cells by MATEs is key to generating a functional immunological synapse and antitumor immune activation, we then analyzed the spatial relations of T cells with PC$^+$ tumor cells. Our results show significantly elevated numbers of CD8$^+$ T cells in near proximity of analyzed PC$^+$ tumor cell in tumors treated with the MATE-expressing virus Ad5/11-αCD3$_{TAT}$-Trimer when compared to the carrier virus (Fig. 8A). More importantly, the quantification of CD8$^+$ T cells in direct contact or in close proximity to PC$^+$ cells showed a significantly higher abundance of CD8$^+$ T cells in the proximity of PC$^+$ cells (Fig. 8B). In addition, we could detect more CD4$^+$ T cells in direct contact or close to PC$^+$ tumor cells (Fig. 8C). Median distance measurements

confirmed a closer contact of T cells with tumor cells in the trimer group which could allow a more intense and effective interaction of effector and target cells (Fig. 8D). Next, we investigated the spatial relations of the T-cell subtypes. By microscopic inspection of the slides, we could find CD8$^+$ and CD4$^+$ T cell clusters mainly in tumors treated with the MATE-expressing virus Ad5/11-αCD3$_{TAT}$-Trimer (Fig. 8E). To quantify eventual cluster formation, we investigated the number of T cells located in proximity to other CD8$^+$ T cells. Tumors treated with anti-PD-1 and Ad5/11-αCD3$_{TAT}$-Trimer showed significantly more CD8$^+$ T cells in direct contact and close to other CD8$^+$ T cells (Fig. 8F). The significantly higher amounts of CD8$^+$ T cells in close proximity to other CD8$^+$ T cells may also reflect the higher total amount of infiltrating CD8$^+$ T cells. We could detect significantly more CD4$^+$ T cells in direct contact or close to CD8$^+$ T cells (Fig. 8G) which would be a prerequisite for a CD4 and CD8 T-cell crosstalk. Consistently, measuring the median distance between CD8$^+$ and CD8$^+$ T cells as well as between CD4$^+$ to CD8$^+$ T cells showed significantly smaller distances in tumors treated with MATE-expressing virus/αPD-1 combination group in comparison to PD-1 monotherapy (Fig. 8H). In summary, our data show that particularly the expression of MATEs by the oncolytic virus promote the interaction of CD8 T cells not only with tumor cells to exert their cytotoxic function but may also promote interaction of CD4$^+$ and CD8$^+$ T cells. A mutual stimulation of both T-cell subtypes probably in intratumoral clusters could be a further explanation for the enhanced immunoactivation in tumors that have been treated with MATE-expressing oncolytic viruses and αPD-1 therapy. Taken together, MATE expression within the tumor leads to an altered T-cell composition and other immune cells within the tumor microenvironment. These alterations are associated with a closer contact of T cells to tumor cells as well as to other T cells. Consequently, we could demonstrate that MATE expression by oncolytic viruses effectively promotes spatial interactions of T cells with tumor cells and other T cells to effectively convert the tumor microenvironment into an immunoactivated state and to facilitate effective PD-1 checkpoint inhibition.

## Discussion

The clinical experience so far suggests that the intrinsic immunogenic properties of oncolytic viruses need further amplification to achieve convincing clinical outcomes in a broader range of patients.

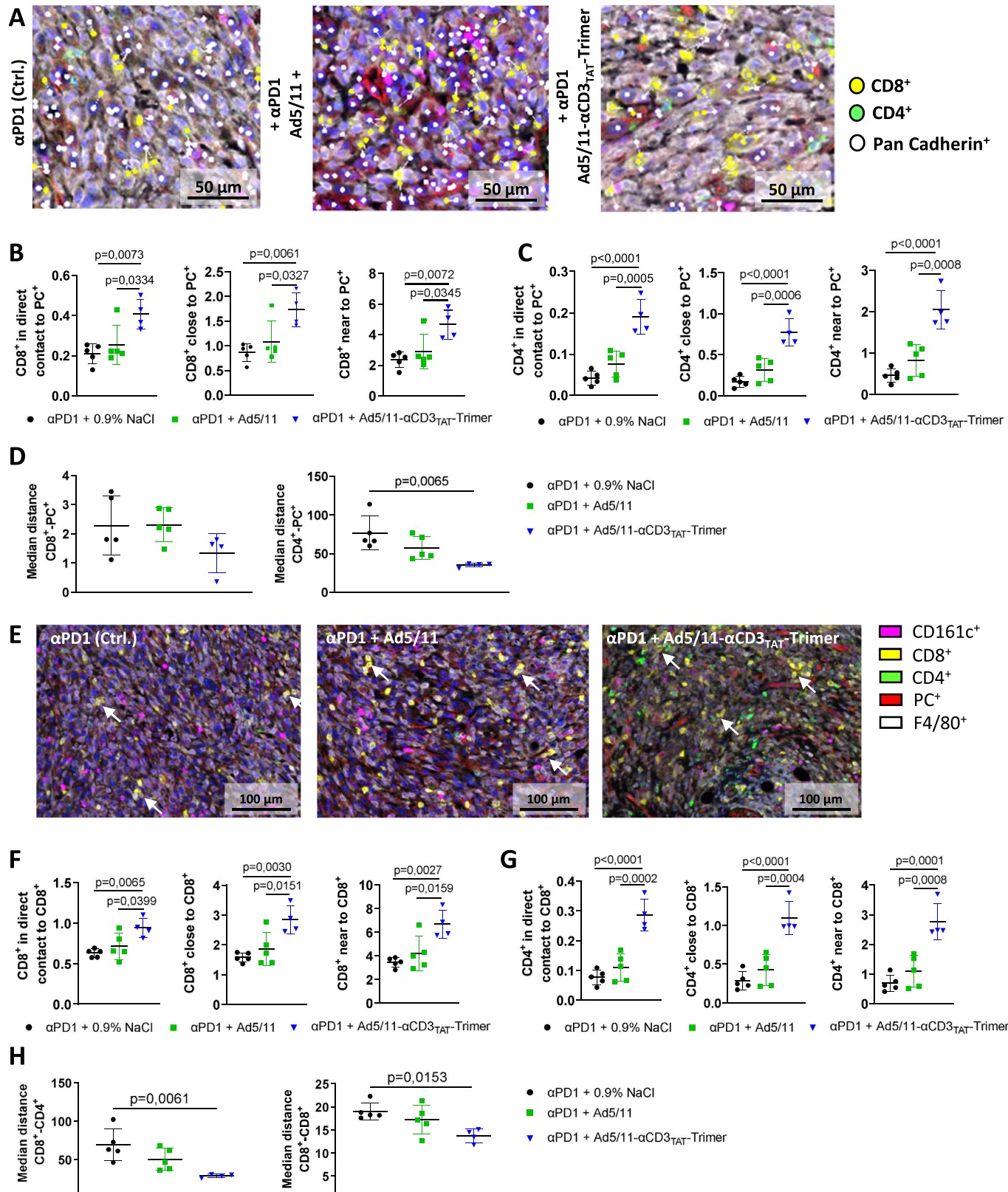

**Figure 8. MATE-expression by oncolytic viruses during αPD1 checkpoint blockade improves the proximity of T cells and tumor cells as well as the interaction of CD4⁺ and CD8⁺ T cells.**

(A–H) S.c. MC38 tumors were treated with a single dose of Ad5/11-αCD3$_{TAT}$-Trimer, Ad5/11, and saline, respectively, and two applications of αPD1 antibodies. Spatial phenotyping of intratumoral immune cells was assessed by multiplex immunohistochemical staining to visualize CD8⁺ T cells, CD4⁺ T cells, CD161c⁺ NK cells, F4/80⁺ macrophages, pan-Cadherin⁺ (PC⁺) tumor cells. The color code is shown in the figure panels. Nuclei have been stained with DAPI. (A–D) Spatial immunohistochemical analysis of the proximity of T-cell subpopulations and tumor cells was performed. (A) Representative composite images from each group reflecting the spatial analysis of CD8⁺ T cells and PC⁺ tumor cells. Closest PC⁺ cell to each CD8⁺ T cell is marked by a line. A distance of 15 μm is defined as direct contact, close proximity is defined by a radius of maximum 30 μm, and near cells are found within a radius of 50 μm. Quantitative analysis of the proximity between tumor cells and CD8⁺ T cells is shown in (B), the proximity of tumor cells and CD4⁺ is shown in (C) ($n = 4$ mice for Ad5/11-αCD3$_{TAT}$-Trimer + αPD1, $n = 5$ for all other groups; mean ± SD; $p$ values by one-way ANOVA with Tukey's post hoc analysis). (D) The median distance in μm between PC⁺ tumor cells and CD8⁺ T cells (left panel) or CD4⁺ T cells (right panel) was analyzed ($n = 4$ mice for Ad5/11-αCD3$_{TAT}$-Trimer + αPD1, $n = 5$ for all other groups; mean ± SD; $p$ values by one-way ANOVA with Tukey's post hoc analysis). (E–H) Spatial immunophenotyping analysis to investigate T-cell clustering within tumors after treatment. (E) Representative composite figures. White arrows indicate CD8 clusters. Distance analysis among CD8⁺ cells is shown in (F) and proximity analysis of CD8⁺ and CD4⁺ T cells is shown in (G) ($n = 4$ mice for Ad5/11-αCD3$_{TAT}$-Trimer + αPD1, $n = 5$ for all other groups; mean ± SD; $p$ values by one-way ANOVA with Tukey's post hoc analysis). (H) Median distance measurements among CD8⁺ T cells (right panel) or between CD8⁺ and CD4⁺ T cell (left panel) ($n = 4$ mice for Ad5/11-αCD3$_{TAT}$-Trimer + αPD1, $n = 5$ for all other groups; mean ± SD; $p$ values by one-way ANOVA with Tukey's post hoc analysis). Source data are available online for this figure.

Several experimental studies have presented BiTE expressing oncolytic viruses as a promising strategy to amplify the immunostimulatory effect of oncolytic viruses thereby focussing the T-cell activation capabilities of BiTEs on the tumor tissue. An oncolytic vaccinia virus armed with an Epha2-directed BiTE and the oncolytic adenovirus ICOVIR-15K expressing an EGFR-targeting BiTE, respectively, showed robust T-cell activation and bystander cell-mediated cytotoxicity in vitro and in murine tumor models (Yu et al, 2014; Fajardo et al, 2017). An Epcam-specific-BiTE potentiated the cytotoxicity of the oncolytic adenovirus Enadeno-tucirev and activated both CD4⁺ and CD8⁺ T cells in malignant ascites (Freedman et al, 2017). However, this strategy limits the range of treatable tumors to entities which are well known to frequently express the target antigen. Furthermore, molecular diagnostics for target verification will be required to justify application of a BiTE-expressing oncolytic agent. A true challenge for antigen-specific strategies in general is the heterogeneity of antigen expression in tumors (Gerlinger et al, 2012). Antigen-negative foci will probably remain unresponsive to therapy and contribute to tumor regrowth. In a process of immunoediting, tumors can also adapt by epigenetically downregulating the antigen thus facilitating escape from antigen-specific T-cell attacks (Dunn et al, 2004). These considerations limit the potential of oncolytics, armed with antigen-specific T-cell engagers, for pharmacological development. To facilitate broad applicability, we have generated MATEs for immunological arming of oncolytic viruses which act as CD3-binding T-cell engagers irrespective of a particular tumor antigen. Binding of MATEs to potential target cells via the Tat-PTD should promote CD3 clustering, T-cell activation and the formation of a pseudoimmunological synapse to allow for target cell killing. Assuming that the aspect of CD3 clustering on T cells is critical for their activation (Haas et al, 2009), we also generated a trimeric version of MATEs which may not only influence CD3 clustering, but also affinity of Tat-medited cell-binding. Both MATEs bound to CD8⁺ and CD4⁺ T cells in isolated murine splenocytes and led to effective T-cell activation as shown by several activation markers, IFNγ production and T-cell proliferation. Membrane attachment mediated by the Tat-PTD was demonstrated to be essential for T-cell activation because Tat-free control proteins and vice versa Tat-proteins lacking the CD3-binding function did not lead to T-cell activation. Application of MATEs in coculture experiments resulted in increased death of MC38 cancer cells in vitro. These findings prove that both functional domains of MATEs were relevant for successful formation of a pseudoimmunological synapse needed for T-cell activation and target cell killing. It is well known that the Tat-PTD binds to biomembranes with significant affinity (Ziegler and Seelig, 2004) and we also demonstrated the essential function of the Tat-PTD domain in MATEs to enable T-cell activation. Nevertheless, it was difficult to detect Tat-PTD-dependent binding to the cell surface of target cells by flow cytometry suggesting that rapid uptake of MATEs upon cell binding may play a considerable role in our experiments. It is known that, even when linked to larger proteins, the Tat-PTD can mediate cell entry within minutes (Fawell et al, 1994). In our study, it has to be considered that a constitutive, Tat-PTD-mediated cellular uptake of MATEs presumably affects bioavailability, efficacy, and also toxicity. This may lead to reduced intratumoral availability and T-cell activation when compared with a BITE approach. On the other hand, the efficacy of the Tat-PTD mediated MATE resorption also represents a safety hallmark in our strategy preventing that excessively secreted MATEs reach toxic levels. However, there are certainly options to further improve the corresponding characteristics of MATEs by choosing alternative PTDs or oligopeptides harboring modified arrays of arginine, lysine and histidine residues to modify affinity or membrane penetration properties (Ho et al, 2001).

An important goal of arming oncolytic viruses with T-cell engagers is enforced T-cell activation within the tumor beyond the sites of infection. It is known that individual T cells, once activated by BiTEs can kill multiple target cells (Hoffmann et al, 2005). Consistently, we observed that T-cell activation in the presence of MATE-expressing tumor cells was superior compared to the activation by a membrane-bound variant of the αCD3-binding scFv supporting our hypothesis that MATEs are capable of activating T cells distant from the origin of their expression. This was further confirmed in cytotoxicity assays showing an improved bystander cell killing when MATEs were actually secreted compared with the membrane-bound variant. For future application in humans, we also generated MATEs harboring a scFv from an established antibody specific for human CD3. Our data show that humanized MATEs bind effectively to CD8⁺ and CD4⁺ T cells in primary human PBMCs and promoted their activation and production of IFNγ and TNFα.

It has to be emphasized here that application of MATEs, as non-target selective agents, are not intended for systemic injections.

Intralesional delivery by an oncolytic virus should minimize extra-tumoral distribution of MATEs and prevent off-target T-cell activation. In our study, we investigated therapeutic efficacy of MATEs after intratumoral delivery with Ad5/11 as carrier oncolytic adenovirus using a genetic arrangement which facilitates low expression level of MATEs. Despite low expression levels, supernatants from 293 cells infected with MATE-expressing Ad5/11 were capable of activating T cells. Consistently, CD8[+] and CD4[+] T cells were activated following intratumoral injection of MATE-expressing Ad5/11 in a subcutaneous murine model. Furthermore, the reduced replicative potency of Ad5/11 in general and the further diminished replication in normal cells are additional features to realize safe applicability. In our study, we exemplarily used oncolytic viruses as delivery vehicles to combine oncolysis and intratumoral delivery of MATEs. However, alternative vehicles for MATE expression, such as LNPs, can also be considered, provided that they effectively target the tumor tissue or are intratumorally applied. Furthermore, the option of a locoregional application, such as an injection into the tumor-supplying artery could be interesting to investigate. Importantly, we could confirm that intralesional injection of low-level MATE-expressing Ad5/11 did not result in a detectable activation of T cells in tumor draining lymph nodes at an early time point as a surrogate for a release of MATEs from the tumor rather than T-cell priming by DCs presenting the tumor debris. This observation confirms that the intratumorally produced MATEs were either retained by the tumor tissue or had been sufficiently diluted to prevent T-cell activation. On the other hand, we found indeed T-cell activation in tumor-draining lymph nodes when we used Ad5/11 variants with elevated MATE expression level suggesting that excessively secreted MATEs can be released from the tumor into the lymphatic system. However, even with Ad5/11 expressing elevated MATE levels, we did not observe any signs of systemic toxicity such as liver damage or of aberrant immune activation related to a cytokine release syndrome. Therefore, our results demonstrate that intratumoral application of MATE-expressing oncolytic virus might be a relatively safe strategy. In general, intratumoral injection is the currently preferred application mode of oncolytic agents to focus immune activation on the tumor tissue and its immune microenvironment. In line with these findings, recent results by Chiocca and collaborators have clearly linked local immunoactivation induced by oncolytic viruses with survival of patients after virotherapeutic treatment thus emphasizing the need for intratumoral delivery (Ling et al, 2023).

Because of the limited permissivity of murine models for adenovirus replication and lysis, our investigations may not reflect the true extent of MATE expression levels and subsequent toxicity observable in a permissive human tumor. Syrian hamster has been described as a promising immunocompetent model permissive for replication of human adenoviruses and vaccinia virus in both normal and tumor tissue (Thomas et al, 2006; Bortolanza et al, 2007; Tysome et al, 2012). Since tumor cell lines from various tumor entities, are already available for this model, it is well suited to evaluate efficacy and toxicity resulting from both vivid viral replication and high levels of virus-encoded immunoactivators (Jia et al, 2023). Consequently, syrian hamsters have already been used for numerous preclinical studies on immunologically-armed oncolytic adenoviruses (Bortolanza et al, 2009; Wang et al, 2017). A future study in a hamster model would therefore be mandatory to fully assess the toxicologic profile of MATE-expressing Ad5/11. To reassure that MATEs are a promising tool for virotherapeutics it

could also be an alternative approach to study MATEs when incorporated in oncolytic virus types that show higher replication in murine models such as vesicular stomatitis virus or vaccinia virus (Stojdl et al, 2003; Breitbach et al, 2007).

In our in vivo experiments, we investigated Ad5/11 variants expressing either the monomeric or the trimeric version of MATEs at both an early and a later time point following treatment. Whereas the monomeric MATE showed a higher T cell activation early after infection, the trimer led to effective immunoactivation at later time points. However, it has to be considered that our expression data suggest that the trimer, due to the higher affinity and resorption, is less effectively released and a critical level of available trimer may therefore lag behind when comparing with the monomer. Differential release kinetics would particularly affect the early analysis of T-cell activation whereas the later time point of analysis is presumably less prone to reflect differences in expression kinetics. Despite lower release, the trimer resulted in effective immunoactivation characteristics in tumor tissue in the longer course which might argue for a qualitative advantage of the trimer in T-cell activation. The kinetics of immediate T-cell activation in vivo should be addressed in future studies using shorter time intervals of analysis to clarify these mechanistical aspects.

Regarding the therapeutic potential of MATEs, we demonstrated that their expression by Ad5/11 significantly delayed tumor growth and improved survival in subcutaneous MC38 colon cancers in mice including complete regressions. This finding demonstrate that MATEs represent a promising strategy for the generation of T-cell engager armed oncolytic viruses with a theoretically broad spectrum of target tumors provided that a direct injection is feasible. Alternative approaches to enlarge the target spectrum of T-cell engager-armed oncolytics have addressed molecular targets present on essential tumor tissue components beyond malignant cells. An enadenotucirev variant expressing a BiTE against fibroblast activating protein (FAP) on fibroblasts has been shown to induce T-cell activity in the tumor stroma (Freedman et al, 2018). Expression of BiTEs targeting CD206 or folate receptor β (FRβ) on M2 macrophages achieved both intratumoral T-cell activation and a diminished immunosuppression (Scott et al, 2019). Our analyses on intratumoral immune cell composition showed that MATE expression by Ad5/11 moderately affected intratumoral T-cell infiltration compared with viral infection per se. However, MATEs significantly improved both the individual CD8/CD4 ratio and the intratumoral frequency of tumor-antigen-specific CD8[+] T cells indicating an augmented tumor-directed immune response. The role of CD8[+] T cells for therapeutic efficacy of MATEs was confirmed because a systemic depletion of CD8[+] T cells eliminated the MATE-dependent survival advantage. When using purified MATEs, the trimer was more effective under saturated assay conditions consistent with our hypothesis that a trimerized form could have an advantageous effect on CD3 clustering. On the other hand, when expressed by Ad5/11, higher levels of freely available monomeric MATE were detectable consistent with a higher T-cell activation when using supernatants obtained from infected cells. This observation is probably due to the lower avidity of the monomer and a stronger retention of the trimer to the MATE-expressing cells. In vivo, both viruses showed a comparable antitumor efficacy suggesting that the intrinsically higher T-cell activating properties of the trimer is probably outweighed by the intratumoral distribution of the monomer. Consistent with a higher

availability of the monomer, results of luminex multiplex assays suggested an elevated extent of intratumoral inflammation. Remarkably, only the monomer led to elevated levels of inflammatory markers associated with immunosuppressive properties such as IL-6, IL-4, and IL10. In contrast, our results on intratumoral immune cell subsets after virotherapy showed that expression of the trimer yielded a more focused T-cell activation profile as confirmed by the favorable CD8/CD4 ratio and the highest frequency of tumor-directed CD8$^+$ T cells against the Adpgk neoepitope in MC38. This was also accompanied by an increased density of PD1$^+$ cells. Consistent with the notion that intratumoral immune activity is an important prerequisite for sensitivity to PD1/PD-L1 checkpoint blockade (Topalian et al, 2012), we demonstrated that Ad5/11 expressing the trimeric MATE, but not the empty virus, effectively breaks tumor resistance to systemic application of a PD1 antibody. Synergy of MATE-expressing virotherapy and checkpoint inhibition led to tumor-free survival of the majority of treated mice associated with an improved CD8/CD4$^+$ T-cell ratio and an elevated frequency of XCR1$^+$ dendritic cells which have been shown to be particularly relevant for the cross presentation of tumor antigen (Bachem et al, 2010). Spatial analyses of immune cell infiltrates by multiplex immunohistochemistry indicated an improved proximity of CD8$^+$ T cells, but not CD4$^+$ T cells, with cancer cells enhancing the probability of immunological synapse formation and tumor cell killing. In addition, we found a closer interaction of CD8$^+$ T and CD4$^+$ T cells suggesting a clustering of these two T cell subtypes. In this regard, it is interesting that we found a higher abundance of XCR1+ cross-presenting DCs. It has been recently shown that within tumor tissue close interactions between both CD8$^+$ and CD4$^+$ T cells at cross-presenting DCs form effective immunological triads important for driving antitumoral T-cell immunity (Espinosa-Carrasco et al, 2024). Together, our data show that MATE-expressing oncolytic viruses such as Ad5/11-αCD3$_{TAT}$-Trimer are powerful tools for localized activation of intratumoral T-cell immunity and may function as effective door-opener for PD1/PD-L1 checkpoint inhibitors in a broad range of tumors.

# Methods

## Genetic engineering of MATEs

For the generation of murinized MATEs including the transmembrane variant, a synthetic DNA-template mCD3 (obtained from Geneart) was used, coding for αCD3$_{TM}$. The coding sequence was equipped with an N-terminal signal peptide derived from the coxsackievirus adenovirus receptor to enable protein secretion, an anti-CD3ε VL and VH-chain from an established αCD3 antibody (Gilliland et al, 1996) connected by a glycine-serine linker, the CH1-CH3 domains of the heavy chain of murine IgG2b including the hinge region to provide for flexibility and spacing between the two functional parts of this adapter protein. Furthermore, a myc-tag and a His-tag were inserted downstream of the αCD3ε domain for detection and purification purposes. C-terminally, a sequence was integrated coding for the amino acids SNKAGLIAGAIIGTLLALALIGLIIFCCRKKRREEK corresponding to the transmembrane region of the human coxsackievirus adenovirus receptor. The DNA was digested with Bgl II and Nhe I and integrated into the expression vector pT3-EF1α resulting in the plasmid MG1. The soluble, monomeric MATE was generated by PCR using the primers M-fw, and M-rev, an oligonucleotide complementary to the C-terminal part of the Hinge region and harboring a non-complementary overhang to introduce the Tat-PTD of HIV followed by a stop codon. Integration into pT3-EF1α was performed as described above resulting in the plasmid MG5. The trimeric conformation of MATEs was generated by assembly PCR. One fragment harboring the fibritin domain (amino acid sequence GYIPEAPRDGQAYVRKDGEWVLLSTFLSPA) was generated with primers Fib-fw and Fib-rev using the plasmid pCARsc-pSia as template which has been decribed previously (Kloos et al, 2015). Using MG5 as template, a second fragment was generated using primers Tat-fw and Tat-rev. A fragment harboring the protein from the N-terminus to the C-terminal part of hinge was amplified using primers P7-fw and P8-rev. The fragments were linked and integrated in pT3-EF1α by using Gibson Assembly (NEB) according to the manufacturer's protocol resulting in the plasmid MP4. Human MATEs were generated using the DNA fragment hCD3 (obtained from Geneart) encoding an anti-human CD3ε-scFv corresponding to the established sequence of Blinatumomab (Löffler et al, 2000). The DNA was integrated into MG5 and MP4 using Bgl II and Xho I thereby replacing the murine CD3ε scFv and resulting in the plasmids MP1127 for expression of the monomeric MATE, and MP1166 for expression of the trimeric MATE, respectively. All generated material

### Reagents and tools table

| Reagent/Resource | Reference or Source | Identifier or Catalog Number |
|---|---|---|
| **Experimental models** | | |
| HEK293 cells | ATCC | CRL-1573 |
| 293T cells | ATCC | CRL-3216 |
| MC38 cells | University Mannheim, Germany | kindly provided by Prof. Michael Neumaier |
| Primary human hepatocytes | Lonza | HUCPG Lot: HUM180201A |
| human peripheral blood mononuclear cells (PBMCs) | Healthy, voluntary donors | |
| Hep3B | ATCC | HB-8064 |
| H1299 | ATCC | CRL-5803 |
| murine splenocytes | C57BL/6-J mice | Animal facility, Hannover medical School |
| C57BL/6-J mice | | Animal facility, Hannover medical School |

| Reagent/Resource | Reference or Source | Identifier or Catalog Number |
|---|---|---|
| **Recombinant DNA** | | |
| pQCXIN | Clontech/Takara | #631514 |
| pQCXIN- αCD3$_{TAT}$ | This work | |
| pQCXIN- αCD3$_{TAT}$-Trimer | This work | |
| pQCXIN- αCD3$_{TM}$ | This work | |
| Ad5/11 | This work | |
| Ad5/11-αCD3$_{TAT}$ | This work | |
| Ad5/11-αCD3$_{TAT}$-Trimer | This work | |
| Ad5/11-PGK-αCD3$_{TAT}$ | This work | |
| Ad5/11-PGK-αCD3$_{TAT}$-Trimer | This work | |
| pCMV-VSV-G | Addgene | Plasmid 8454 |
| pCMV-gag-pol | Hannover Medical School | Kindly provided by Prof. Axel Schambach |
| NW17991 | Hannover Medical School | Kindly provided by Dr. Norman Woller |
| HN60-pPGK-SB13 | Univ. of Minnesota | Kindly provided by Dr. David Largaespada |
| **Antibodies** | | |
| anti-c-myc clone 9E10 | Santa Cruz Biotechnology | sc-40 |
| HRP-linked polyclonal α-mouse-IgG | Cell Signaling Technology | CST7076 |
| αPD-1 clone RMP1-14 | BioXCell | #BE0146 |
| CD8a GoInVivo™ clone 53-6.7 | Biolegend | #100767 |
| CD90.2-PerCP clone 53-2.1 | Biolegend | #140316 |
| CD45.2-PerCP clone 104 | Biolegend | #109826 |
| CD8a-FITC clone 53–6.7 | Biolegend | #100706 |
| CD8a-APC/Cyanine7 clone 53–6.7 | Biolegend | #100712 |
| CD8a-PE clone 53–6.7 | Biolegend | #100708 |
| CD4-PE clone GK1.5 | Biolegend | #100408 |
| CD4-FITC clone GK1.5 | Biolegend | #100406 |
| CD11c-APC/Cyanine7 clone N418 | Biolegend | #117324 |
| CD11c-APC clone N418 | Biolegend | #117310 |
| F4/80-FITC clone BM8 | Biolegend | #123108 |
| F4/80-APC/Cyanine7 clone BM8 | Biolegend | #123118 |
| IFNγ-APC clone XMG1.2 | Biolegend | #505810 |
| NK1.1-PE clone PK136 | Biolegend | #108708 |
| CD69-PE clone H1.2F3 | Biolegend | #104508 |
| CD69-APC clone H1.2F3 | Biolegend | #104514 |
| CD25-PE clone PC61 | Biolegend | #102008 |
| CD25-APC clone PC61 | Biolegend | #102012 |
| Ki67-PE clone SolA15 | eBioscience | #61-5698-82 |
| CD3-APC/Cyanine7 clone 17A2 | Biolegend | #100222 |
| CD44-APC clone IM7 | Biolegend | #103012 |
| CD44-FITC clone IM7 | Biolegend | #103006 |
| CD49b-APC clone DX5 | Biolegend | #108909 |
| MHCII (I-A/I-E)-PE clone M5/114.15.2 | Biolegend | #107608 |
| XCR1-APC clone ZET | Biolegend | #148206 |
| TruStain FcX (CD16/32) clone 93 | Biolegend | #101320 |
| IgG1-PE clone RMG1-1 | Biolegend | #406608 |

| Reagent/Resource | Reference or Source | Identifier or Catalog Number |
|---|---|---|
| FoxP3-APC clone FJK-16s | eBioscience | #17-5773-82 |
| TNFα-PE clone MP6-XT22 | Biolegend | #506306 |
| CD4-PerCP clone OKT4 | Biolegend | #317432 |
| CD4-PE clone OKT4 | Biolegend | #317410 |
| CD8-APC clone HIT8a | Biolegend | #300912 |
| CD8-FITC clone HIT8a | Biolegend | #300906 |
| CD8-APC/Cyanine7 clone HIT8a | Biolegend | #300926 |
| CD69-PE clone FN50 | Biolegend | #310906 |
| CD137-PE clone 4B4-1 | Biolegend | #309804 |
| CD137-APC clone 4B4-1 | Biolegend | #309810 |
| IFNγ-APC clone B27 | Biolegend | #506510 |
| TNFα-PE clone Mab11 | Biolegend | #502909 |
| CD3 clone D4V8L | Cell Signaling Technology | CST99940 |
| CD4 clone D7D2Z | Cell Signaling Technology | CST25229 |
| CD8 clone D4W2Z | Cell Signaling Technology | CST98941 |
| CD161c clone E6Y9G | Cell Signaling Technology | CST39197 |
| F4/80 clone D2S9R | Cell Signaling Technology | CST70076 |
| PanCadherin polyclonal | Cell Signaling Technology | CST4068 |
| PD-1 clone D7D5W | Cell Signaling Technology | CST84651 |
| goat anti-rabbit IgG (HRP) | Abcam | Ab205718 |
| Alexa Fluor® 488 goat anti-rabbit IgG (H + L) | Invitrogen | A11034 |
| E1A | Bioss | BSS-BS-6136R |
| myc AB | Invitrogen | AB_25355826 |
| Alexa Fluor® 568 goat anti-chicken IgG (H + L) | Invitrogen | AB_2535826 |
| **Oligonucleotides and other sequence-based reagents** | | |
| MC38 mutated neoantigen adpgk-R304M | GenScript | ASMTNMELM |
| mCD3 | Geneart | Gilliland et al, 1996 |
| hCD3 | Geneart | Löffler et al, 2000 |
| M-fw | Eurofins Genomics | TACTCCGAAAGATCTGTCGACATGGCCCTCCTGCTG |
| M-rev | Eurofins Genomics | TTGCTAGCGGCCGCCTATCTTCGTCGCTGTCTCCGCTTC TTCCTGCCTTTGCCAGGAGTCCGGGAGATGGTCTTCTT |
| Fib-fw | Eurofins Genomics | AAGACCATCTCCCGGACTCCT GGCAAAGGTTATATTCCTGAAGCT |
| Fib-rev | Eurofins Genomics | TCCGGACCCTCCTCCTCCTGCTGGTGATAAAAAGGT |
| Tat-fw | Eurofins Genomics | TCACCAGCAGGAGGAGGAGGGTC CGGAGGCAGGAAGAAGCGGAGA |
| Tat-rev | Eurofins Genomics | GAGTCCCGGGTACGCGTTGCTAGCT GATCTATCTTCGTCGCTGTCTCCG |
| P7-fw | Eurofins Genomics | TTTCGACATTTTAAGATCTAGATATGGCC CTCCTGCTGTGC |
| P8-rev | Eurofins Genomics | TTTGCCAGGAGTCCGGGAGAT |
| Age-CarL-fw | Eurofins Genomics | AAACCGGTATGGCCCTCCTGCTGTGC |
| BamHI-TAT-rev | Eurofins Genomics | AAGGATCCCTATCTTCGTCGCTGTCT |
| BamHI-TM-rev | Eurofins Genomics | AAGGATCCTCATTTTTCTTCTCTGCG |
| CMV-fw | Eurofins Genomics | CTAGCATCGTAACTATAACGGT CCTAAGGTAGCGAAAGCTAGTACCATGGTGATGC |
| CMV-rev | Eurofins Genomics | CCCTCGAGGATCTGACGGTTCACTAAACCAG |

| Reagent/Resource | Reference or Source | Identifier or Catalog Number |
|---|---|---|
| Swa-fw | Eurofins Genomics | ATAAGCTTGATCACGCGTTTCGAACGATTTA AATTTAAATCTCGAGAATAAAGGTACCTT |
| Swa-rev | Eurofins Genomics | AAGGTACCTTTATTCTCGAGATTTAAATTTAAATCGTTCG AAACGCGTGATCAAGCTTAT |
| E4-fw | Eurofins Genomics | TACTGCGCGCTGACTCTTAAGGACTAGTTTCGC |
| E4-rev | Eurofins Genomics | CGGTACCCCGGGTTCGAAATCGTTAATTAACGATGTA |
| PGK-fw | Eurofins Genomics | TACCGGGTAGGGGAGGCGCTTTTCCCAAGGCAG |
| PGK-rev | Eurofins Genomics | TCGAAAGGCCCGGAGATGAGGAAGAGGAGAACAGC |
| **Chemicals, Enzymes and other reagents** | | |
| Ni-NTA-Agarose Beads | Qiagen | #30210 |
| imidazole | AppliChem | A3635,0050 |
| L-Histidin | Roth | 3852.2 |
| DMEM+GlutaMAX™ | Gibco | 31966-021 |
| FCS | Biowest | S1810 |
| Penicillin/streptomycin | Bio&Sell | BS.A2213 |
| RPMI GlutaMAX™ | Gibco | 72400-21 |
| HCM™ hepatocytes culture medium Kit | Lonza | CC-3198 |
| Lipofectamine | Invitrogen | 18324012 |
| Polybrene/Hexadimethrine bromide | Sigma-Aldrich | 107689 |
| Geneticin G-418 | Roth | CP11.3 |
| Pancoll | PAN Biotech | P04-601000 |
| RBC Lysis buffer | Biolegend | #420302 |
| Zombie Green™ Fixable Viability Kit | Biolegend | #423112 |
| CellTrace™ CFSE Cell Proliferation Kit | Invitrogen | C34554 |
| Zombie NIR™ Fixable Viability Kit | Biolegend | #423106 |
| CsCl/Cesium Chloride | AppliChem | A1098 |
| mouse serum albumin | Sigma-Aldrich | A3139 |
| AdenoX™ Rapid Titer Kit | Takara | 632250 |
| MTT/ Thiazolyl Blue Tetrazolium Bromide | Sigma-Aldrich | M5655 |
| DAPI | Roth | 6335.1 |
| DMSO | Sigma-Aldrich | D5879 |
| DNAse | Sigma-Aldrich | DN26 |
| Hyaluronidase | Sigma-Aldrich | H3884 |
| Collagenase IA | Sigma-Aldrich | C9891 |
| Collagenase IV | Sigma-Aldrich | C5138 |
| GolgiPlug™ | BD Biosciences | 555029 |
| CytoFix/CytoPerm Kit | BD Biosciences | 554714 |
| Foxp3/Transcription Factor Staining Kit | eBioscience | 00-5523-00 |
| Bio-Plex™ Cell Lysis Kit | Biorad | #171304011 |
| BCA Protein Assay Kit | Pierce | 23227 |
| Bio-Plex Pro Mouse Cytokine 23-plex kit | Biorad | M60009RDPD |
| Hematoxylin/Mayer's Hemalun solution | Merck | 1.09249.0500 |
| Eosin/Eosin Y Solution Aqueous | Sigma-Aldrich | HT110232 |
| DAB | Vector Laboratories | SK-4105 |
| Hoechst 33258 | Thermo Scientific | H1399 |

| Reagent/Resource | Reference or Source | Identifier or Catalog Number |
|---|---|---|
| Opal™ 4-Color Anti-Rabbit Automation IHC Kit | Akoya Biosciences® | NEL830001KT |
| Opal™ 620 Reagent Pack; | Akoya Biosciences® | SKU FP1495001KT |
| Opal™ 780 Reagent Pack | Akoya Biosciences® | SKU FP1501001KT |
| Opal™ Polymer Anti-Rabbit HRP Secondary Antibody Kit | Akoya Biosciences® | ARH3001KT |
| LEGENDplex™ Multi-Analyte Flow Assay Kit | Biolegend | 741024 |
| ALT Activity Assay | RayBiotech | MA-ALT |
| Western Lightning® Plus ECL | Perkin Elmer | 50-904-9326 |
| Software | | |
| FlowJo 10 | https://www.flowjo.com | |
| GraphPad Prism 10 | https://graphpad.com | |
| NIS Elements BR Version 5.20.00 | https://microscope.healthcare.nikon.com | |
| inForm v2.4.10 | https://akoyabio.com/support/software/inform-tissue-finder-software/ | |
| Phenoptr&phenoptrReports | https://akoyabio.com/support/support/software | |
| LEGENDplex™ Data Analysis Software Suite | https://www.biolegend.com/de-de/immunoassays/legendplex/support/software | |
| QuPath | https://qupath.github.io/ | |
| BioRender | https://biorender.com | |
| ImageJ | https://imageJ.net | |
| Other | | |
| NanoDrop Lite | Thermo Scientific | |
| FACS Canto II | BD | |
| Bio-Plex Reader | Biorad | |
| Nikon Eclipse Ti2 | Nikon | |
| Nikon DS-Ri2 | Nikon | |
| Vectra Polaris | Akoya Biosciences | |
| Microplate Reader MQX200 | BioTek | |

including plasmids, transgenic cells, and viruses can be provided for research purposes upon request.

## MATE protein isolation

The recombinant murine and human MATEs were isolated using affinity purification using Ni-NTA-Agarose Beads (Qiagen). 72 h post-seeding of stable-transduced MATE-expressing HEK293 cells, the supernatant was collected and filtered (45 μm). Sodium phosphate buffer (pH 8.0; 50 mM) and Ni-NTA-Agarose beads were added to the filtrate. The filtrate was then rotated overnight in an overhead shaker at 4 °C. After pelleting, beads were washed twice with phosphate buffer (pH 8.0; 50 mM) containing NaCl (150 mM) and imidazole (1 mM). Elution of MATEs was carried out by a 4 h incubation at 4 °C in an overhead shaker with phosphate buffer (pH 8.0; 50 mM) containing NaCl (150 mM) and L-Histidin (100 mM). After centrifugation, the eluate was collected, protein concentration was measured using Nanodrop and quality was assessed by western blot. The final protein solution was snap-frozen and stored at −80 °C until use.

## Western blot

Western blot analyses were performed according to standard protocols using Ni-NTA affinity purified MATE preparations. Samples were either separated by conventional 10% SDS-PAGE under denaturing conditions or by a Blue-native PAGE according to a previously described method (Wittig et al, 2006) that was partially modified. In brief, 0.5 μg of MATEs were mixed with 5% glycerol and 0.01% Ponceau S and were loaded onto a 5% acrylamide gel buffered with Tris-HCl, pH 8.8. Cathode buffers containing Coomassie blue G-250 and the anode buffer were used as described in the original protocol. As controls, 8 μg of bovine serum albumin (66 kDa) and alcohole dehydrogenase (150 kDa) were used which has been kindly provided by Jana Führing (Dept. Biochemistry, Hannover Medical School). After protein separation and protein transfer to a PVDF membrane, the membrane was cut between standards and samples. The membrane with standards was then destained with 25% Methanol/10% acetic acid to remove excess Coomassie blue. The membrane with MATE samples was

completely destained for 3 min in 100% methanol. For detection of recombinant MATE proteins an anti-c-myc antibody (1:5000, clone: 9E10, Santa Cruz Biotech) was used as a primary antibody at 4 °C overnight. As secondary antibody, a HRP-linked polyclonal α-mouse-IgG (1:5000, Cell Signaling Technology®) was applied for 2 h at RT. Both membrane parts were then correctly reassembled and developed using Western Lightning ECL Plus Reagent.

## Cell lines and cell culturing

All cancer cell lines were cultured in DMEM+GlutaMAX™ (Gibco) medium supplemented with 10% (v/v) FCS (Biowest), 100 U/ml penicillin and 100 mg/ml streptomycin (Bio&Sell) at 37 °C and 5% $CO_2$. HEK293, 293T, Hep3B, and H1299 cells were obtained from ATCC. Only early passages have been used. The murine cancer cell line MC38 was kindly provided by Michael Neumaier, University Mannheim, Germany. MC38 cells were authenticated by expression of the neoantigen Adpgk-R304M. Cell lines were regularly tested for mycoplasma contamination. Isolated murine splenocytes and human PBMCs were cultured with or without MATEs for 24–48 h in RPMI +GlutaMAX™ (Gibco) medium supplemented with 10% (v/v) FCS (Biowest), 100 U/ml penicillin and 100 mg/ml streptomycin (Bio&Sell) at 37 °C and 5% $CO_2$. Primary human hepatocytes (obtained from Lonza; Lot: HUM180201A) were cultured in HCM™ hepatocytes culture medium Kit (Lonza) at 37 °C and 5% $CO_2$. Using the plasmids MG1, MG5, and MP4 (described below) as templates, MATE-encoding sequences were PCR-amplified using the primers AgeI-CarL-fw and BamHI-TAT-rev, or BamHI-TM-rev, respectively. The fragments were then integrated into the AgeI and BamHI sites of pQCXIN (Takara-Clontech) to yield retroviral expression vectors pQCXIN-αCD3$_{TAT}$, pQCXIN-αCD3$_{TAT}$-Trimer and pQCXIN-αCD3$_{TM}$. Retroviral particles were produced in HEK293T cells after lipofectamine (ThermoFisher) cotransfection of the pQCXIN vectors along with plasmids for expression of VSV-G and gag-pol. Retrovirus containing supernatants were collected, supplemented with polybrene (8 µg/ml, Sigma) and then applied to the target cells. After 48 h cells were selected with culture medium containing G418/geneticin (1 mg/ml).

## Splenocyte isolation

Spleens were mashed through a nylon cell strainer (40 µm). The cell strainer was rinsed with 10 ml RPMI and the cells were centrifuged (10 min at $300 \times g$ and 4 °C). The supernatant was discarded and the cell pellet was resuspended in 1x RBC lysis buffer (5 ml per spleen) and incubated on ice for 5 min. By adding 15 ml RPMI the reaction was stopped. Subsequently, the splenocytes were pelleted by centrifugation (10 min at $300 \times g$ and 4 °C).

## PBMC isolation

Isolation of human peripheral blood mononuclear cells (PBMCs) isolation was performed using 8–16 ml of whole blood from healthy donors. Protocol and study design have been approved by the ethical committee of the Hannover Medical School. Blood was mixed with PBS (1x) and 35 ml blood/PBS mixture was layered onto 15 ml Pancoll (PAN Biotech). The gradient was centrifuged for 15 min at RT at $1000 \times g$ without brake. The cell layer containing mononuclear cells collected and cells were washed once with PBS. After pelleting the cells, erythrocyte lysis was performed by adding 10 ml of RBC Lysis buffer (Biolegend). After an additional washing step, cells were used for activation assays.

## Binding assays

Binding assays were performed with freshly isolated murine splenocytes or human PBMCs. Cells were incubated with 0.5–1 µg purified MATEs for 15 min at RT. Subsequently, MATE binding was determined by flow cytometry using an anti-c-myc antibody (1:100).

## T-cell activation and in vitro cytotoxicity assays

For assays using purified MATE proteins, splenocytes ($1 \times 10^6$ cells) or PBMCs ($0.5 \times 10^6$ cells) in 1 ml RPMI+GlutaMAX™ medium (Gibco; 10% FCS, 1% P/S) were seeded in a 24-well plate. The amount of MATE proteins as indicated in the figures, was added to the cell suspension. Splenocytes were cultured for 24–48 h at 37 °C and 5% $CO_2$. Subsequently, flow cytometry analysis of activation markers was performed using antibodies against CD69 and CD25 (1:100 each).

T-cell activation assays with supernatants of MATE-expressing virus infected HEK293 were performed 48 h after infection. For these experiments, a multiplicity of infection (MOI) of 10 was used. The MATE-containing supernatant was collected and filtered (0.2 µm). Splenocytes ($1 \times 10^6$ cells) were incubated for 24 h (37 °C, 5% $CO_2$) with 0.5 ml fresh RPMI+GlutaMAX™ (Gibco) and 0.5 ml of filtered supernatant.

For T-cell killing assays $1 \times 10^5$ MC38 cells were suspended in RPMI+GlutaMAX™ medium (Gibco; 10% FCS, 1% P/S) and seeded into a 24-well plate. Additional 1 µg of isolated MATE protein was added per well. After a 10 min incubation step at RT, 0.5 ml splenocyte suspension ($2 \times 10^6$ cells/ml) was added. After 24 h of culture, dead MC38 cells were stained with Zombie Green™ Fixable Viability Kit (Biolegend) according to manufacturer instructions. The labeled cells were analyzed via flow cytometry.

## T-cell activation and in vitro cytotoxicity assays using mixed target cells

MC38 murine tumor cells stably expressing MATEs (αCD3$_{TAT}$/αCD3$_{TAT}$-Trimer) or a membrane-bound T-cell engager (αCD3$_{TM}$) were generated by retroviral transduction as described above. MATE-expressing cells were seeded together with paternal, non-transgenic MC38 in a 96-well plate and cultured overnight. Isolated splenocytes were added to the MC38 cell mixture at a ratio of 1:10 (target:splenocyte). Splenocytes and MC38 cells were co-cultured for 24 h (37 °C, 5% $CO_2$). Subsequently, flow cytometric analyses of activation markers were performed. For the respective cytotoxicity assays, MATE-transgenic MC38 were labeled using 5 µM CFSE according to manufacturer's recommendation (CellTrace™ CFSE Cell Proliferation Kit, Invitrogen). CFSE-labeled, transgenic MC38 and empty MC38 were seeded at a ratio of 1:50 in 24-well plates in a total cell number of $1 \times 10^5$ cells/well. Isolated splenocytes were added at a ratio of 1:10. After co-culturing for 48 h, cells and supernatants were collected and dead MC38 cells were stained with Zombie NIR™ Fixable Viability Kit (Biolegend) and analyzed via flow cytometry.

## Oncolytic virus generation and preparation

For the generation of oncolytic adenoviruses a cloning system was used which has been initially developed by Mizuguchi and Kay (Mizuguchi and Kay, 1998). The oncolytic adenovirus Ad5/11 and MATE-expressing variants were generated based on the adenoviral vector Ad-iREP2 and the corresponding shuttle plasmid which have been described previously (Gurlevik et al, 2009). Ad-iREP2 contains a ΔN4-25 variant of E1A and reduced replication in normal cells is accomplished by p53-dependent expression of a shRNA network targeting essential adenoviral mRNAs. First, the wt-E1A Promoter in pShuttle-iREP2 was replaced by a short CMV promoter of 245 bp in length by integration between the I-Ceu I and Xho-I sites upstream of ΔN4-25 E1A. The required fragment was amplified using the primers CMV-fw, CMV rev, and the plasmid pBK-CMV (Stratagene) as template. Using the oligonucleotides Swa-fw and Swa-rev, a fragment was inserted into the Mlu I site downstream of the E1B55k-IRES-locus encoding a Swa I restriction site for subsequent insertion of heterologous genes resulting in the plasmid pShuttle-sCMV. The chimeric backbone Ad5/11-iREP2 was generated by exchanging the Ad5 wild type sequence of E4 by a corresponding sequence harboring a serotype 11 derived insertion resulting in a chimeric E4orf4 protein and a functional deletion of E4orf3. The exchange was performed by PCR using the primers E4-fw and E4-rev, and the adenoviral plasmid NW17991 as template. The resulting fragment was integrated into a single Spe I site and the Pac I site located 3' of the rITR by Gibson assembly to replace the wild-type sequence. To finally generate the plasmid pAd5/11, the E1A/E1B regulatory region was exchanged by the corresponding sequence of pShuttle-sCMV using I-Ceu I and PI-Sce I. For the generation of Ad5/11-αCD3$_{TAT}$ and Ad5/11-αCD3$_{TAT}$-Trimer the MATE-encoding cDNAs were linked to the E1B locus via an IRES motif using the Swa I site downstream of IRES. Further Ad5/11 variants for enhanced expression of MATEs were generated by integrating a separate expression cassette under transcriptional control of a phosphoglycerate kinase (PGK) promoter into the Mlu I site downstream of IRES. The required PGK promoter sequence was amplified from the plasmid HN60-pPGK-SB13 (kindly provided by D. Largaespada, Univ. of Minnesota) using the primers PGK-fw and PGK-rev. The established Ad5-based oncolytic hTertAd for comparative purposes has been previously described (Wirth et al, 2003). The generated viruses were produced in HEK293 cells after release of the viral genomes by a Pac I digestion. After large scale propagation, viral particles were purified by CsCl density gradient centrifugation. Obtained viral particles were stored at −20 °C in a glycerol containing storage buffer (glycerol 50% v/v; mouse serum albumin 0.1% m/v; Tris pH 8.0; 10 mM; NaCl 100 mM). Infectious titers were quantified by hexon staining with the AdenoX™ Rapid Titer Kit (Takara) according to the manufacturer's instructions. For in vivo experiments viruses were dialyzed against a buffer containing Tris pH 8.0, 20 mM; NaCl 25 mM; MgCl$_2$ 1.25 mM. Viruses were aliquoted and stored at −80 °C for a maximum of 6 months.

## Replication and cell lysis assay

For replication analysis, cells were seeded in 6 cm dishes ($1.5 \times 10^6$ cells). Cells were cultured overnight and infected with different viruses using an MOI of 2. At indicated time points, cells were harvested and stored in dialysis buffer at −20 °C. Infectious viral particles were quantified via AdenoX™ Rapid Titer Kit (Takara).

Cell lysis assays were performed using MTT-assay. Cells were seeded in 96-well plates and infected at an MOI as shown in the figure. At the indicated time points, MTT reagent (5 mg/ml) diluted 1:11 in glucose-containing Krebs-Ringer buffer was applied to the cells and incubated for 30–45 min at 37 °C. Subsequently, the converted dye formazan was dissolved with DMSO and photometrically analyzed at a wavelength of 562 nm.

## Animal experimentation

For the in vivo experiments, 6–8-week-old male immunocompetent C57BL/6-J mice (bred at the Central Animal Facility, Hannover Medical School, Germany) were used. Experimental animals were fed a standard diet (1324 M, Altromin, Lage, Germany) with access to food and water ad libitum. Interventions were done during the 14 h-day cycle. Subcutaneous (s.c.) tumors were induced by injection of $1 \times 10^7$ MC38 cells in a total volume of 100 µl PBS into the right flank of mice. Virotherapy was applied by single intratumoral (i.t.) injection of $1 \times 10^9$ infectious particles (in a total volume of 50 µl 0.9% NaCl solution) of Ad5/11 or MATE-expressing viruses. Controls received an injection of 50 µl 0.9% NaCl solution. Anti-PD-1 immune checkpoint inhibition was carried out 1 day after virotherapy by intraperitoneal (i.p.) injection of a αPD-1 antibody (BioXCell; clone: RMP1-14; 75 µg in 0.9% NaCl solution). The application of αPD-1 was repeated twice a week as indicated in the figures. In vivo depletion of CD8$^+$ T cells was performed by i.p. application of 75 mg/mouse of anti-mouse CD8a GoInVivo™ clone 53-6.7 (Biolegend). Depleting antibodies were applied four times with the first dose administered prior to virotherapy. All in vivo experiments were performed according to the German legal guidelines for animal care and experimentation (TierSchG) and in accordance with the guidelines of the European Parliament and the council on the protection of animals used for scientific purposes (2010/63/EU). The experimental design has been approved by the review board of Hannover Medical School and by the regional governmental board (LAVES).

## Flow cytometric analysis of immune cells

Flow cytometry analyses and peptide stimulation was performed using single-cell suspensions of tumor tissue. These cell suspensions were obtained by digestion of tumors in RPMI+GlutaMAX™ containing DNAse (60 µg/ml; Sigma), hyaluronidase (0.2 mg/ml; Sigma), collagenase IA (0.2 mg/ml; Sigma) and collagenase IV (0.2 mg/ml; Sigma) at 37 °C for 30 min. After digestion, the cell suspension was passed through a nylon cell strainer (40 µm) to obtain a single-cell suspension and cells were washed with RPMI +GlutaMAX™. Subsequently, cells were subjected to peptide stimulation experiments or directly analyzed via flow cytometry using a FACS Canto II (BD). For flow cytometry, cells were stained with the following conjugated antibodies (1:100 dilution): CD90.2-PerCP (Biolegend, clone: 53-2.1), CD45.2-PerCP (Biolegend, clone: 104), CD8a-FITC (Biolegend, clone: 53–6.7), CD8a-APC/Cyanine7 (Biolegend, clone: 53–6.7), CD8a-PE (Biolegend, clone: 53–6.7), CD4-PE (Biolegend, clone: GK1.5), CD4-FITC (Biolegend, clone: GK1.5), CD11c-APC/Cyanine7 (Biolegend, clone: N418), CD11c-APC (Biolegend, clone: N418), F4/80-FITC (Biolegend, clone: BM8), F4/80-APC/Cyanine7 (Biolegend, clone: BM8), IFNγ-APC (Biolegend, clone: XMG1.2), NK1.1-PE

(Biolegend, clone: PK136), CD69-PE (Biolegend, clone: H1.2F3), CD69-APC (Biolegend, clone: H1.2F3), CD25-PE (Biolegend, clone: PC61), CD25-APC (Biolegend, clone: PC61), Ki67-PE (Invitrogen, clone: SolA), CD3-APC/Cyanine7 (Biolegend, clone: 17A2), CD44-APC (Biolegend, clone: IM7), CD44-FITC (Biolegend, clone: IM7), CD49b-APC (Biolegend, clone: DX5), MHCII (I-A/I-E)-PE (Biolegend, clone: M5/114.15.2), XCRI-APC (Biolegend, clone: ZET), TruStain FcX (CD16/32)-unconjugated (Biolegend, clone: 93), c-myc-unconjugated (Santa Cruz Biotechnology; clone: 9E10), IgG1-PE (Biolegend, clone: RMG1-1), FoxP3-APC (eBioscience; clone: FJK-16s; 1,25:100), and TNFα-PE (Biolegend, clone: MP6-XT22). Human PBMCs were stained with the following conjugated antibodies (1:20 dilution): CD4-PerCP (Biolegend, clone: OKT4), CD4-PE (Biolegend, clone: OKT4), CD8-APC (Biolegend, clone: HIT8a), CD8-FITC (Biolegend, clone: HIT8a), CD8-APC/Cyanine7 (Biolegend, clone: HIT8a), CD69-PE (Biolegend, clone: FN50), CD137-PE (Biolegend, clone: 4B4-1), CD137-APC (Biolegend, clone: 4B4-1), IFNγ-APC (Biolegend, clone: B27), and TNFα-PE (Biolegend, clone: Mab11). For identification of tumor-specific T-cell responses intracellular IFNγ staining was performed after incubation (16 h, 37 °C, 5% $CO_2$) of immune cells with a MC38 mutated neoantigen adpgk-R304M (obtained from GenScript). Addition of GolgiPlug™ (BD, 1:1000) inhibited the secretion of the produced cytokines. Intracellular staining of IFNγ and TNFα T cells was performed using the BD CytoFix/CytoPerm Kit. Intranuclear staining of Ki-67 or FoxP3 was performed using eBioscience™ Foxp3/Transcription Factor Staining Kit. Flow cytometry data were analyzed using FlowJo 10. Exemplary dot plots reflecting the used gating strategies are displayed in the Figures and also in the Appendix Figure S1.

## Luminex multiplex assay

For Bio-Plex multiplex immunoassay analysis of cytokines within tumor tissue, the Bio-Plex™ Cell Lysis Kit (BioRad) was used. For this purpose, a tumor piece with a size of max. 3 mm was shredded in 200 µl cell lysis buffer. The cells were disrupted by one freeze-thaw cycle at −80 °C and subsequent ultrasonication. After an additional freeze-thaw cycle, the cell debris was separated by centrifugation and the amount of protein in the supernatant was determined using the Pierce™ BCA Protein Assay Kit (Thermo-Fisher). Analysis of the intratumoral cytokine composition was investigated with the Bio-Plex Multiplex immunoassay system (BioRad). Tumor lysates were processed and measured using the Bio-Plex Pro Mouse Cytokine 23-plex kit and Bio-Plex Reader. All steps were performed according to the manufacturer's instructions.

## Cytokine release syndrome assay

Serum cytokine levels were analyzed with the LEGENDplex™ Multi-Analyte Flow Assay Kit (Biolegend), which allows quantification of 13 different cytokines that are associated with Cytokine Release Syndrome. Serum samples were obtained from the vena cava of sacrificed mice. Prior to analysis blood coagulation was timed for 30 min at room temperature and serum supernatant was collected after a centrifugation step at $4000 \times g$. Quantification of serum cytokines was performed according to the manufacturer's instructions. All samples were diluted 2-fold with Assay buffer and measured in duplicates using a FACS Canto II (BD). For statistical analysis, LEGENDplex™ Data Analysis Software Suite was used.

## Alanine transaminase (ALT) activity assay

The activity of ALT was measured in serum samples using a colorimetric assay kit (RayBiotech). Serum was collected as described for the LEGENDplex™ assay. Undiluted samples were processed following the manufacturer's instructions and measured in duplicates. The absorbance decrease was photometrically assessed at 340 nm using a MQX200 Microplate Reader (BioTek). Blank-corrected rates were compared to an extinction coefficient of 1.836 mM$^{-1}$ to calculate the ALT activity in U/L.

## Histology, immunohistochemistry, and multiplex immunohistochemistry

Tumor tissue was harvested at time points indicated in the figure legends. To obtain FFPE material, tissue samples were fixed using 4% paraformaldehyde (PFA) and then paraffine-embedded. 7 µm sections were performed and stained with hematoxylin/eosin (H/E) for histologic examination. For immunohistochemistry and opal multiplex immunohistochemistry the following rabbit-derived primary antibodies from Cell Signaling Technology® were used: CD3 (CST99940; 1:200), CD4 (CST25229; 1:200), CD8 (CST98941; 1:400), CD161c (CST39197; 1:1000), F4/80 (CST70076; 1:600), PanCadherin (CST4068; 1:1000), and PD-1 (CST84651; 1:100). For diaminobenzidine (DAB) staining, the primary antibody staining was followed by an incubation step with HRP-coupled goat anti-rabbit IgG secondary antibody (Abcam; 1:2000). Subsequently, tumor sections were incubated with DAB solution (Vector Laboratories) and nuclei were stained with hematoxylin. For immunofluorescence stainings of PD-1, the conjugated secondary antibodies Alexa Fluor® 488 goat anti-rabbit IgG (H + L) (Invitrogen; 1:500) was used in combination with Hoechst 33258 staining (0.5 µg/µl) of nuclei. For double-staining of E1A (adenovirus detection) and c-myc (MATE detection), the following antibodies were used: primary E1A BSS-BS-6136R (Bioss, 1:500) combined with a conjugated Alexa Fluor® 488 goat anti-rabbit IgG (H + L) (Invitrogen, 1:500), primary myc AB 25355826 (Invitrogen, 1:250) with conjugated secondary antibody Alexa Fluor® 568 goat anti-chicken IgG (H + L) (Invitrogen, 1:250). Images were acquired using the 40x objective of Nikon Eclipse Ti2 microscope at ex:340-380 nm/em:435-485 for DAPI, at ex:465-495/em:515-555 for Alexa Fluor® 488, at ex:528-553/em:590-650 for Alexa Fluor® 568.

Akoya multiplex staining was performed according to the manufacturer instructions (Opal™ 4-Color Anti-Rabbit Automation IHC; Opal™ Polymer Anti-Rabbit HRP Secondary Antibody Kit; Akoya Biosciences®). To facilitate sequential staining of five targets additional Opal fluorescent dyes 620 and 780 were purchased as extension of the Opal™ 4-Color Anti-Rabbit Automation IHC Kit. Slides were scanned using the 'Vectra Polaris' (Akoya Biosciences®) platform. Spectral unmixing and further tissue and cell segmentation, as well as phenotyping, were completed using the inForm Advanced Image Analysis software (inform v2.4.10, Akoya Biosciences®). Resulting data was further processed with RStudio (Rstudio v1.1.456) using R- packages 'Phenoptr' and 'PhenoptrReports'. Extracted data sets were finally summarized with Microsoft Excel.

## The paper explained

### Problem

Oncolytic viruses expressing Bispecific-T-cell engagers (BiTEs) for intratumoral T-cell activation are promising tools for immunotherapy of solid tumors. Because BiTEs require the presence of a defined target antigen in tumors to exert their effect, a corresponding oncolytic virus is only applicable in a limited number of tumors. T-cell activating molecules capable of promiscuous T-cell activation in tumors in a target-independent manner are urgently needed to facilitate broad applicability of oncolytic viruses with T cell activator functions.

### Results

We have generated membrane-associated T-cell engager molecules (MATEs) consisting of a CD3 binding domain as well as a Tat-PTD motif to facilitate their attachment to the surface of cells independent of the presence of a defined tumor target antigen. We showed that MATEs are capable of binding to murine and human T cells leading to their effective activation in vitro and that their expression can confer T-cell mediated cytotoxicity to neighboring cells. Intratumoral infection of syngeneic MC38 tumors in mice by MATE expressing oncolytic adenoviruses resulted in T-cell activation, tumor immunoactivation, and therapeutic benefit. Adverse events such as liver toxicity or a cytokine release syndrome were not observed despite target-independent T-cell activating properties of MATEs indicating safe applicability of MATE expressing oncolytic adenoviruses. It could be furthermore demonstrated in mice that intratumoral MATE expression by oncolytic viruses effectively sensitized tumors for PD-1 checkpoint inhibition leading to promising antitumoral effects including long-term survival.

### Impact

In this study, we have developed novel oncolytic adenoviruses for the intratumoral delivery of MATEs to achieve strong T-cell activation not requiring the presence of a specific tumor antigen. Expression of MATEs by oncolytic adenoviruses is a promising strategy to equip oncolytic agents with direct and effective T-cell activating properties thereby not restricting their broad applicability. Due to their tumor immunoactivating properties, MATE-expressing oncolytic viruses also represent effective tools to prepare insensitive tumors for the application of PD1/PD-L1 checkpoint inhibitors.

## Ethics statement

All experiments involving human material were performed with permission of the ethics committee of Hannover Medical School. Informed consent was obtained from all human subjects. It is confirmed that the experiments conformed to the principles set out in the WMA Declaration of Helsinki and the Department of Health and Human Services Belmont Report.

## Statistical analysis

Mice were randomly grouped into treatment and control groups. It was taken care that tumor sizes were evenly distributed among the groups. Full blinding was not possible due to legal requirements allowing the identification of individual mice by legal authorities at any time. A partial blinding was realized by numbering the mice instead of group labeling until analyses had been finalized. Statistical analysis was performed with the software GraphPad Prism 10. The required sample size was determined by a priori power analysis and by negotiations with the responsible legal authorities. The sample size is indicated in the Figure legends. Single values were excluded from analysis if they were both beyond the expected range and also not consistent with other related parameters within the same panel of analyses. Exclusion from statistical analysis has been annotated in the figure legend. Statistical significance between two groups was determined using the unpaired student's $t$-test, while for group sizes >2 one-way ANOVA with Tukey's post hoc analysis test was used. Gaussian distribution was verified by a Shapiro–Wilk test. In case of non-gaussian distribution a Mann–Whitney Test was applied. Survival curves were analyzed using log-rank Mantel–Cox test. $P$-values < 0.05 were considered statistically significant.

## Data availability

This study includes no data deposited in external repositories.

The source data of this paper are collected in the following database record: biostudies:S-SCDT-10_1038-S44321-024-00187-y.

## Peer review information

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

## Acknowledgements

FK was supported by Deutsche Krebshilfe (German Cancer Aid) (project 70113873), Deutsche Forschungsgemeinschaft (DFG) (project KU2682/6-1), Bundesministerium für Bildung und Forschung (Federal Ministry of Education and Research) (project 16LW0220), and the Claudia-von-Schilling-Stiftung.

## Author contributions

**Malin Peter**: Conceptualization; Formal analysis; Investigation; Visualization; Writing—original draft. **Bettina Mundt**: Investigation; Visualization; Methodology; Project administration; Writing—review and editing. **Arne Menze**: Formal analysis; Investigation; Visualization; Writing—review and editing. **Norman Woller**: Resources; Writing—review and editing. **Valery Volk**: Formal analysis; Methodology. **Amanda M Ernst**: Resources; Writing—review and editing. **Leon A Öhler**: Resources. **Steven R Talbot**: Formal analysis; Writing—review and editing. **Heiner Wedemeyer**: Resources; Writing—review and editing. **Christine Falk**: Resources; Methodology; Writing—review and editing. **Friedrich Feuerhake**: Resources; Methodology; Writing—review and editing. **Thomas C Wirth**: Conceptualization; Writing—review and editing. **Florian Kühnel**: Conceptualization; Supervision; Writing—original draft; Project administration; Writing—review and editing.

Source data underlying figure panels in this paper may have individual authorship assigned. Where available, figure panel/source data authorship is listed in the following database record: biostudies:S-SCDT-10_1038-S44321-024-00187-y.

## Funding

## Disclosure and competing interests statement

Thomas C. Wirth, Malin Peter, Norman Woller, and Florian Kühnel have filed patent applications describing the virus Ad5/11 (WO2021239586-A1) and MATE technology (EP23162746.4).

# Expanded View Figures

**Figure EV1. Membrane-associated T-cell engager (MATE) proteins activate different murine CD8$^+$ T cells subsets and confer cytotoxicity against tumor cells in vitro.** ▶

(**A**) SDS-PAGE and Western-blot analysis of Ni-NTA-purified MATEs using an α-myc antibody is shown in the left panel, a non-denaturing Blue-native PAGE and subsequent western blot analysis is shown on the right hand side. Standards have been stained with Coomassie. (**B**) Murine splenocytes were incubated with 500 ng of purified MATEs and binding to CD8$^+$ T-cell subsets, as defined by CD44 and CD62L markers, was analyzed by flow cytometry. (**C**) Murine splenocytes were incubated with increasing amounts of purified MATEs and T-cell activation was measured (number of replicates $n = 3$; mean ± SD; $p$ values by one-way ANOVA with Tukey's post hoc analysis). (**D**) The polarization of CD4$^+$ T cells to regulatory T cells (Tregs) in the course of MATE-dependent activation was investigated. The amount of Tregs was analyzed by the intranuclear staining of FoxP3 in CD4$^+$ CD25$^+$ T cells after 48 h of incubation with 1 μg of purified MATEs. (number of replicates $n = 3$; mean ± SD; $p$ values by one-way ANOVA with Tukey's post hoc analysis). (**E**) Expression analysis of transgenic MC38 cells stably expressing a transmembrane T-cell engager (αCD3$_{TM}$) using flow cytometry. (**F**) Western blot analysis of supernatants obtained from MC38 cells stably expressing MATEs. (**G**) T-cell activation of murine CD4$^+$ and CD8$^+$ T cells 24 h after coculture with transmembrane T-cell engager-expressing, MATE-expressing or non-expressing MC38 tumor cells (1:10 target:splenocyte) (number of replicates $n = 3$; mean ± SD; $p$ values by one-way ANOVA with Tukey's post hoc analysis). (**H**) Proliferation analysis of CD4$^+$ and CD8$^+$ T cells after 48 h coculture (1:10 target:splenocyte) with transgenic MC38 cells expressing either MATEs or the transmembrane T-cell engager (number of replicates $n = 3$; mean ± SD; $p$ values by one-way ANOVA with Tukey's post hoc analysis).

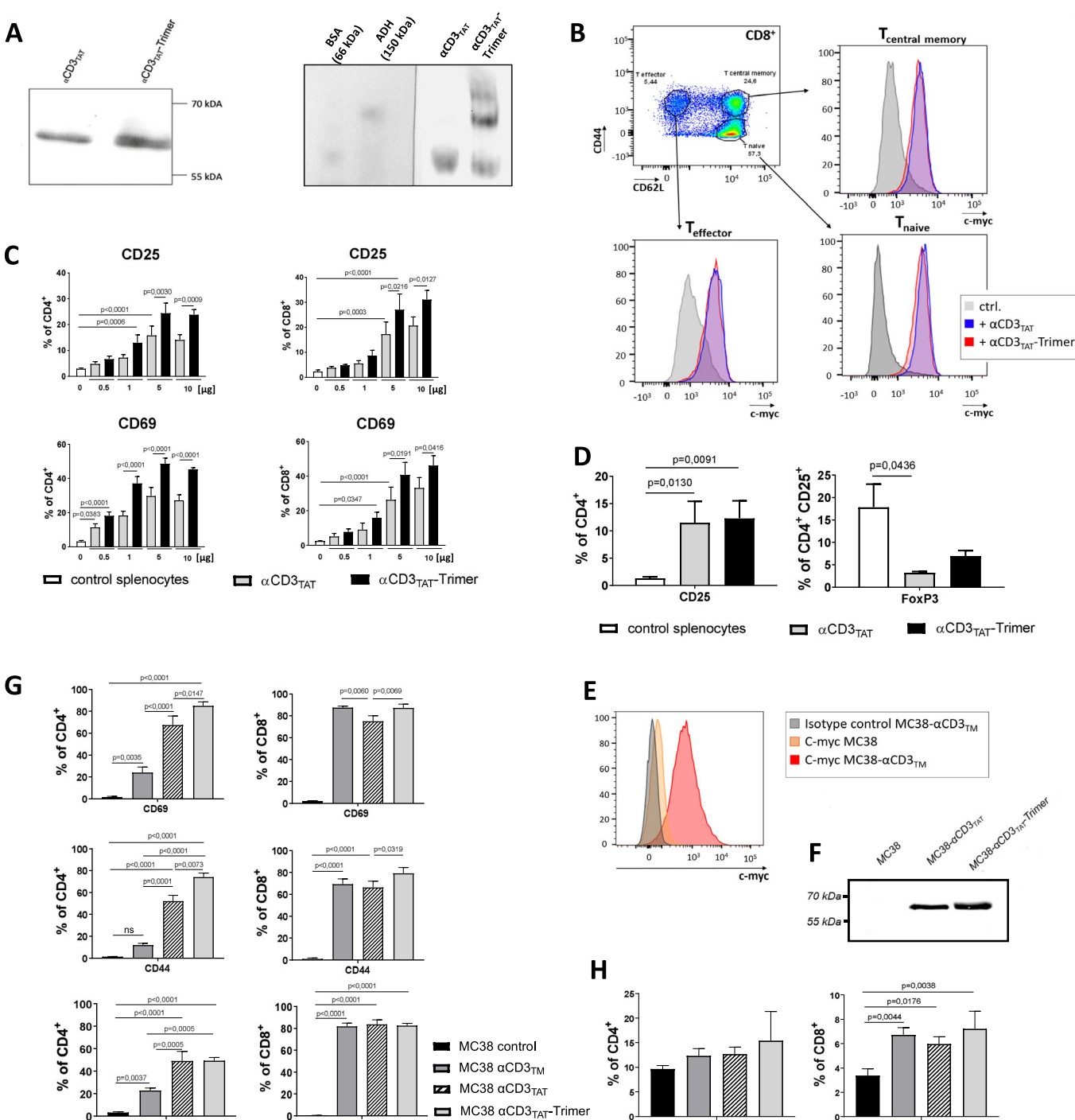

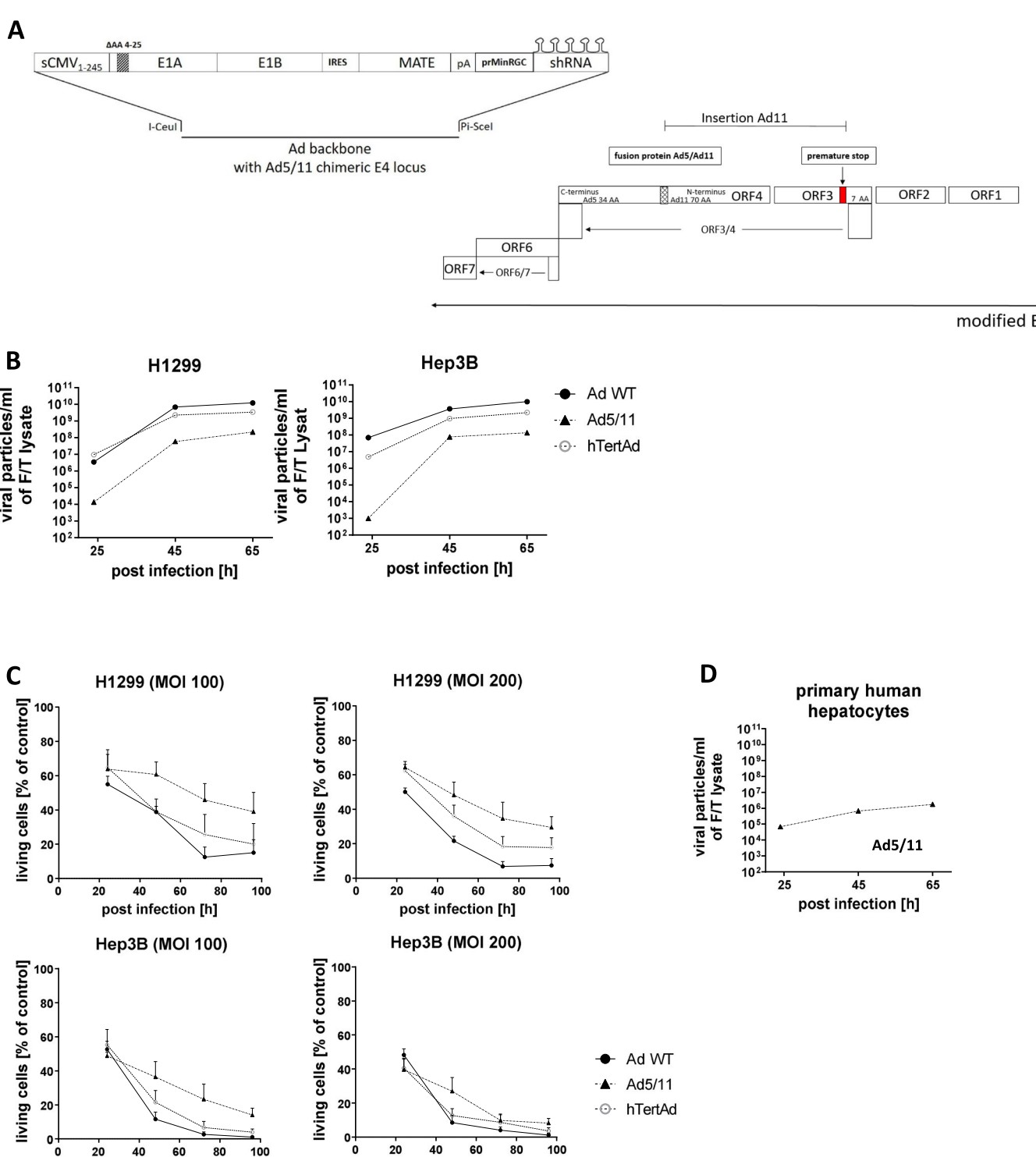

**Figure EV2. The oncolytic adenovirus Ad5/11 facilitates replication in and lysis of various human tumor cell lines but is inhibited in normal hepatocytes.**

(A) Schematic illustration of MATE-expressing Ad5/11 highlighting the regulatory E1 region including the locus for MATE expression (left) and the insertion of adenovirus serotype 11 derived sequences in the E4orf3/orf4 locus (right). (B) Replication kinetics of Ad5/11 compared with the Ad5-based oncolytic adenovirus hTertAd and adenovirus wild type (Ad WT) following infection with MOI of 2 in human tumor cell lines Hep3B and H1299. Cells were harvested at time points indicated and infectious viral particles were determined by using Rapid Titer Assay. (C) Human tumor cell lines Hep3B and H1299 were infected with Ad5/11, the oncolytic adenovirus hTertAd and adenovirus wild type (Ad WT) at the MOI indicated in the panels. 48 h after infection cell lysis was determined by MTT assays (number of replicates $n = 3$; mean ± SD). (D) Replication kinetics of Ad5/11 after infection of primary human hepatocytes with an MOI of 2. Infectious viral particles were determined using the Rapid Titer Assay.

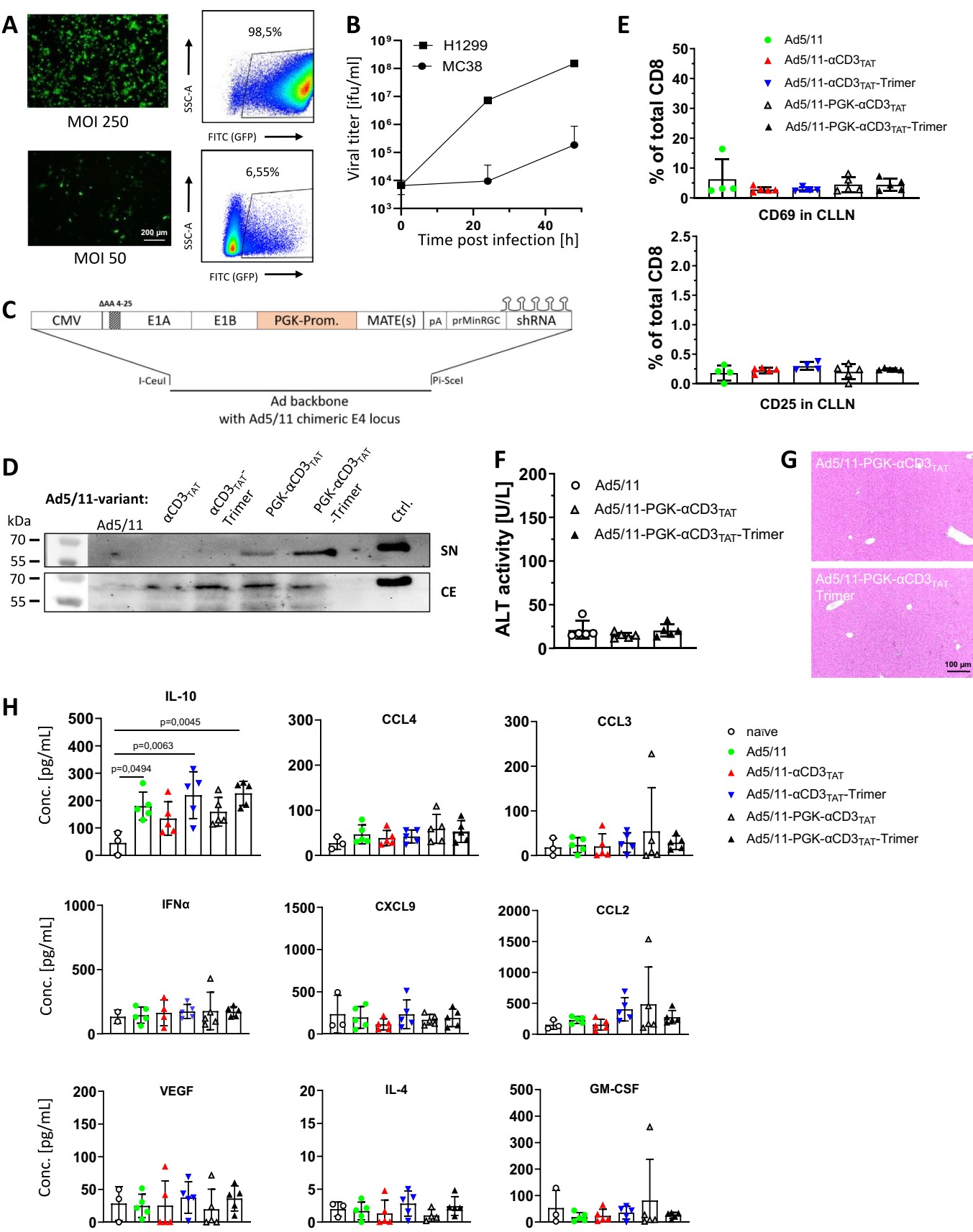

◀  **Figure EV3.  Intratumoral virotherapy with enhanced MATE-expression does not lead to systemic toxicity.**

(A) To assess susceptibility of MC38 cells for MATE-expressing adenoviruses, cells were treated in vitro with a replication-deficient adenoviral serotype 5 based vector expressing a GFP-reporter gene using a MOI as indicated. 48 h following infection, GFP-expression was monitored by fluorescence microscopy and quantified by flow cytometry. Representative images are shown (magnification 100×; scale bar indicates 200 μm). (B) Human H1299 cells and murine MC38 cells were infected with MOI 50 of Ad5/11. After 4 h excess virus was removed and generated virus particles were counted by Rapid Titer assay at the time points indicated ($n = 3$, mean ± SD). (C) Illustration showing the genetic setup of Ad5/11 variants harboring MATEs in a separate expression cassette under transcriptional control of a phosphoglycerate kinase (PGK) promoter for enhanced MATE expression levels. (D) MC38 cells were infected with Ad5/11 or MATE-expressing Ad5/11 variants and supernatants were collected after 48 h. Cell extracts were prepared using RIPA-buffer. MATEs in the supernatant were further enriched by Ni-NTA purification. MATEs were analyzed by western blot analyses using an α-myc antibody. (E) S.c. MC38 tumors in C57BL/6 mice were treated with MATE-expressing Ad5/11 variants as shown in the figure and as described in Fig. 4. T-cell activation was determined in the contralateral lymph nodes (CLLN) after 48 h ($n = 4$ mice for Ad5/11 and Ad5/11-αCD3$_{TAT}$-Trimer, $n = 5$ mice for all other groups; mean ± SD) and ALT-activity of the PGK-variants and control virus is shown to complement Fig. 4C (F) ($n = 5$ mice; mean ± SD). (G) Representative images of H/E stained FFPE liver tissue of mice treated with PGK-variants (magnification 100×; scale bar indicates 100 μm; representative images from each group are shown). (H) Additional measurements supplementing main Fig. 4 showing the analysis of serum cytokine levels of treated mice using the Legendplex mouse cytokine release syndrome panel ($n = 3$ mice for naïve controls, $n = 5$ mice for all other groups; mean ± SD; $p$ values by one-way ANOVA with Tukey's post hoc analysis, in case of IFNα two outliers were not included in the panel and no statistical analysis was performed).

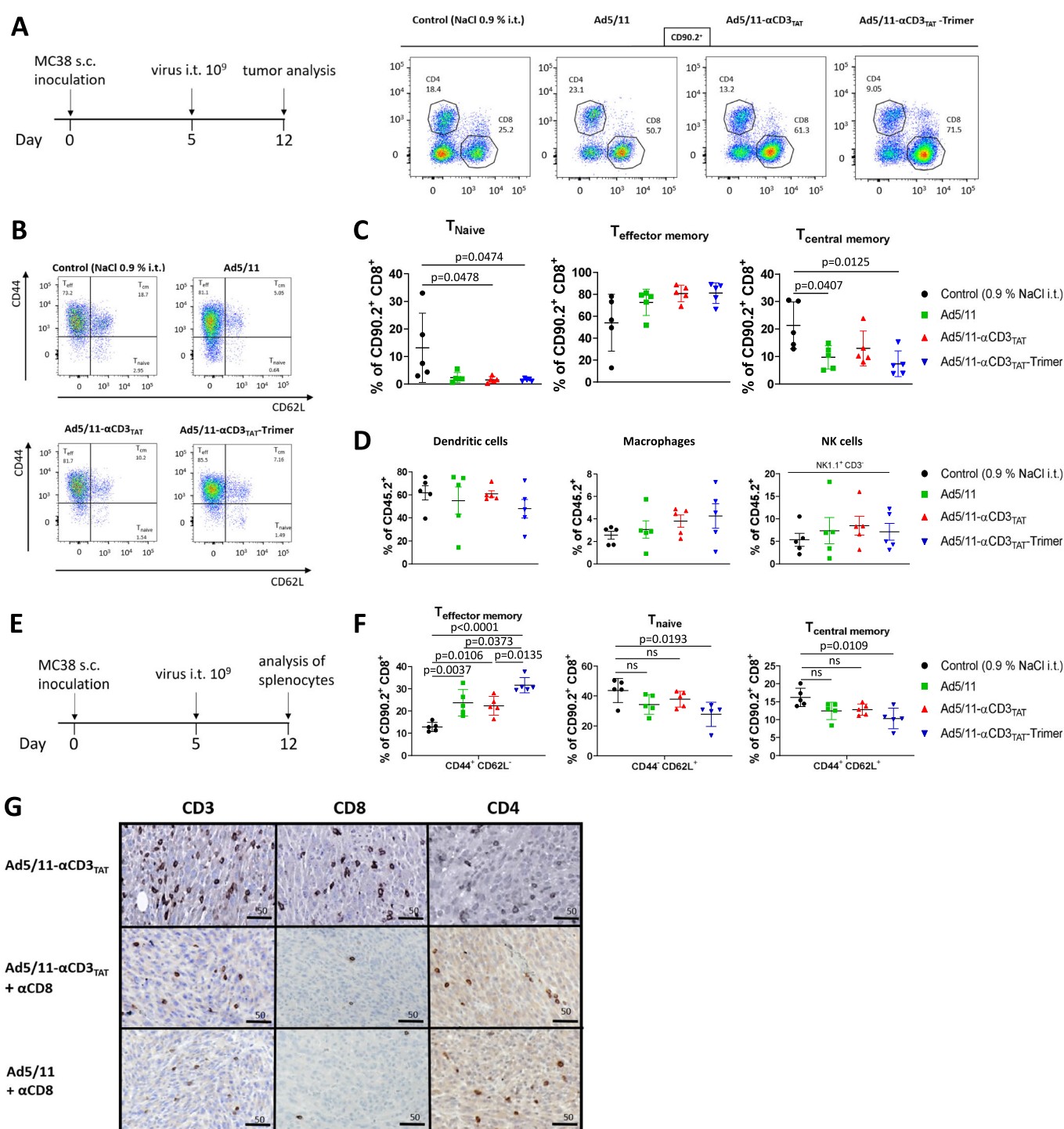

**Figure EV4. Intratumoral virotherapy with MATE-expressing Ad5/11 supports intratumoral and systemic immunoactivation.**

(A–D) S.c. MC38 tumors were established and treated with MATE-expressing Ad5/11 variants as described in the scheme. Mice were sacrificed at day seven after intratumoral virotherapy and leukocyte infiltration in tumor tissue was examined by flow cytometry. (A) Representative FACS plots showing CD4+ and CD8+ T-cell populations. (B) Representative FACS plots illustrating the subsequent characterization of CD8+ T-cell subsets. (C) T-cell subsets were calculated as percentage of CD90.2+ CD8+ lymphocytes ($n = 5$ mice; mean ± SD; $p$ values calculated by one-way ANOVA with Tukey's post hoc analysis). (D) Abundance of dendritic cells (CD11c+, F4/80−), macrophages (F4/80+) and NK cells (NK1.1+, CD3−) within the CD45.2+ leukocyte population ($n = 5$ mice; mean ± SD). After tumor treatment as illustrated in (E), splenocytes of treated mice were subjected to immune analyses. (F) Shows the characterization of CD8+ T-cell subsets within the CD90.2+ CD8+ lymphocyte population ($n = 5$ mice; mean ± SD; $p$ values by one-way ANOVA with Tukey's post hoc analysis). (G) MC38 tumor-bearing mice were treated as described in Fig. 6A. Immunohistochemical staining of T-cell populations within tumor tissue following systemic application of a CD8+ depleting antibody was performed to confirm the success of CD8 T cell depletion (Magnification 200×, bars indicate 50 μm, Images showing CD3 and CD8 staining in the Ad5/11-αCD3TAT group have been reused from Fig. 5C).

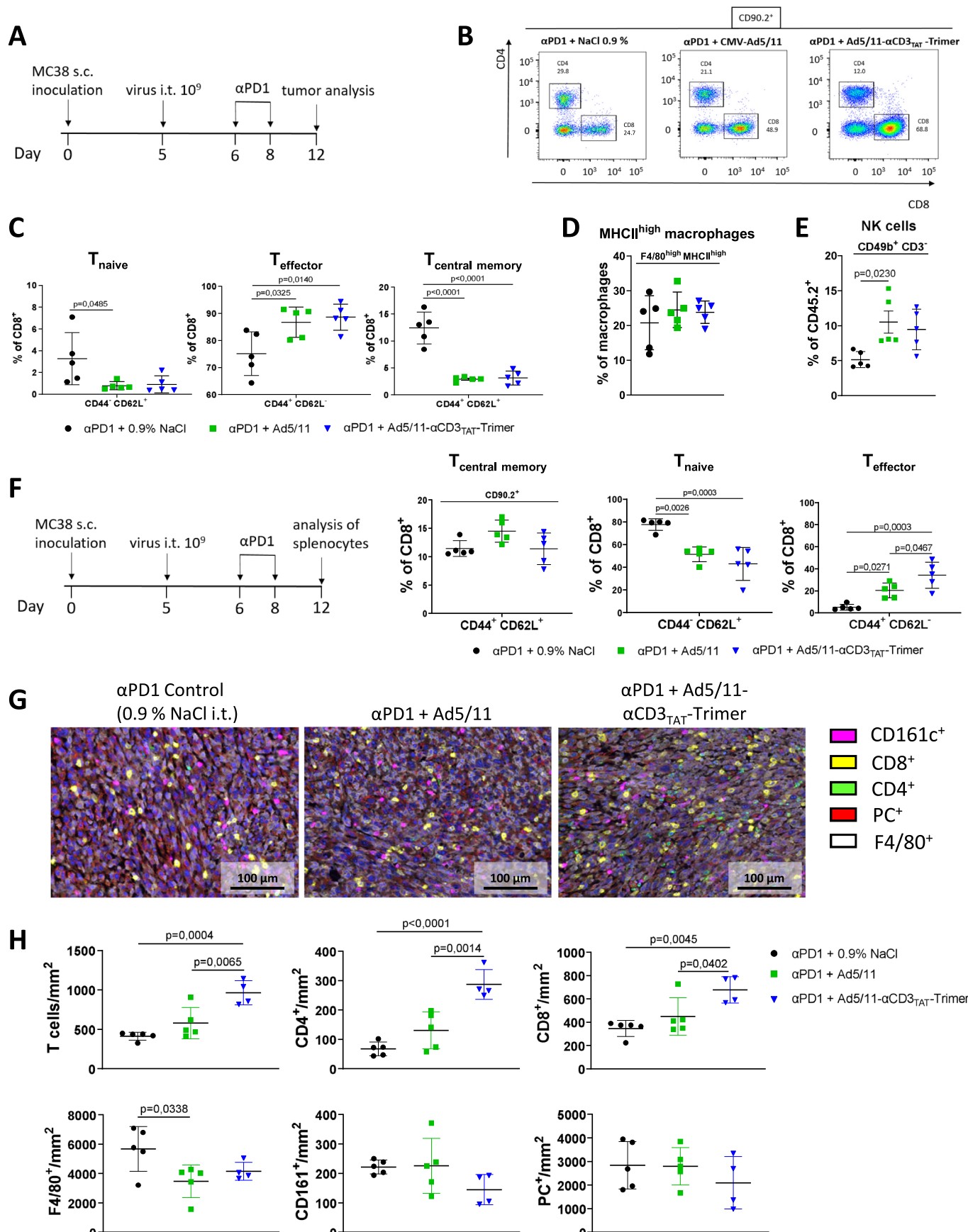

◀

**Figure EV5. Combination of MATE-expressing virotherapy and PD1 checkpoint blockade modulates the tumor microenvironment and promotes intratumoral T-cell infiltration.**

(A) S.c. MC38 tumors were established and treated with virotherapy followed by two i.p. applications of αPD1 antibodies as illustrated. Seven days after virotherapy, tumor tissue was investigated via flow cytometry to determine CD4$^+$ and CD8$^+$ T-cell infiltration (B), to characterize CD8$^+$ T-cell subsets (C) and to determine the frequency of further myeloid immune cells (D, E). (C) T-cell subsets are given as frequency within the CD8$^+$ T-cell population ($n = 5$ mice; mean ± SD; p values by one-way ANOVA with Tukey's post hoc analysis). (D) shows the amount of the M1-subtype (MHCII$^{high}$) within the macrophage population (CD45.2$^+$, F4/80$^+$), Frequency of NK cells (CD49b$^+$, CD3$^-$) was calculated within the CD45.2$^+$ leukocyte population (E) ($n = 5$ mice; mean ± SD; p values by one-way ANOVA with Tukey's post hoc analysis). (F) Spleens of treated mice were prepared to analyze CD8$^+$ T-cell subsets ($n = 5$ mice; mean ± SD; p values by one-way ANOVA with Tukey's post hoc analysis). (G) As described for main Fig. 8, MC38 tumor tissue was obtained from tumor-bearing mice 7 days after intratumoral virotherapy followed by two applications of αPD1 antibodies. Spatial immune cell phenotyping was performed by multiplex immunohistochemical staining of CD8$^+$ T cells (yellow), CD4$^+$ T cells (green), CD161c$^+$ NK cells (magenta), F4/80$^+$ macrophages (white), pan-Cadherin$^+$ (PC$^+$) tumor cells (red), and DAPI staining for cell nuclei (blue). Additional representative composite images from each group are shown (G). The densities of the investigated cell types as numbers per area are shown in (H) ($n = 4$ mice for Ad5/11-αCD3$_{TAT}$-Trimer + αPD1, all other groups $n = 5$ mice; mean ± SD; p values by one-way ANOVA with Tukey's post hoc analysis).

