## [Peer Review File · EMBO Molecular Medicine]

Oncolytic viruses expressing MATEs facilitate target-independent T-cell activation in tumors

Malin Peter, Bettina Mundt, Arne Menze, Norman Woller, Valery Volk, Amanda Ernst, Leon Öhler, Steven Talbot, Heiner Wedemeyer, Christine Falk, Friedrich Feuerhake, Thomas Wirth, and Florian Kuehnel

Corresponding author: Florian Kuehnel (kuehnel.florian@mh-hannover.de)

Review Timeline:

Submission Date:	16th Feb 24
Editorial Decision:	14th Mar 24
Revision Received:	1st Oct 24
Editorial Decision:	22nd Oct 24
Revision Received:	28th Nov 24
Editorial Decision:	4th Dec 24
Revision Received:	12th Dec 24
Accepted:	16th Dec 24

Editor: Lise Roth

Transaction Report:

14th Mar 2024

Dear Prof. Kuehnel,

Thank you for the submission of your manuscript to EMBO Molecular Medicine. We have now received feedback from the three reviewers who agreed to evaluate your manuscript. As you will see from the reports below, the referees acknowledge the interest of the study and are overall supporting publication of your work pending appropriate revisions.

In their reports, the referees raise several major concerns related to potential toxicity of MATE, limited efficacy, and questions on the trimeric vs. monomeric form. We further discussed these concerns with the referees and agreed that additional in vivo models might be beyond the scope of the study. Therefore, we would like you to use in vitro systems to provide more in-depth analysis of the expression and secretion of the MATEs in different cell cultures (mouse and human), their toxicity (comparing monomers and trimers) and bystander killing of tumor cells. It will also be important to perform staining of MATE and virus in the studied tumors. Please also provide data on the permissiveness levels of the MC38 model (how much replication and MATE secretion compared to a fully permissive human tumor).

Adequately addressing these and other reviewers' concerns will be necessary for further considering the manuscript in our journal, and acceptance of the manuscript will entail a second round of review. EMBO Molecular Medicine encourages a single round of revision only and therefore, acceptance or rejection of the manuscript will depend on the completeness of your responses included in the next, final version of the manuscript. For this reason, and to save you from any frustrations in the end, I would strongly advise against returning an incomplete revision.

We are expecting your revised manuscript within three months, if you anticipate any delay, please contact us.

We require:

- 1) A .docx formatted version of the manuscript text (including legends for main figures, EV figures and tables). Please make sure that the changes are highlighted to be clearly visible.
- 2) Individual production quality figure files as .eps, .tif, .jpg (one file per figure). For guidance, download the 'Figure Guide PDF' (<https://www.embopress.org/page/journal/17574684/authorguide#figureformat>).
- 3) At EMBO Press we ask authors to provide source data for the main figures. Our source data coordinator will contact you to discuss which figure panels we would need source data for and will also provide you with helpful tips on how to upload and organize the files.
- 4) A .docx formatted letter INCLUDING the reviewers' reports and your detailed point-by-point responses to their comments. As part of the EMBO Press transparent editorial process, the point-by-point response is part of the Review Process File (RPF), which will be published alongside your paper.
- 5) A complete author checklist, which you can download from our author guidelines (<https://www.embopress.org/page/journal/17574684/authorguide#submissionofrevisions>). Please insert information in the checklist that is also reflected in the manuscript. The completed author checklist will also be part of the RPF.
- 6) It is mandatory to include a 'Data Availability' section after the Materials and Methods. Before submitting your revision, primary datasets produced in this study need to be deposited in an appropriate public database, and the accession numbers and database listed under 'Data Availability'. Please remember to provide a reviewer password if the datasets are not yet public (see <https://www.embopress.org/page/journal/17574684/authorguide#dataavailability>). In case you have no data that requires deposition in a public database, please state so in this section ("This study includes no data deposited in external repositories."). Note that the Data Availability Section is restricted to new primary data that are part of this study.
- 7) For data quantification: please specify the name of the statistical test used to generate error bars and P values, the number (n) of independent experiments (specify technical or biological replicates) underlying each data point and the test used to calculate p-values in each figure legend. The figure legends should contain a basic description of n, P and the test applied.

Graphs must include a description of the bars and the error bars (s.d., s.e.m.). Please provide exact p values.

8) Our journal encourages inclusion of *data citations in the reference list* to directly cite datasets that were re-used and obtained from public databases. Data citations in the article text are distinct from normal bibliographical citations and should directly link to the database records from which the data can be accessed. In the main text, data citations are formatted as follows: "Data ref: Smith et al, 2001" or "Data ref: NCBI Sequence Read Archive PRJNA342805, 2017". In the Reference list, data citations must be labeled with "[DATASET]". A data reference must provide the database name, accession number/identifiers and a resolvable link to the landing page from which the data can be accessed at the end of the reference. Further instructions are available at .

9) The paper explained: EMBO Molecular Medicine articles are accompanied by a summary of the articles to emphasize the major findings in the paper and their medical implications for the non-specialist reader. Please provide a draft summary of your article highlighting

10) For more information: There is space at the end of each article to list relevant web links for further consultation by our readers. Could you identify some relevant ones and provide such information as well? Some examples are patient associations, relevant databases, OMIM/proteins/genes links, author's websites, etc...

11) Author contributions: CRediT has replaced the traditional author contributions section because it offers a systematic machine readable author contributions format that allows for more effective research assessment. Please remove the Authors Contributions from the manuscript and use the free text boxes beneath each contributing author's name in our system to add specific details on the author's contribution. More information is available in our guide to authors.

12) Every published paper now includes a 'Synopsis' to further enhance discoverability. Synopses are displayed on the journal webpage and are freely accessible to all readers. They include a short stand first (maximum of 300 characters, including space) as well as 2-5 one-sentences bullet points that summarizes the paper. Please write the bullet points to summarize the key NEW findings. They should be designed to be complementary to the abstract - i.e. not repeat the same text. We encourage inclusion of key acronyms and quantitative information (maximum of 30 words / bullet point). Please use the passive voice. Please attach these in a separate file or send them by email, we will incorporate them accordingly.

13) As part of the EMBO Publications transparent editorial process initiative (see our Editorial at <http://embomolmed.embopress.org/content/2/9/329>), EMBO Molecular Medicine will publish online a Review Process File (RPF) to accompany accepted manuscripts.

In the event of acceptance, this file will be published in conjunction with your paper and will include the anonymous referee reports, your point-by-point response and all pertinent correspondence relating to the manuscript. Let us know whether you agree with the publication of the RPF and as here, if you want to remove or not any figures from it prior to publication. Please note that the Authors checklist will be published at the end of the RPF.

I look forward to receiving your revised manuscript.

Yours sincerely,

Lise Roth

***** Reviewer's comments *****

Referee #1 (Comments on Novelty/Model System for Author):

Unclear replication levels of the oncolytic virus tested in the MC38 model.

Hamster models are more permissive. Humanized model with fully permissive human tumor cell line and lymphos before any allogenic reactions is detected could be used.

Referee #1 (Remarks for Author):

In this paper authors present oncolytic adenoviruses armed with newly designed "MATEs", which are secreted bispecific T-cell engagers (BiTE) that use a scFv to bind CD3 on T cells and the TAT protein transduction domain to bind target tumor cells. In contrast, previous reports have used classical BiTEs that target the tumor cell with a scFV against a tumor antigen. MATEs offer the advantage that are not dependent on a specific tumor antigen, broadening the targeting and reducing the possible selection of antigen-loss variants of the tumors.

The novelty of this clever approach is worth publishing.

The oncolytic virus used is a well-designed Ad5-Ad11 hybrid with multiple insertions to control virus replication in a p53-defective tumor cells.

Monomeric and a trimeric versions of the MATE is tested.

Efficacy is well proven in the MC38 mouse model, with a detailed study of immune cells infiltrating the tumors.

My main concern is on toxicity. As the TAT domain will bind any cell non-specifically the key issue of the approach is to restrict the MATE presence to the tumor site. Authors recognize this major limitation and use intratumoral administration of an oncolytic Ad to achieve tumor- selective expression of the MATE. Even in this context, the study of potential toxicity due to the action of the MATE on normal cells should be studied carefully. Oncolytic viruses can be nearby normal cells that when killed can cause major toxicity, for example endothelial cells. Authors mention that safety is achieved by means of reduced viral replication and low levels of MATE expression, which is really a shortcoming of the strategy. In this regard, the main limitation of the safety conclusions of the paper is that authors use a mouse model (MC38) where human adenovirus replication is very limited (authors should describe how permissive is this model) and therefore the level of virus and MATE could be much higher in a human permissive tumor. Only virus permissive models can really tell the potential toxicity of the approach.

Another issue that it is not clear is why a fratricide (T cells killing each other) is not observed with the MATE, despite induced activation and proliferation as shown in figures 1 and 2.

Authors try to demonstrate induced antitumor specific immune responses by measuring T cells against a tumor epitope of the MC38 model. This epitope spread caused by the MATE approach is really important to finally obtain efficacy beyond the bystander action of the virus. In this regard the results are rather weak (no statistical differences in fig 5b, and some differences if Fig6e). It would be nice to see further studies on tumor specific immunity. Do treated animals reject distant tumors or challenges of MC38?

Despite the large amount of markers stained intratumorally to understand the immune effect of the intratumoral injection of the virus no antiviral or anti-MATE staining is done. Authors should stain for virus and MATE presence in tumors to understand where is the virus and the MATE in relation to the observed immune cells. A clear picture of the amount and distribution of the virus and the MATE in the tumors is important to interpret the results.

Referee #2 (Comments on Novelty/Model System for Author):

Only MC38 colon tumor model has been used.

Referee #2 (Remarks for Author):

Antigen-specific T cell engager-armed oncolytic viruses (OVs) have been shown to be quite efficacious for a number of tumor models in mice. However, one of the major challenges for antigen-specific immunotherapy strategies is the heterogeneity of antigen expression in tumors. This hallmark of cancer limits the potential of oncolytics by BiTE-armed OVs. To bypass this limitation, the authors have come up with a novel strategy, designed a new class of T cell engagers, called membrane-associated T-cell engagers (MATEs), which harbor the protein transduction of the HIV-Tat protein to achieve non-selective binding to target cells. They constructed a monomeric and the trimer versions and incorporated into oncolytic adenoviruses. The authors have designed and conducted a full set of experiments to demonstrate that the molecules behaves as it supposed to. Overall, the strategy is novel and worth publishing.

However, the authors tried to say that the trimeric version is better than monomeric version, yet data do not always show that. It is not sure that is a technical issue or an intrinsic property of the trimeric molecule, Thus I would suggest that some of the assays are repeated.

The other issue is that despite the oncolytic viruses armed with α CD3TAT-monomer or trimer, the efficacy is still very limited. MC38 colon tumor model has been used in many studies with oncolytic viruses, and some other OV's have been much more efficacious in this tumor. Is this due to the limited potency of oncolytic adenovirus?

Some minor issues are:

1. In last section of the Results. (neither page label nor lines labeled in the manuscript)

"Virotherapy and MATE-expressing virotherapy decreased tumor cell density suggesting a less compact structure of the tumor tissue consistent with immunoactivation (Suppl. Fig 7). In the investigated tumor areas, combination of anti PD-1 immune checkpoint blockade with Ad5/11- α CD3TAT-Trimer showed a significantly higher amount of total T cells including both total CD8+ and CD4+ T cells (Suppl. Fig 7 a-b)."

I think that in both places, it is Figure 7, not "Suppl. Fig 7". There is no Suppl Fig. 7, anyway.

2. Fig.1 legend. The letter for the panel "h" should be "f"?

3. Fig. 5b and 6a. There are no p values for comparison of important groups. For example, in Fig. 6a, there should be a value for two groups " α PD1 +Ad5/11" versus " α PD1+ CD3TAT-Trimer".

4. Suppl Fig. 2c. MOI100, MOI200. There should be a space between MOI and the numbers.

5. There are two types of issues with references.

A. Some essential information (volume and/or article number/page number) is missing.

Scott EM et al., 2018.

Scott EM et al., 2019.

B. Please delete the word "preprint" from the following references:

Benencia F. et al. 2008.

Bernt KM et al., 2005.

Bortolanza S. et al., 2009.

Dunn GP. Et al. 2004.

Engeland CE. Et al., 2014.

Kashentseva EA et al., 2002.

Referee #3 (Remarks for Author):

Florian Kühnel and colleagues report a conceptually novel approach for inducing anti-tumor immunity by intra-tumoral expression of a secreted T cell engager that non-specifically binds to (tumor) cells. The developed T cell engager format MATE (for membrane-associated T cell engager) consists of a CD3-binding single chain variable fragment (scFv), like in BiTEs, fused to a protein transduction domain, either monomeric or trimeric by insertion of a protein trimerization domain.

This non-tumor targeted T cell engager approach seems counter-intuitive at first sight. However, it is of clear medical relevance as it has the potential to overcome two key problems that established bispecific formats face: Resulting from inter- and intra-tumoral heterogeneity, respectively, bispecifics (i) require pre-treatment diagnostics and (ii) suffer from tumor escape by selection of target-negative cancer cells. Furthermore, the MATE approach, in principle, should be applicable to a broad range of tumor entities. However, it requires intra-tumoral application, as these molecules lack tumor-targeting properties. To this end, Kühnel and colleagues encode the molecule by oncolytic adenoviruses (oAd) which specifically infect and lyse tumor cells, thereby delivering their genetic payload to tumors.

The manuscript is well written, clear, instructive and of high scientific quality. The authors show proof-of-principle for MATE-mediated T cell engagement and tumor cell killing using mouse and human cell co-culture models. In comprehensive in vivo studies using syngeneic mouse models they demonstrate anti-tumor immune activation and therapeutic activity of MATE-encoding oAds, antigen spreading and efficacy of combination treatment with immune checkpoint inhibitors. One concern of the oAd-MATE approach is auto-immunity triggered by leakage of MATEs to healthy tissues. In this regard, the authors observed no early T cell activation in draining lymph nodes, thus indicating safety of the approach.

There are some limitations and open questions that the authors should address:

1. Do the authors have evidence that the MATE containing the trimerization domain does indeed trimerize?

2. Analysis of MATE-binding to CD3 on T cells: How can the authors be sure that MATEs bind to CD3 on T cells rather than via their protein transduction domain?

3. Considering the non-specific cell-binding properties of MATEs, which includes binding to producer cells, it is reassuring that the authors can demonstrate that MATEs are secreted from transfected cells, provide bystander killing and are superior to membrane-bound, CD3-specific scFvs. Could the authors elaborate on/did they assess how efficient the release of MATEs relative to retention in the producer / virus-infected cells is? Could the authors discuss how the perfusion of tumors beyond infected cells might compare with standard BiTEs and how therapeutic efficacy might be affected? And on a more global scale of their non-specific T cell engager approach: Could the authors discuss alternatives to the chosen protein transduction domain for mediating non-specific cell binding? What sets protein transduction domains apart? Are their membrane-penetrating properties a problem? Is there a window of opportunity / necessity for further improvement?

4. Killing of tumor cells in co-cultures (Fig. 1f): The increase of the percentage of dead tumor cells from a background of approx.

10% to approx. 15% by MATEs seems rather low. Can the authors elaborate?

5. The statement regarding data shown in Fig. 2f that "Consistently,.....MATEs induced production of IFN γ " seems to be too strong (true only for CD8, not CD4 cells and for monomer, not trimer).

6. Anti-tumor activity and immune activation: The similar tumor growth inhibition of the viruses encoding the MATE monomer and trimer shown in Fig. 4 is surprising considering the considerable stronger T cell activation shown for the monomer-encoding viruses shown in Fig. 3. Also, there is a considerable variation of tumor responses within the different treatment groups with high numbers of non-responders (Fig. 4a). Could the authors comment? For FACS data shown in Fig. 4b and d: Could the authors comment on the gating strategy (in comparison to the data shown in Fig. 3d) and how the T cell numbers presented reflect total numbers of intra-tumoral T cells?

7. Combination with checkpoint inhibition: Could the authors elaborate on why they excluded a MATE-encoding virus w/o checkpoint inhibition from the analysis?

8. Spatial proximity analyses: It is difficult to judge to what part the reported lower proximities shown in Fig. 7b and f result from higher cell numbers versus stronger activation of T cells. Could that be dissected by showing also cell percentages in addition to absolute numbers?

9. MATE expression by oAds: Is there a window for improving outcome by strategies for stronger and/or replication-dependent MATE expression by oAds (also considering the results for trimeric versus monomeric MATEs)?

Minor points:

- oAd: What is the rationale for the E4 modifications?
- Why did the authors not use equimolar amounts of protein for experiments shown in Figs 1 and 2?
- How does the activity of virus-encoded MATEs compare to purified recombinant MATEs?
- Materials: Could the authors provide references for the adenoviruses used ("previously described")?
- Figure legends: Fig. 1 - h should be f; Fig. 5 - isolated 7 (not 5) days following treatment?
- Results: first in vivo paragraph: reference to Fig. 2a should be 3a; reference to Suppl. Fig. 7 should be Suppl. Fig. 6
- Discussion: Eph2a should be EphA2
- Suppl. Fig. 1c: the units for the amount of protein used is missing (μg ?).

Point by point response to the Reviewers' Comments:

First of all, we want to thank all reviewers for their efforts and the overall positive evaluation of our manuscript. We also greatly appreciate their valuable advice and helpful remarks and also their ideas that allowed us to significantly improve our manuscript. We are convinced that we have sufficiently addressed their concerns. We have marked all substantial alterations by underlining. Please note that, due to the additional new data, the number of main Figures increased. Also, the supplemental figures (now Extended View Figures) have been modified to meet the restrictions set by EMBO MM. Therefore, the order of all figures has been changed and adjusted. To maintain clarity of Figures and Extended View Figures, some arrangements of representative FACS dot plots (also used to illustrate gatings, but not absolutely required to understand the figures) have been removed and included in the appendix.

Referee #1 (Comments on Novelty/Model System for Author):

*Unclear replication levels of the oncolytic virus tested in the MC38 model.
Hamster models are more permissive. Humanized model with fully permissive human tumor cell line and lymphos before any allogenic reactions is detected could be used.*

Response: We appreciate the suggestions of alternative models that we will consider for our follow-up projects. MC38 cells, similar to other murine cells reported in the literature, show only low levels of virus particle formation after infection with type 5 based human adenovirus or Ad5/11. We have addressed more aspects concerning susceptibility of MC38 cells for Ad5/11 in vitro and in vivo in the revised version and described below. Using the MC38 model does not reflect the viral load and concomitant levels and duration of MATE secretion that may occur in a fully permissive human therapy situation. We also agree that hamster models are more permissive for human adenovirus regarding replication and subsequent oncolysis. However, it was our primary goal to understand the immunological function of MATE expression on tumor immune activation and remodeling of the tumor microenvironment in an vivo model which is well established in the tumor immunotherapy community. The advantage of the MC38 model concerning the analytical focus on immune activation is that observable effects are preferably MATE-mediated whereas tumor tissue lysis plays a minor role. However, we are aware that effects mediated by strong oncolysis will be certainly an important amplifier for the observed MATE effects here in our study and that systemic risks of MATEs might therefore be underestimated. We also have addressed this in the same model by using novel virus variants with enhanced MATE expression as shown in the new figures 4 and EV3 as described in detail below.

We also agree with the reviewer that humanized models could provide additional information on the aspect of replication, but these models are less well suited for monitoring adaptive immune responses and modulation of the tumor microenvironment in a longer perspective.

Referee #1 (Remarks for Author):

In this paper authors present oncolytic adenoviruses armed with newly designed "MATEs", which are secreted bispecific T-cell engagers (BiTE) that use a scFv to bind CD3 on T cells and the TAT protein transduction domain to bind target tumor cells. In contrast, previous reports have used classical BiTEs that target the tumor cell with a scFV against a tumor antigen. MATEs offer the advantage that are not dependent on a specific tumor antigen, broadening the targeting and reducing the possible selection of antigen-loss variants of the tumors.

The novelty of this clever approach is worth publishing.

The oncolytic virus used is a well-designed Ad5-Ad11 hybrid with multiple insertions to control virus replication in a p53-defective tumor cells.

Monomeric and a trimeric versions of the MATE is tested.

Efficacy is well proven in the MC38 mouse model, with a detailed study of immune cells infiltrating the tumors.

Response: We greatly appreciate the positive evaluation of our approach by reviewer I!

Referee #1 My main concern is on toxicity. As the TAT domain will bind any cell non-specifically the key issue of the approach is to restrict the MATE presence to the tumor site. Authors recognize this major limitation and use intratumoral administration of an oncolytic Ad to achieve tumor-selective expression of the MATE. Even in this context, the study of potential toxicity due to the action of the MATE on normal cells should be studied carefully. Oncolytic viruses can be nearby normal cells that when killed can cause major toxicity, for example endothelial cells.

Response: We agree with reviewer 1 that release of a non-selective agent is the crucial point of our study regarding safe application of the agent. We have therefore performed additional tumor infection experiments in mice to investigate possible hazards and adverse events in more detail and higher sensitivity including liver toxicity (histology and ALT-release in serum of treated mice) and surrogate markers of a cytokine release syndrome (see below). To mimic higher levels of MATEs in these investigations without changing the entire model system we have also applied novel Ad5/11 variants with elevated MATE expression levels, that we have recently generated. In these variants as described in the new Fig. 4 and Fig. EV3 strong MATE expression is facilitated by an expression cassette under transcriptional control of a PGK-promoter (new scheme in Fig. EV3 C). These viruses may be useful to explore further eventual limits of the concept and to develop this technology further. What we found is that these viruses led to much more MATE expression after infection of MC38 cells in vitro (Fig. EV3 D). After in vivo infection of MC38 tumors, we were able to observe increase of T cell activation in tumor draining lymph nodes suggesting that MATEs have indeed been produced in excess and have been released into the lymphatic system (new Fig 4 F). However, in all animals investigated including those treated with the new Ad5/11 variants we could neither detect liver damage nor signs of a cytokine release syndrome and if some elevated cytokines could be seen this did not correlate with the highest T cell activation seen in tumor draining lymph nodes. Importantly, the lead CRS cytokine IL-6 remained very low in all groups. Observable differences in single cytokines were not significant. When regarding the cytokine array overview (Fig 4 G), there is no effect visible that can be attributed to elevated MATE levels (PGK-variant vs. the corresponding non-PGK variant of Ad5/11). This suggests that even strongly elevated MATEs with detectable T cell activation in tumor draining lymph nodes may not cause systemic toxicity or a cytokine release syndrome. The results text on this new passage can be found in lines 236-247, 263-319 and in the discussion between lines 585-596.

Referee #1 Authors mention that safety is achieved by means of reduced viral replication and low levels of MATE expression, which is really a shortcoming of the strategy. In this regard, the main limitation of the safety conclusions of the paper is that authors use a mouse model (MC38) where human adenovirus replication is very limited (authors should describe how permissive is this model) and therefore the level of virus and MATE could be much higher in a human permissive tumor. Only virus permissive models can really tell the potential toxicity of the approach.

Response: We agree with the reviewer, that the low permissivity of the used model for virus replication certainly reduces the MATE levels and duration of MATE release. First, we have stated in the results text, that it is well known, that replication of human adenovirus is generally low in murine cells (lines 238-241). To investigate permissivity of MC38 in more detail, we have first experimentally addressed susceptibility for virus entry in these cells, we have included an infection assay using an Ad5-based EGFP vector (new Fig EV3 A) demonstrating that virus entry in MC38 is fairly well. Particle formation is experimentally addressed in our new Fig. EV3 B in comparison to well permissive human H1299 cells. Additionally, we have confirmed successful tumor infection in vivo by immunohistochemical staining for E1A (virus) and MATEs (see new Fig 4 E). The respective text passages can be found in lines 243-247, and in lines 272-274.

Referee #1 Another issue that it is not clear is why a fratricide (T cells killing each other) is not observed with the MATE, despite induced activation and proliferation as shown in figures 1 and 2.

Response: We can definitely not exclude that fratricide indeed occurred in our assays. Actually, we did not directly investigate this aspect because in our assays the applied murine T cells (in splenocytes) were rather short-lived and quickly die, probably involving a certain degree of fratricide and AICD in the presence of MATEs thus aggravating a detailed discrimination of these mechanisms. Nevertheless, the aspect of fratricide is certainly an interesting point that should be addressed with refined culturing and assay conditions in future studies.

Referee #1 Authors try to demonstrate induced antitumor specific immune responses by

measuring T cells against a tumor epitope of the MC38 model. This epitope spread caused by the MATE approach is really important to finally obtain efficacy beyond the bystander action of the virus. In this regard the results are rather weak (no statistical differences in fig 5b, and some differences if Fig6e). It would be nice to see further studies on tumor specific immunity. Do treated animals reject distant tumors or challenges of MC38?

Response: Sorry, we did not rechallenge the mice, and we have not used double-sided tumors. Testing a potential epitope spread is certainly of great interest for a further understanding how MATEs may drive antitumoral immunity. However, in our assays, probably because of low MATE levels secreted by the applied Ad5/11 variants, the levels of adpgk-pos. cells that we obtained from tumor tissue were rather low, also with respect to the tumor size. Because T cells specific for the adpgk-mutation belong to the more frequent among the tumor infiltrating T cells, it would require a large number of treated animals to facilitate a successful search for even less frequent T cells when using these first generation viruses. Our novel viruses with enhanced MATE expression maybe provide us with the opportunity to treat even larger tumor sizes and to perform these studies in the future.

Referee #1 Despite the large amount of markers stained intratumorally to understand the immune effect of the intratumoral injection of the virus no antiviral or anti-MATE staining is done. Authors should stain for virus and MATE presence in tumors to understand where is the virus and the MATE in relation to the observed immune cells. A clear picture of the amount and distribution of the virus and the MATE in the tumors is important to interpret the results.

Response: Using FFPE tumor material from virotherapy treated mice, we have included a merged virus staining (EIA) and a staining for MATE expression (c-myc) which is now shown in our new Figure 4E confirming that tumor cells have been successfully infected in vivo (manuscript text, lines 272-274). Further confirmation of successful tumor infection comes from our studies with novel generated variants expressing higher levels of MATEs (new Figure 4F). The respective text passages can be found in lines 263-319. There we could show, that after 2 days, high T cell activation was observed in the tumor draining lymph node indicating a sufficient tumor cell infection in vivo and a release of excessively produced MATEs into the lymphatic system. However, a detailed spatial analysis of replication/MATE release in relation to the observed immune cells was not possible within the time constraints of a revision. This would mean almost an additional project including the establishment of new Multiplex panels and new antibodies to meet the needs of the Akoya method. Furthermore, viral replication and MATE release processes certainly peak at day 2 and will quickly vanish due to the low permissivity of the used model. Most obvious and interesting immune alterations we have observed at day 7 and there is probably too little temporal overlap to allow for promising immunohistochemical study at one given time point.

Referee #2 (Comments on Novelty/Model System for Author):

Only MC38 colon tumor model has been used.

Response: Thanks for the advice, it will be certainly helpful to further validate our results in other tumor models. We are planning this in subsequent studies when testing also elevated MATE expression levels.

Referee #2 (Remarks for Author):

Antigen-specific T cell engager-armed oncolytic viruses (OVs) have been shown to be quite efficacious for a number of tumor models in mice. However, one of the major challenges for antigen-specific immunotherapy strategies is the heterogeneity of antigen expression in tumors. This hallmark of cancer limits the potential of oncolytics by BiTE-armed OVs. To bypass this limitation, the authors have come up with a novel strategy, designed a new class of T cell engagers, called membrane-associated T-cell engagers (MATEs), which harbor the protein transduction of the HIV-Tat protein to achieve non-selective binding to target cells. They constructed a monomeric and the trimer versions and incorporated into oncolytic adenoviruses. The authors have designed and conducted a full set of experiments to demonstrate that the molecules behaves as it supposed to. Overall, the strategy is novel and worth publishing.

Response:

We thank this reviewer for the positive overall evaluation of our manuscript!

Referee #2: However, the authors tried to say that the trimeric version is better than monomeric version, yet data do not always show that. It is not sure that is a technical issue or an intrinsic property of the trimeric molecule, Thus I would suggest that some of the assays are repeated.

Response:

We agree with the reviewer that some of our data do not fully support a general superiority of the trimer compared with the monomer. Maybe our argumentation was also biased by our trimer results at later time points, which showed the more convincing tumor immune activation and was considered the more relevant readout and prompted us to focus on studying the trimer for subsequent checkpoint combination studies. We tried to better explain this in the text to avoid being misunderstood (Modified text passages on this can be found between lines 140-141, 231-236 and 420-421). The monomer appeared to result in better T cell activation at early time points. However, such early readouts are more influenced by different kinetics of MATE availability in the environment of producing cells compared with the later readouts at day 7 which better reflect a 'final' readout. The expression data we have obtained with our first generation Ad5/11 variants show lower levels of available trimer in the supernatant of infected cells (Fig. 3 B) which was observed repeatedly. This availability of MATEs in the supernatant is probably due to the different affinity of both MATEs to the surface of cells (presumably higher in case of the trimer) and also affected by the fact that Tat-PTD-containing proteins are subject to rapid cellular uptake and resorption (see comment and citation in lines 111-113), in the first place at the producer cell. Because binding and resorption at the producing cell is presumably more efficient in case of the trimer until saturation is being reached, it is likely that the monomer is being easier and earlier released leading to higher levels in the supernatant. We think that these aspects of different release kinetics are probably reflected by our observations at early time points. It would certainly be interesting to address these aspects in more detail with closer time points. To elevate MATE-levels and to further improve therapeutic efficacy of the concept, we have already generated novel variants with increased MATE expression levels. First expression studies (like in MC38 cells shown in new Fig. EV3 D) suggest that these viruses effectively secrete the trimeric MATE into the supernatant, maybe because the resorption capacity of that system has been effectively overcome. This is consistent with the in vivo T cell activation results in tumor draining lymph nodes shown in the new Fig. 4F. The corresponding text passages on these new virus variants in the revised manuscript can be found in lines 236-247, 263-319 and lines 585-596. We are eager to learn in upcoming experiments how these new variants will impact therapeutic outcome.

Referee #2: The other issue is that despite the oncolytic viruses armed with α CD3TAT-monomer or trimer, the efficacy is still very limited. MC38 colon tumor model has been used in many studies with oncolytic viruses, and some other OV's have been much more efficacious in this tumor. Is this due to the limited potency of oncolytic adenovirus?

Response: The reviewer is right. Low virus replication in this model is certainly a major cause for the limited efficacy of the virus even when armed with MATEs. It can be seen in all experimental groups involving naked Ad5/11 that virus replication/lysis is low and affect the performance of the model regarding this aspects. However, with respect to the added value of MATEs, the low expression level in our first generation Ad5/11 variants has to be taken into account. As described, MATEs in these viruses were expressed by the E1B locus downstream of an IRES-motif, which is known to be less powerful compared with the upstream position of the IRES or when controlled by strong promoters. Our expression data all suggest low levels of available MATEs in the supernatants of infected cells (even after infection of virus-producer cells, HEK293, enrichment of Ni-beads was required to allow for positive detection of MATEs by Western Blot). Regarding the availability of MATEs in the supernatant (or later in the tumor microenvironment) it has to be considered that Tat-PTD based proteins are subject to turnover by cell binding and subsequent uptake by either producing cells or bystander cells thus reducing the free floating levels. We commented this in lines 111-113. On the other hand this resorption which should take place first in the MATE-producing tumor area is part of our safety considerations and also crucial for our approach.

Referee #2 Some minor issues are:

1. In last section of the Results. (neither page label nor lines labeled in the manuscript) "Virotherapy and MATE-expressing virotherapy decreased tumor cell density suggesting a less compact structure of the tumor tissue consistent with immunoactivation (Suppl. Fig 7). In the investigated tumor areas, combination of anti PD-1 immune checkpoint blockade with Ad5/11- α CD3TAT-Trimer showed a significantly higher amount of total T cells including both total

CD8+ and CD4+ T cells (Suppl. Fig 7 a-b)."

I think that in both places, it is Figure 7, not "Suppl. Fig 7". There is no Suppl Fig. 7, anyway.

Response: Thanks for the advice. We have added line and page numbers and apologize for the mislabeling of figures that we have carefully reviewed in the revised version. Please be aware that most figure labeling has changed due to additional data, the new Figure 4 and the modified arrangement of former supplemental figures in max. 5 Extended View Figures.

Referee #2

2. Fig.1 legend. The letter for the panel "h" should be "f"?

Response: Thanks for the advice, We have corrected this, please also see my previous response.

Referee #2

3. Fig. 5b and 6a. There are no p values for comparison of important groups. For example, in Fig. 6a, there should be a value for two groups "αPD1 +Ad5/11" versus "αPD1+ CD3TAT-Trimer".

Response: Thanks for the advice, We have added p-values when those were below 0.05. In 6a (previous version) which is now Fig 7a, the two mentioned groups were statistically not different. We indicated this in the figure and modified the results text accordingly (lines 385-283).

Referee #2

4. Suppl Fig. 2c. MOI100, MOI200. There should be a space between MOI and the numbers.

Response: We have changed this.

Referee #2

5. There are two types of issues with references.

A. Some essential information (volume and/or article number/page number) is missing.

Scott EM et al., 2018.

Scott EM et al., 2019.

B. Please delete the word "preprint" from the following references:

Benencia F. et al. 2008.

Bernt KM et al., 2005.

Bortolanza S. et al., 2009.

Dunn GP. Et al. 2004.

Engeland CE. Et al., 2014.

Kashentseva EA et al., 2002.

Response: We apologize these incomplete references and have corrected this accordingly.

Referee #3 (Remarks for Author):

Florian Kühnel and colleagues report a conceptually novel approach for inducing anti-tumor immunity by intra-tumoral expression of a secreted T cell engager that non-specifically binds to (tumor) cells. The developed T cell engager format MATE (for membrane-associated T cell engager) consists of a CD3-binding single chain variable fragment (scFv), like in BiTEs, fused to a protein transduction domain, either monomeric or trimeric by insertion of a protein trimerization domain. This non-tumor targeted T cell engager approach seems counter-intuitive at first sight. However, it is of clear medical relevance as it has the potential to overcome two key problems that established bispecific formats face: Resulting from inter- and intra-tumoral heterogeneity, respectively, bispecifics (i) require pre-treatment diagnostics and (ii) suffer from

tumor escape by selection of target-negative cancer cells. Furthermore, the MATE approach, in principle, should be applicable to a broad range of tumor entities. However, it requires intra-tumoral application, as these molecules lack tumor-targeting properties. To this end, Kühnel and colleagues encode the molecule by oncolytic adenoviruses (oAd) which specifically infect and lyse tumor cells, thereby delivering their genetic payload to tumors. The manuscript is well written, clear, instructive and of high scientific quality. The authors show proof-of-principle for MATE-mediated T cell engagement and tumor cell killing using mouse and human cell co-culture models. In comprehensive in vivo studies using syngeneic mouse models they demonstrate anti-tumor immune activation and therapeutic activity of MATE-encoding oAds, antigen spreading and efficacy of combination treatment with immune checkpoint inhibitors. One concern of the oAd-MATE approach is auto-immunity triggered by leakage of MATEs to healthy tissues. In this regard, the authors observed no early T cell activation in draining lymph nodes, thus indicating safety of the approach. There are some limitations and open questions that the authors should address:

Response: We thank the reviewer for this overall positive evaluation of our manuscript.

Referee #3

1. Do the authors have evidence that the MATE containing the trimerization domain does indeed trimerize?

Response: The fibrin-trimerization domain is well established for similar adapter proteins and has been published by the Curiel group as mentioned in the methods section. We have also used this motif in earlier (but unpublished) projects, where it increased the efficacy of bispecific adapters to facilitate adenovirus entry in non-permissive target cells compared to monomeric versions. As shown in our new Fig. EVI A (right panel) we checked trimerization in a protein preparations by a Blue-Native PAGE using a 5% separation gel under non-denaturing conditions with Coomassie-containing cathode buffers to visualize the appropriate size controls (66 kD and 150 kD) commented in lines 124-125 of the revised manuscript. MATEs were then detected by Western blot. Details have been described in the respective methods section and the according figure legend. It can be seen that at least in the preparation used here, the trimeric conformation also contained a monomeric fraction, maybe due to incomplete folding or due to damage during purification at the Ni-NTA resin.

Referee #3

2. Analysis of MATE-binding to CD3 on T cells: How can the authors be sure that MATEs bind to CD3 on T cells rather than via their protein transduction domain?

Response: It is definitely possible that MATEs bind to T cells via their PTD-domain. We think that at least what we see in our antibody stainings and FACS analyses is primarily the binding to CD3. According to our experimental experience, it was difficult to detect MATE binding (when using a protein consisting only of Tat-PTD and Hinge but lacking the antiCD3 domain (data not shown). Though we cannot exclude a technical issue of the FACS detection in case of Tat-mediated cell binding, the most plausible explanation is that rapid resorption of the Tat-protein removes the complex from the cell surface. Quick resorption of Tat-linked proteins as mentioned in the results text is well known (see lines 111 – 113 and citation). Regarding the binding of Tat to the cell surface we also had excellent experience using Tat-containing adapters to mediate adenovirus binding and internalization in non-permissive cells (Kühnel et al. 2004, J. Virol). On the other hand, when using a true MATE with antiCD3scFv it can be easily visualized on the cell surface as shown in Fig.1. The endocytotic uptake of MATEs and removal from the surface is therefore obviously prevented when bound to its CD3 target also suggesting that MATE binding to CD3 is more stable than the membrane association via Tat. The high endocytotic uptake of MATEs has implications for the approach. The resorption reduces the availability of MATEs in the tumor microenvironment, as far as the target CD3 has not yet been bound. Together, this assumption supports our aim of safe applicability.

Referee #3

3. Considering the non-specific cell-binding properties of MATEs, which includes binding to producer cells, it is reassuring that the authors can demonstrate that MATEs are secreted from transfected cells, provide bystander killing and are superior to membrane-bound, CD3-specific scFvs.

Response: In our first version we already showed increased efficacy of T cell activation when comparing the presence of MATE-expressing cells compared with a membrane-bound T cell engager. In our revised version we have now strengthened these findings with new cytotoxicity assays shown in the new Figs. I G and I H. The according text passages in the manuscript can be found between lines 166 – 180 and 559-561. In these assays, we discriminated the T-cell mediated cytotoxicity on MATE-producer cells vs. bystander cells using different labelling. Differential cell killing was observable but less pronounced when regarding the producer cells, which could be expected regarding the assumption that MATEs should be present at saturated levels on the producer cell surface. Killing of bystander cells was more effective by MATEs compared with the membrane bound version. The effect was significant for both MATE versions, when the cell killing ratios were determined. The data therefore demonstrate that MATEs improve T cell-mediated cytotoxicity against bystander cells.

Referee #3 Could the authors elaborate on/did they assess how efficient the release of MATEs relative to retention in the producer / virus-infected cells is?

Response: We have not directly addressed this aspect. It is certainly a very interesting mechanistic point to understand for further improvement of the system.

Referee #3 Could the authors discuss how the perfusion of tumors beyond infected cells might compare with standard BiTEs and how therapeutic efficacy might be affected?

Response: In case of BiTEs I would expect that the resorption is absent or less pronounced that is characteristic for Tat-PTD containing proteins. Therefore, our guess would be more intense tumor perfusion with BiTEs. However, this may depend on the use of specific BiTE target molecule, particularly when able to trigger any endocytotic processes promoting the removal of the BiTE from the cell surface.

Referee #3 And on a more global scale of their non-specific T cell engager approach: Could the authors discuss alternatives to the chosen protein transduction domain for mediating non-specific cell binding? What sets protein transduction domains apart? Are their membrane-penetrating properties a problem? Is there a window of opportunity / necessity for further improvement?

Response: These are definitely important questions for further advancing the technology. There are many other protein transduction domains available of both natural origin such as a PTD from Antennapedia protein or synthetic one containing a stretch of arginine residues such as 9 x Arg. The Tat-PTD was mainly chosen arbitrarily by us as a starting point, driven by good experimental experiences with Tat-PTD containing adapter proteins in the past [1]. Since our approach using Tat-PTD worked well in vitro we proceeded quickly to proof of principle in virus infections and in vivo before comparing with alternative Tat-replacements. However, it could possibly provide an advantage to apply a cell attachment domain with less pronounced membrane transduction abilities like Tat in order to reduce the loss by resorption. On the other hand, the efficacy of Tat-PTD-mediated resorption also represents a safety hallmark because resorption by tumor cells prevents that excessively released MATEs may reach toxic levels. There are certainly several parameters to work on to optimise MATE levels and availability by testing different PTDs and PTD equivalent motifs in concert with different expression control mechanisms as we have realized now with our new variants harboring a separate cassette under control of a PGK promoter.

Referee #3

4. Killing of tumor cells in co-cultures (Fig. 1f): The increase of the percentage of dead tumor cells from a background of approx. 10% to approx. 15% by MATEs seems rather low. Can the authors elaborate?

Response: Spontaneous death of MC38 is relatively high in culture with splenocytes. Furthermore, freshly isolated murine splenocytes are very sensitive and rapidly die under cell culture conditions possibly explaining high background and low increase of cell killing in presence of MATEs. To strengthen our cytotoxicity data, we have now included in our revised version further assays to show bystander cell killing as described above and shown in the new figures I G and H as commented above.

Referee #3

5. The statement regarding data shown in Fig. 2f that "Consistently,.....MATEs induced production of IFN γ " seems to be too strong (true only for CD8, not CD4 cells and for monomer, not trimer).

Response: Thanks for the advice, the statement has been appropriately adjusted, (lines 196-198)

Referee #3

6. Anti-tumor activity and immune activation: The similar tumor growth inhibition of the viruses encoding the MATE monomer and trimer shown in Fig. 4 is surprising considering the considerable stronger T cell activation shown for the monomer-encoding viruses shown in Fig. 3.

Response: Fig. 3 shows an early time point of T cell activation reflecting a snapshot. This is presumably influenced by the kinetics of MATE release which is presumably different between the two conformations. Our expression data all suggest that the trimer is less effectively released and may therefore also lagging behind. We believe that our immediate T-cell activation results may reflect these differences. On the other hand, the trimer showed more favorable immunoactivation characteristics in the longer course of the experiment representing a more stable and biological relevant readout. This is also in line with our in vitro activation and cytotoxicity assays where the trimer seemed to have a qualitative advantage compared with the monomer. The kinetics of immediate T cell activation in vivo should be addressed in future studies with smaller time intervals to clarify these mechanistical questions.

The contrast between the lower T cell activation markers by the trimer at early time points but improved immunoactivation characteristic in tumor tissue in the long run also argues for a qualitative advantage of the trimer in T cell activation.

Referee #3 Also, there is a considerable variation of tumor responses within the different treatment groups with high numbers of non-responders (Fig. 4a). Could the authors comment?

Response: We still do not know about the decisive factors tipping the balance towards complete response and what prevents complete response in the MC38 model. Maybe variable injection procedures, as well as slight differences in tumor size, stroma extent and vascularization at the time point of injection may also play a role.

Referee #3 For FACS data shown in Fig. 4b and d: Could the authors comment on the gating strategy (in comparison to the data shown in Fig. 3d) and how the T cell numbers presented reflect total numbers of intra-tumoral T cells?

Response: In the old Figure 3 D (now Fig 4 B) only the CD4 and CD8 gates within the CD45 Gates are analyzed for activation markers. Therefore, any conclusions regarding the total numbers is difficult. However, it is unlikely that total T cell numbers vary between these groups at this early time point. The gatings in the old figure 4, now Figure 5 also do not directly include a calculation referring to the total numbers but indirectly the presented T cell numbers should strongly correlate with the total numbers of intratumoral T cells regarding the immunohistochemistry shown in Fig. 5C and 5D.

Referee #3

7. Combination with checkpoint inhibition: Could the authors elaborate on why they excluded a MATE-encoding virus w/o checkpoint inhibition from the analysis?

Response: A good question and the simple explanation is that this is the experimental setup according to our animal experimentation proposal and the experimental setup for which we finally obtained legal permission.

Referee #3

8. Spatial proximity analyses: It is difficult to judge to what part the reported lower proximities shown in Fig. 7b and f result from higher cell numbers versus stronger activation of T cells. Could that be dissected by showing also cell percentages in addition to absolute numbers?

Response: Actually, we tried to arrange the data as cell percentages, but the figures were not really different from those shown in the manuscript. However, we have now made a further attempt for each distance range using Pearson correlation analysis to investigate whether the number of T cells in proximity is simply a linear correlation depending on the frequency of the respective cell type. Such a

linear correlation can be clearly seen in the close (30 μm), and near (50 μm) distance range showing that within these ranges the number of defined T cells in proximity to others is indeed a function of cell frequency. In contrast, the correlation analysis suggests that the correlation coefficient significantly decreases in the nearest distance range (15 μm), defined by us as direct contact. This effect was more pronounced in the MATE groups (trimer), compared to Ad5/11 vector. However, when looking at the few individual data points it is clear that more data points would be needed to clarify a biological significance of the findings so that we finally decided to not include the analysis in the manuscript. The Pearson correlation analysis is shown below for your personal information.

Referee #3

9. MATE expression by oAds: Is there a window for improving outcome by strategies for stronger and/or replication-dependent MATE expression by oAds (also considering the results for trimeric versus monomeric MATEs)?

Response: There is certainly a window for improvement to raise MATE levels indirectly as a consequence of stronger replication when using more potently replicating viruses as carriers compared to Ad5/11. Also modified genetic arrangements can be considered, e.g. control by strong promoters as we have done now with our current PGK promoter control shown in the new Figs. 4 and Fig EV3 (see above in responses to reviewer 1 and 2). Harmonization with replication when using the major late promoter could also be an interesting option though not easy to predict what may be the consequence.

Referee #3

Minor points:

- oAd: What is the rationale for the E4 modifications?

Response: The rationale for the modifications was to obtain a virus with modestly, but not essentially reduced replication properties but with at least maintained or even increased immunogenicity.

Referee #3 • Why did the authors not use equimolar amounts of protein for experiments shown in Figs 1 and 2?

Response: We considered that a comparison on a stoichiometrical base could also cause confusion regarding the fundamental difference of these two concepts (e.g. affinity to both targets). We found applying similar amounts was in this initial phase of the study the most 'unbiased' approach.

Referee #3 • How does the activity of virus-encoded MATEs compare to purified recombinant MATEs?

Response: We have not tested this directly. I would expect the virus-encoded MATE more active because there should be no loss of function that can be usually not completely avoided when proteins are subject to purification procedures.

Referee #3 • Materials: Could the authors provide references for the adenoviruses used ("previously described")?

Response: The reference has been provided in the revised version.

Referee #3 • Figure legends: Fig. 1 - h should be f; Fig. 5 - isolated 7 (not 5) days following treatment?

Response: has been corrected

Referee #3 • Results: first in vivo paragraph: reference to Fig. 2a should be 3a; reference to Suppl. Fig. 7 should be Suppl. Fig. 6

Response: has been corrected

Referee #3 • Discussion: Eph2a should be Epha2

Response: has been corrected

Referee #3 • Suppl. Fig. 1c: the units for the amount of protein used is missing (ug?).

Response: has been added

References:

I. Kühnel F, Schulte B, Wirth T, Woller N, Schäfers S, Zender L, et al. Protein transduction domains fused to virus receptors improve cellular virus uptake and enhance oncolysis by tumor-specific replicating vectors. *J Virol.* 2004; 78:13743–54.

22nd Oct 2024

Dear Prof. Kuehnel,

Thank you for submitting your revised study. We have now received the reports from the three initial referees. As you will see from the reports below, while referees #1 and #3 are overall satisfied with the revisions, referee #2 still raises concerns on the model used (MC38 murine colon tumor model in syngeneic C56BL/6 mice). I have further consulted with the referees on this point, and they all agreed that no additional experiments should be requested at this stage (considering also the general lack of good model for immunotherapy with human oncolytic adenoviruses), however the limitations of the model system should be adequately discussed in the final manuscript. Moreover, in a minor round of revisions, please address the following points:

1/ Please consider the minor comment from referee #3 on including information in the manuscript.

2/ Manuscript text:

- Please indicate in track changes mode any new modification.
- We noted the following name discrepancies: Bettina Mundt in the manuscript file vs. Bettina Fleischmann-Mundt in the submission system; Valery Volk in the manuscript vs. Valer Volk in the submission system. Please clarify.
- Please provide up to 5 keywords.
- The manuscript sections should be in the following order: Title page - Abstract & Keywords - Introduction - Results - Discussion - Methods - Data Availability - Acknowledgments - Disclosure Statement & Competing Interests - References - Figure Legends - (Main Tables with legends) - Expanded View Figure Legends.
- Methods:
 - o Thank you for providing a reagents and tools table, please remove it from the manuscript, and upload it as a separate file.
 - o Please remove "All sequences and cloning details can be provided upon request" (genetic engineering of MATEs), as per journal policy, all necessary information should be available in the manuscript.
 - o Antibodies: please provide dilutions/concentrations for all experiments.
 - o Cell lines: please indicate whether the cells were authenticated and tested for mycoplasma contamination.
 - o Animal experiments: please provide the origin and gender of the mice, as well as housing and husbandry conditions.
 - o Human samples: please include a statement confirming that informed consent was obtained from all subjects and that the experiments conformed to the principles set out in the WMA Declaration of Helsinki and the Department of Health and Human Services Belmont Report.
 - o Statistics: please provide information on blinding, randomization, sample size and inclusion/exclusion criteria.
- Acknowledgments: the information provided here and in the submission system should match; Claudia-von-Schilling-Stiftung is missing in the submission system.
- Author contributions: CRediT has replaced the traditional author contributions section because it offers a systematic machine readable author contributions format that allows for more effective research assessment. Please remove the Authors Contributions from the manuscript and use the free text boxes beneath each contributing author's name in our system to add specific details on the author's contribution. More information is available in our guide to authors.
- Disclosure and competing interest statement: please provide more information regarding the patents (who and what).

3/ Figures and Appendix:

- We do not accept figures in ppt format. Please upload them as EPS, TIF(F) or PDF format (1 file per figure).
- The Appendix should be in PDF format; the legend should be removed from the manuscript and provided after the figure in the Appendix file; the file should have a table of content with page number; the reference in the manuscript text needs to be updated to Appendix Figure S1 (there is one instance of "Supplemental Fig 1").
- Figure/image re-use should be indicated in the legend (i.e. Figure 5C and Figure EV4G).
- Please address the queries from our copy editors in the figure legends:
- Please define the annotated p values **/**** as well as provide the exact p-values for the same in the legend of figure EV 4c, f; as appropriate.
- Please note that the exact p values are not provided in the legends of figures 1c-e, g-h; 2d, g; 3c; 8c, g; EV 1c, g; EV 5c.
- Please indicate the statistical test used for data analysis in the legends of figures EV 3h; EV 5h.
- Please note that information related to n is missing in the legends of figures 4c, f, h; 5a; EV 3b; EV 4d.
- Although 'n' is provided, please describe the nature of entity for 'n' in the legends of figures 2d-g; 3c; 5d, f; 6c-d; 7a-c; 8b-d, f-h; EV 1c-d; EV 2c; EV 3h; EV 4c, f; EV 5b, f, h.
- Please note that the error bars are not defined in the legends of figures EV 3b; EV 4d; EV 5h.
- Please note that the measure of center for the error bars needs to be defined in the legends of figures 2d-g; 5b, f; 6b-c; 7b-e; 8b-c, f-g; EV 1c-d, g-h; EV 2c.
- Please note that the scale bar is missing for figure 8a.
- Please note that the scale bar needs to be defined for figure EV 3g.
- Please note that scale bar and its definition are missing for figures EV 3a; EV 5g.
- Please note that the white arrows are not defined in the legend of figure 8e. This needs to be rectified.

4/ At EMBO Press we ask authors to provide source data for the main figures. Our source data coordinator has been in touch with you to discuss which figure panels we would need source data for and will also provide you with helpful tips on how to upload and organize the files.

5/ Checklist:

- please clarify which restrictions apply to your work
- please adjust the section on authentication and mycoplasma checks
- please check the section "human research participants"
- please fill in the whole section "Experimental study design and statistics"
- please fill in "sample definition and in-laboratory replication", first entry
- please fill in "Ethics"/"informed consent" (second entry)

6/ Thank you for providing The Paper Explained. Please include it in the manuscript file.

7/ Synopsis:

- Text: I slightly edited your synopsis to match our style and format. Please let me know if you agree with the following or amend as you see fit:

"Oncolytic adenoviruses were designed to reach potent intratumoral T-cell activation via expression of bifunctional-T-cell engagers (MATEs). MATEs consist of a CD3 binding domain to bind T cells and a protein transduction domain (Tat-PTD) to enable target-independent attachment to the cell surface.

- In vitro, MATEs promote potent activation of murine and human T cells, induce their proliferation and trigger T cell-mediated bystander toxicity.
- MATEs can be expressed by an oncolytic adenovirus and are secreted by cells upon virus infection.
- Tumor infection with MATE-expressing oncolytic adenoviruses leads to T cell activation and tumor immunoactivation in murine tumor models, and results in a CD8-dependent therapeutic benefit.
- Intratumoral T cell activation by MATE-expressing oncolytic adenovirus does not result in systemic toxicity or a cytokine release syndrome.
- Intratumoral T cell activation by MATE-expressing oncolytic viruses sensitizes tumors to PD-1 checkpoint inhibitor therapy."

- Visual abstract: please upload as jpeg/tiff/png format, 550 px wide x 300-600 px high. Please make sure the text remains legible. If you used an image database for scientific iconography (e.g., BioRender), let us know if you have a license that allows for publication in an academic journal, and reference the image database in the methods section. A cropped portion (115px x 70px) of this image will serve as thumbnail for the table of content on our webpage.

8/ As part of the EMBO Publications transparent editorial process initiative (see our Editorial at <http://embomolmed.embopress.org/content/2/9/329>), EMBO Molecular Medicine will publish online a Review Process File (RPF) to accompany accepted manuscripts.

This file will be published in conjunction with your paper and will include the anonymous referee reports, your point-by-point response and all pertinent correspondence relating to the manuscript. Let us know whether you agree with the publication of the RPF and as here, if you want to remove or not any figures from it prior to publication.

I look forward to receiving your revised manuscript.

Yours sincerely,

Lise Roth

***** Reviewer's comments *****

Referee #1 (Remarks for Author):

None. Authors have properly addressed my concerns in the reviewed version of the manuscript.

Referee #2 (Comments on Novelty/Model System for Author):

The authors have used an defective tumor model for this oncolytic virus in the study.

Referee #2 (Remarks for Author):

In the revised version, the authors have clarified a number of points and concerns reviewers had. However, some major issues and defects in the experimental design still exist.

For example, the authors have utilized MC38 murine colon tumor model in syngeneic C56BL/6 mice. As investigators in the field know, the used adenovirus (hybrid serotypes 5/11) is human adenovirus, hardly replicating in MC38 cancer cells, thus may not even qualified to define as an oncolytic virus in this tumor model. In addition, as human adenovirus does not replicate in murine normal cells/normal tissues, the toxicity of this virus to the host cannot be observed or studied. Ideally, they should have used a tumor model in Syrian hamsters that is permissive to human adenovirus.

As murine MC38 cancer is not permissive to the human adenovirus used by the authors, this also explain why their therapeutic results were poor as compared to other oncolytic virus, even an unarmed oncolytic virus (such as herpes simplex virus or vaccinia virus) in this tumor model.

Referee #3 (Remarks for Author):

Florian Kühnel and colleagues have revised their manuscript about a conceptually novel approach for inducing anti-tumor immunity by intra-tumoral expression of a secreted T cell engager that non-specifically binds to (tumor) cells. The developed T cell engager format MATE (for membrane-associated T cell engager) consists of a CD3-binding single chain variable fragment (scFv), like in BiTEs, fused to a protein transduction domain, either monomeric or trimeric by insertion of a protein trimerization domain.

The authors thoroughly revised their study according to the referees' comments and thereby addressed all my criticism satisfactorily. The revisions include the generation of novel viruses, extensive novel data and new discussion points. Importantly, the additional data further support the validity and functionality of the novel oncolytic adenovirus-MATE therapeutic approach. In conclusion, the manuscript has strongly improved in quality and clarity and is recommended for publication. Minor point: During finalization of the manuscript, the authors might consider to include the very helpful clarifications and explanations of their responses to my comments 2, 3, 6 and minor 1 into their manuscript to make them available to the researchers interested in the study.

Point by point response

Editorial requests are in italics, our responses are below

We have now received the reports from the three initial referees. As you will see from the reports below, while referees #1 and #3 are overall satisfied with the revisions, referee #2 still raises concerns on the model used (MC38 murine colon tumor model in syngeneic C56BL/6 mice). I have further consulted with the referees on this point, and they all agreed that no additional experiments should be requested at this stage (considering also the general lack of good model for immunotherapy with human oncolytic adenoviruses), however the limitations of the model system should be adequately discussed in the final manuscript.

We have included a detailed discussion on the limitations of mice and the particular advantages of the syrian hamster model to investigate adenovirus-based oncolytics including several new citations describing some outstanding studies that have been already proven the benefit of this model (lines 622-636). Cell line numbering refer to the document in track change mode.

Moreover, in a minor round of revisions, please address the following points:

1/ Please consider the minor comment from referee #3 on including information in the manuscript.

We have included parts of the corresponding explanations on the reviewer's previous comments 2 (see lines 139-145), 3 (see lines 558 - 577), 6 (see lines 637-649), and minor 1 (see lines 228 - 230).

2/ Manuscript text:

- Please indicate in track changes mode any new modification.

done

- We noted the following name discrepancies: Bettina Mundt in the manuscript file vs. Bettina Fleischmann-Mundt in the submission system; Valery Volk in the manuscript vs. Valer Volk in the submission system. Please clarify.

Bettina Mundt and Valery Volk is correct. We apologize, but during our first submission it was impossible to type the 'y' of Valery Volk into the EMBO-MM input form due to unknown reason.

- Please provide up to 5 keywords.

Have been provided

- The manuscript sections should be in the following order: Title page - Abstract & Keywords - Introduction - Results - Discussion - Methods - Data Availability - Acknowledgments - Disclosure Statement & Competing Interests - References - Figure Legends - (Main Tables with legends) - Expanded View Figure Legends.

Has been considered

- Methods:

o Thank you for providing a reagents and tools table, please remove it from the manuscript, and upload it as a separate file.

done

o Please remove "All sequences and cloning details can be provided upon request" (genetic engineering of MATEs), as per journal policy, all necessary information should be available in the manuscript.

We have replaced these sentences and have provided a detailed description of cloning procedures

o Antibodies: please provide dilutions/concentrations for all experiments.

Have been provided

o Cell lines: please indicate whether the cells were authenticated and tested for mycoplasma contamination.

We have included a statement regarding cell line authentication and mycoplasma control.

o Animal experiments: please provide the origin and gender of the mice, as well as housing and husbandry conditions.

Has been included in the methods chapter 'animal experimentation'

o Human samples: please include a statement confirming that informed consent was obtained from all subjects and that the experiments conformed to the principles set out in the WMA Declaration of Helsinki and the Department of Health and Human Services Belmont Report.

An according passage has been included in the ethics part of Material and Methods chapter

o Statistics: please provide information on blinding, randomization, sample size and inclusion/exclusion criteria.

An according passage has been included in the animal experimentation part of the methods chapter

- Acknowledgments: the information provided here and in the submission system should match; Claudia-von-Schilling-Stiftung is missing in the submission system.

Has been aligned

- Author contributions: CRediT has replaced the traditional author contributions section because it offers a systematic machine readable author contributions format that allows for more effective research assessment. Please remove the Authors Contributions from the manuscript and use the free text boxes beneath each contributing author's name in our system to add specific details on the author's contribution. More information is available in our guide to authors.

Has been removed from the manuscript

- Disclosure and competing interest statement: please provide more information regarding the patents (who and what).

We have included the authors's contributions to patent applications into the disclosure and competing interest statement.

3/ Figures and Appendix:

- We do not accept figures in ppt format. Please upload them as EPS, TIF(F) or PDF format (1 file per figure).

- The Appendix should be in PDF format; the legend should be removed from the manuscript and provided after the figure in the Appendix file; the file should have a table of content with page number; the reference in the manuscript text needs to be updated to Appendix Figure S1 (there is one instance of "Supplemental Fig 1").

- Figure/image re-use should be indicated in the legend (i.e. Figure 5C and Figure EV4G).

Format guidelines have been considered. Reuse of Fig. 5C images on CD3 and CD8 staining in the Ad5/11-monomer in figure EV4G has been indicated in the legend of Figure EV4G.

- Please address the queries from our copy editors in the figure legends:

*- Please define the annotated p values */**/**** as well as provide the exact p-values for the same in the legend of figure EV 4c, f; as appropriate.*

We have provided exact p-values instead of asterisks and have removed the definition of the asterisks. We have provided exact p values when statistical significance levels have been reached.

- Please note that the exact p values are not provided in the legends of figures 1c-e, g-h; 2d, g; 3c; 8c, g; EV 1c, g; EV 5c.

Have all been included

- Please indicate the statistical test used for data analysis in the legends of figures EV 3h; EV 5h.

Has been indicated

- Please note that information related to *n* is missing in the legends of figures 4c, f, h; 5a; EV 3b; EV 4d.

Has been included

- Although '*n*' is provided, please describe the nature of entity for '*n*' in the legends of figures 2d-g; 3c; 5d, f; 6c-d; 7a-c; 8b-d, f-h; EV 1c-d; EV 2c; EV 3h; EV 4c, f; EV 5b, f, h.

Has been clarified

- Please note that the error bars are not defined in the legends of figures EV 3b; EV 4d; EV 5h.

Have been defined

- Please note that the measure of center for the error bars needs to be defined in the legends of figures 2d-g; 5b, f; 6b-c; 7b-e; 8b-c, f-g; EV 1c-d, g-h; EV 2c.

Has been defined

- Please note that the scale bar is missing for figure 8a.

Has been added

- Please note that the scale bar needs to be defined for figure EV 3g.

Has been defined

- Please note that scale bar and its definition are missing for figures EV 3a; EV 5g.

Has been added

- Please note that the white arrows are not defined in the legend of figure 8e. This needs to be rectified.

Have been defined

4/ At EMBO Press we ask authors to provide source data for the main figures. Our source data coordinator has been in touch with you to discuss which figure panels we would need source data for and will also provide you with helpful tips on how to upload and organize the files.

5/ Checklist:

- please clarify which restrictions apply to your work

Has been clarified

- please adjust the section on authentication and mycoplasma checks

Done

- please check the section "human research participants"

checked

- please fill in the whole section "Experimental study design and statistics"

Has been filled-in

- please fill in "sample definition and in-laboratory replication", first entry

Has been filled-in

- please fill in "Ethics"/"informed consent" (second entry)

Has been filled-in

6/ Thank you for providing The Paper Explained. Please include it in the manuscript file.

We have positioned this section at the end of the manuscript file

7/ Synopsis:

- Text: I slightly edited your synopsis to match our style and format. Please let me know if you agree with the following or amend as you see fit:

"Oncolytic adenoviruses were designed to reach potent intratumoral T-cell activation via expression of bifunctional-T-cell engagers (MATEs). MATEs consist of a CD3 binding domain to bind T cells and a protein transduction domain (Tat-PTD) to enable target-independent attachment to the cell surface.

- In vitro, MATEs promote potent activation of murine and human T cells, induce their proliferation and trigger T cell-mediated bystander toxicity.*
- MATEs can be expressed by an oncolytic adenovirus and are secreted by cells upon virus infection.*
- Tumor infection with MATE-expressing oncolytic adenoviruses leads to T cell activation and tumor immunoactivation in murine tumor models, and results in a CD8-dependent therapeutic benefit.*
- Intratumoral T cell activation by MATE-expressing oncolytic adenovirus does not result in systemic toxicity or a cytokine release syndrome.*
- Intratumoral T cell activation by MATE-expressing oncolytic viruses sensitizes tumors to PD-1 checkpoint inhibitor therapy."*

I agree with your editing. Thanks a lot!

- Visual abstract: please upload as jpeg/tiff/png format, 550 px wide x 300-600 px high. Please make sure the text remains legible. If you used an image database for scientific iconography (e.g., BioRender), let us know if you have a license that allows for publication in an academic journal, and reference the image database in the methods section. A cropped portion (115px x 70px) of this image will serve as thumbnail for the table of content on our webpage.

BioRender was used with an institutional license allowing for publication. BioRender is cited in the Reagents and Tools table. We followed the guidelines for the visual abstract.

8/ As part of the EMBO Publications transparent editorial process initiative (see our Editorial at <http://embomolmed.embopress.org/content/2/9/329>), EMBO Molecular Medicine will publish online a Review Process File (RPF) to accompany accepted manuscripts.

This file will be published in conjunction with your paper and will include the anonymous referee reports, your point-by-point response and all pertinent correspondence relating to the manuscript. Let us know whether you agree with the publication of the RPF and as here, if you want to remove or not any figures from it prior to publication.

I agree with publication of the RPF.

In addition to the reviewers and editorial requirements we have again checked the figures and the underlying source data and have made the following modifications when compared to our previous submission:

Fig 5 C: We have discovered a tiny arrow in the submitted figure which we have now removed in a repaired new figure 5,

Fig 6 D: We found that representative images had not been accurately attributed which was corrected now. The scientific conclusion of the figure is not altered by this modification.

Fig 8 B: We found that the y-axis labelling in the middle panel and the right hand panel had been in a different order compared with neighboring panels. We have switched this now. The scientific conclusion has not been altered by this modification.

4th Dec 2024

Dear Prof. Kuehnel,

Thank you for submitting your revised study. I will be able to accept your manuscript once the following editorial matters are addressed:

1/ Thank you for addressing referee #3 comments. However, in doing so, you included "Data not shown", which is not allowed in our journal. As per our guidelines, all data referred to in the manuscript should be displayed in the main or supplementary figures. Please adjust accordingly.

2/ Please make sure there is not potential infringement on the Amgen's trademark for the term "BiTE".

3/ Methods:

- Cell lines: please also indicate in the methods whether the cells were tested for mycoplasma contamination.
- Human samples: please include a statement confirming that informed consent was obtained from all subjects.
- Statistics: please add a statement on sample size and inclusion/exclusion criteria.

4/ Thank you for completing the checklist and competing interest statements. Regarding restrictions that may apply to your work, please note that as per journal policy, it is understood that by publishing a paper in EMBO Molecular Medicine, the authors agree to make available to colleagues in academic research all new reagents, including organisms (or means to produce them), viruses, cells, nucleic acids and antibodies, that were used in the research reported and that are not available from public repositories or commercial suppliers.

5/ Please structure your Paper Explained as follows:

- medical issue
- results
- clinical impact

This may be edited to ensure that readers understand the significance and context of the research.

6/ Synopsis: I have cropped the attached portion of your image to serve as a thumbnail for the table of content on our webpage. Please let me know if you agree or provide an alternative image (115px x 70px) as changes at proofing stage are usually not allowed.

I look forward to receiving your revised manuscript.

Yours sincerely,

Lise Roth

The authors addressed the remaining editorial issues.

16th Dec 2024

Dear Prof. Kuehnel,

Thank you for submitting your revised files and for addressing the remaining minor editorial concerns. I am pleased to inform you that your manuscript is accepted for publication and is now being sent to our publisher to be included in the next available issue of EMBO Molecular Medicine.

If you have any questions, please do not hesitate to contact the Editorial Office. Thank you for your contribution to EMBO Molecular Medicine!

With my best wishes for the holiday season,

Lise Roth
